**Quantifying the migration rate of drainage divides from**
**high-resolution topographic data**
Chao Zhou[1], Xibin Tan[2, *], Yiduo Liu[2], Feng Shi[1, 3]
[1] State Key Laboratory of Earthquake Dynamics, Institute of Geology, China
Earthquake Administration, Beijing 100029, China
[2] Key Laboratory of Mountain Hazards and Surface Processes, Institute of Mountain
Hazards and Environment, Chinese Academy of Sciences, Chengdu 610299, China
[3] Shanxi Taiyuan Continental Rift Dynamics National Observation and Research
Station, Beijing 100029, China
*Corresponding author. E-mail address: tanxibin@imde.ac.cn

# Abstract

12        The lateral movement of drainage divides is co-influenced by tectonics,

lithology, and climate, and therefore archives a wealth of geologic and climatic
information. It also has wide-ranging implications for topography, the sedimentary
record, and biological evolution, thus has drawn much attention in recent years.
Several methods have been proposed to determine drainage divides' migration state
(direction and rate), including geochronological approaches (e.g., $^{10}$Be) and
topography-based approaches (e.g., χ-plots or Gilbert metrics). A key object in these
methods is the channel head, which separates the hillslope and channel. However, due
to the limited resolution of topography data, the required channel-head parameters in
the calculation often cannot be determined accurately, and empirical values are used

in the calculation, which may induce uncertainties. Here, we propose two methods to

calculate the migration rate of drainage divides, based on the relatively accurate

channel-head parameters derived from high-resolution topographic data. We then

apply the methods to an active rift shoulder (Wutai Shan) in the Shanxi rift, and a

tectonically stable area (Yingwang Shan) in the Loess Plateau, to illustrate how to

calculate drainage-divide migration rates. Our results show that the Wutai Shan

drainage divide is migrating northwestward at a rate between 0.21 to 0.27 mm/yr,

whereas the migration rates at the Yingwang Shan are approximately zero. This study

indicates that the drainage-divide stability can be determined more accurately using

high-resolution topographic data. Furthermore, this study takes the cross-divide

differences in the uplift rate of channel heads into account in the measurement of

drainage-divide migration rate for the first time.

## Keywords

Drainage divide; Migration rate; High-resolution topographic data; DEM; Channel

head

# 1. Introduction

The evolution of the Earth's surface is jointly controlled by tectonics, lithology,

and climatic conditions (e.g., Molnar and England, 1990; Whipple, 2009; Gallen,

2018; Bernard et al., 2021; Hoskins et al., 2023), providing a basis for reconstructing

the past tectonic (Pritchard et al., 2009; Kirby and Whipple, 2012; Shi et al., 2021) or

climatic processes (Tucker and Slingerland, 1997; Hancock et al., 2002; Schildgen et
al., 2022) through topography. The evolution of unglaciated terrestrial terrains is
fundamentally coupled with changes in drainage systems through river's vertical
(changes in river long profile) and lateral movements (drainage divide migration and
river captures) (Whipple, 2001; Clark et al., 2004; Bonnet, 2009; Willett et al., 2014).
Previous studies have extensively investigated how river channel profiles respond to
tectonic uplift (Whipple, 2001; Crosby and Whipple, 2006; Kirby and Whipple,
2012), lithological differences (Duvall et al., 2004; Safran et al., 2005; Forte et al.,
2016), and precipitation perturbations (Schlunegger et al., 2011; Bookhagen and
Strecker, 2012). River long profiles have been used to study earthquake events (e.g.,
Burbank and Anderson, 2001; Wei et al., 2015) and the spatio-temporal variations of
uplift (e.g., Whipple et al., 1999; Kirby et al., 2003; Pritchard et al., 2009; Goren et
al., 2014). Recent studies show that the widespread lateral movement of river basins
driven by geological and/or climatic disturbance (Yang et al., 2019; Zondervan et al.,
2020; Zhou et al., 2022a; Bian et al., 2024) also interacts with the adjustment of
channel profiles (Willett et al., 2014). Drainage-divide migration, one form of river
lateral movement, may not only carry information on geological and/or climatic
disturbance (Su et al., 2020; Zondervan et al., 2020; He et al., 2021; Shi et al., 2021;
Zhou et al., 2022a; Zeng and Tan, 2023) but also influence the extraction of tectonic
information from channel profiles (Goren et al., 2014; Ma et al., 2020; Jiao et al.,
2022). Moreover, it has multi-facet consequences for landscape evolution
(Scheingross et al., 2020; Stokes et al., 2023), sedimentary processes (Clift &
Blusztajn, 2005; Willett et al., 2018; Deng et al., 2020; Zhao et al., 2021), and
biological evolution (Waters et al., 2001; Zemlak et al., 2008; Hoorn et al., 2010;
Musher et al., 2021). For this reason, the stability of drainage divides has drawn more
and more attention in recent years (e.g., Authemayou et al., 2018; Vacherat et al.,
2018; Chen et al., 2021; Shelef and Goren, 2021; Sakashita and Endo, 2023; Bian et
al., 2024).

Drainage-divide migration is essentially controlled by the cross-divide

difference in erosion and topographic slope (Beeson et al., 2017; Dahlquist et al.,
2018; Chen et al., 2021; Zhou et al., 2022a). The erosion rates are routinely derived
from geochronological techniques, such as cosmogenic nuclides (e.g., $^{10}$Be)
concentration measurements (Mandal et al., 2015; Struth et al., 2017; Young and
Hilley, 2018; Sassolas-Serrayet et al., 2019), which can be used to calculate the
migration rates of drainage divides (Beeson et al., 2017; Godard et al., 2019; Hu et al.,
2021). However, these techniques are usually based on samples collected from a
catchment outlet that is several, or even tens of, kilometers away from the drainage
divide and thus may not represent the erosion rates close to the drainage divide
(Sassolas-Serrayet et al., 2019; Zhou et al., 2022a). Besides, the high cost of sample
processing makes it challenging to determine the drainage divide's motion by
measuring the erosion rates throughout the large landscapes. Hence, it would be ideal
to find an accessible and efficient method that can be applied to the entire landscape
and make full use of the $^{10}$Be-derived erosion rates.

The advancement of the digital elevation model (DEM) has promoted the

development of geomorphic analysis, making it possible to determine the drainage
divide's transient motion through topography analysis. For example, Willett et al.
(2014) applied the χ method to map the dynamic state of river basins. Forte and
Whipple (2018) proposed the cross-divide comparison of "Gilbert metrics" (including
channel heads' relief, slope, and elevation) to determine a drainage divide's migration
direction. Others adopted the comparison of slope angle or relief of the hillslopes
across a drainage divide to deduce its stability (Scherler and Schwanghart, 2020; Ye et
al., 2022; Zhou et al., 2022b). These geomorphic techniques, so far, could only
determine the migration direction of drainage divides. Braun (2018) provided an
equation that considers both alluvial and fluvial areas to calculate the migration
velocity of an escarpment (also a drainage divide). Zhou et al. (2022a) developed a
technique to calculate the migration rate through the high base-level χ values on both
sides of a drainage divide. These new approaches require channel-head parameters to
calculate the migration rate. However, the location of the channel heads sometimes
cannot be accurately identified because of the limitation in the resolution of DEMs in
natural cases. For this reason, empirical values of channel-head parameters are used in
these studies, which may induce uncertainties.
This study aims to establish an approach to derive the migration rate of drainage
divides, at a high precision and low cost, based on topographic analysis. We choose a
tectonically active area (i.e., the Wutai Shan in the Shanxi Rift) and a tectonically
inactive area (i.e., the Yingwang Shan in the Loess Plateau) to demonstrate how to
quantify drainage-divide migration rates (Fig. 1). We use the aerial photography
acquired by unmanned aerial vehicles (UAVs) and the Structure from Motion (SfM)
technology to obtain the high-resolution DEM data of these two areas (0.67 m and
0.84 m spatial resolution in the Wutai Shan and the Yingwang Shan, respectively).
Benefiting from the high-resolution data, the location of channel heads can be
identified more accurately. We then develop two methods to calculate the drainage-
divide migration rates. One is based on the measured channel-head parameters, and
the other is based on an improved method of Zhou et al (2022a). Combining with the
geological and low-temperature thermochronology studies of the Wutai Shan
(Middleton et al., 2017; Clinkscales et al., 2020), we also quantify the cross-divide
difference in uplift rates to improve the precision of drainage-divide migration rate.

## 2. Methods

### 2.1 Channel-head-point method

According to the detachment-limited stream power model (Howard and Kerby,
1983; Howard, 1994), the channel's erosion rate ($E$) can be expressed as:

$$E = KA^m S^n \qquad (1)$$

where $K$ is the erosion coefficient, $A$ is the upstream drainage area, $S$ is the gradient of
the river channel, and $m$ and $n$ are empirical constants.
Because of thresholds such as erosion threshold (the shear stress of overland flow
must exceed the threshold of the cohesion of bed material to generate river incision)
(Howard and Kerby, 1983; Perron et al., 2008) or landslide threshold (landslides
occur when the threshold of soil or rock strength is exceeded in high relief region)
(Burbank et al., 1996; Tucker and Bras, 1998), river channels (following Eq. 1)
emerge at a certain distance from the drainage divide. The region between the channel
head and the drainage divide is referred to as the hillslope area, where the erosion is
controlled by landslide, collapse, and diffusion processes (Carson and Kirkby, 1972;
Stock and Dietrich, 2006; Stark, 2010; Braun et al., 2018; Dahlquist et al., 2018). The
channel-head point is the highest and the closest point to the drainage divide on a
river channel (Clubb et al., 2014). Therefore, the erosion rate at channel-head points
($E_{ch}$) can be described as:
$$E_{ch} = K A_{cr}^m S_{ch}^n \qquad (2)$$

where $E_{ch}$ is the erosion rate at channel-head points, $A_{cr}$ is the critical upstream
drainage area of a channel-head point (Duvall et al., 2004; Wobus et al., 2006), and
$S_{ch}$ is the channel-head gradient measured along the channel near the channel-head
point. Eq. 2 indicates that the side of a drainage divide with a higher $A_{cr}$ or $S_{ch}$ can
have a higher erosion rate than the other side, and is more likely to pirate the opposite
drainage basin. Besides, a high erosion coefficient can amplify the drainage basin's
erosion rate.

Drainage-divide migration is essentially controlled by the cross-divide difference

in erosion rates and topographic slope (Beeson et al., 2017; Dahlquist et al., 2018;
Chen et al., 2021; Zhou et al., 2022a; Stokes et al., 2023). Furthermore, the
differential uplift should also be considered when using the cross-divide erosion rates
at the channel heads to calculate the erosion difference across the divide, especially in
the case of tectonic tilting uplift (Zhou et al., 2022a). The drainage-divide migration
rate ($D_{mr}$) can be obtained according to the cross-divide difference in erosion rate and
uplift rate and the slopes across the divide (Zhou et al., 2022a):

$$D_{mr} = \frac{\Delta E_{ch} - \Delta U_{ch}}{\tan\alpha + \tan\beta} \tag{3}$$

where $\Delta E_{ch}$ is the difference in erosion rate between the two sides (annotated as $\alpha$ and
$\beta$) of the drainage divide ($\Delta E_{ch} = E_{ch\alpha} - E_{ch\beta}$). The choice of $\alpha$ or $\beta$ is arbitrary, and the
positive direction of the migration rate is assigned from the $\alpha$ to the $\beta$ side whereas the
negative is the opposite. $\Delta U_{ch}$ is the cross-divide difference in uplift rate ($\Delta U_{ch} = U_{ch\alpha}$
$- U_{ch\beta}$), and $\tan\alpha$ and $\tan\beta$ are the average gradients (along the normal-divide
direction) upslope of the channel head (not including the hilltop part) on the $\alpha$ side
and the $\beta$ side, respectively. Assuming the erosion coefficient ($K$) is the same on both
sides of a drainage divide, Eqs. 2 and 3 allow us to derive the equation of drainage
divide's migration rate according to the parameters at the channel-head points:

$$D_{mr} = \frac{K\left[(A_{cr}^m S_{ch}^n)_\alpha - (A_{cr}^m S_{ch}^n)_\beta\right] - \Delta U_{ch}}{\tan\alpha + \tan\beta} \tag{4}$$

If the exact value of $K$ is unknown, the drainage divide's unilateral erosion rate
can be used as a substitution:

$$D_{mr} = \frac{E_\alpha\left[1 - \frac{(A_{cr}^m S_{ch}^n)_\beta}{(A_{cr}^m S_{ch}^n)_\alpha}\right] - \Delta U_{ch}}{\tan\alpha + \tan\beta} \tag{5}$$

or:

$$D_{mr} = \frac{E_\beta\left[\frac{(A_{cr}^m S_{ch}^n)_\alpha}{(A_{cr}^m S_{ch}^n)_\beta} - 1\right] - \Delta U_{ch}}{\tan\alpha + \tan\beta} \tag{6}$$

$E_\alpha$ and $E_\beta$ are the erosion rates of the $\alpha$ and the $\beta$ side of the drainage divide,
respectively, which can be derived through cosmogenic nuclides ([10]Be) concentration
measurements (Beeson et al., 2017; Godard et al., 2019; Hu et al., 2021). The regional
average erosion rate ($\bar{E} = \frac{E_\alpha + E_\beta}{2}$) can also be used to calculate the migration rate:
$$D_{mr} = \frac{2\bar{E}\left[\frac{(A_{cr}^m S_{ch}^n)_\alpha - (A_{cr}^m S_{ch}^n)_\beta}{(A_{cr}^m S_{ch}^n)_\alpha + (A_{cr}^m S_{ch}^n)_\beta}\right] - \Delta U_{ch}}{tan\alpha + tan\beta} \tag{7}$$

Based on Eqs. 4-7, the migration rate of drainage divides can be estimated using
channel-head parameters combined with one of the erosion-related parameters,
erosion coefficient ($K$), erosion rate at one side of a drainage divide ($E_\alpha$ or $E_\beta$), or
regional average erosion rate ($\bar{E}$).

## 2.2 Channel-head-segment method

A channel-head segment is the channel segment just below the channel head

(Zhou et al., 2022a). Zhou et al. (2022a) developed a method based on the cross-
divide $\chi$ contrast of channel-head segments to calculate the migration rate of drainage
divides. The essence of the method is the cross-divide comparison of the channel-
head segments' normalized channel steepness ($k_{sn}$) values. $k_{sn}$ is a widely used index
(Whipple et al., 1999; Wobus et al., 2006; Hilley and Arrowsmith, 2008; Kirby and
Whipple, 2012) that is quantitatively related to $E$ and $K$ ($k_{sn} = \left(\frac{E}{K}\right)^{\frac{1}{n}}$). $\chi$ is an integral
function ($\chi = \int_{x_b}^{x} \left(\frac{A_0}{A(x)}\right)^{\frac{m}{n}} dx$) of a channel's upstream area ($A$) to horizontal distance
($x$) (Royden et al., 2000; Perron and Royden, 2012), and $A_0$ is an arbitrary scaling area
to make the integrand dimensionless.

In the method of Zhou et al. (2022a), the location of channel heads cannot be

accurately identified, because it is limited by the resolution of DEM. Therefore, an
empirical value of $A_{cr} = 10^5$ m$^2$ was used in the calculation. Benefiting from the high-
resolution DEM in this study, we improve the method in Zhou et al. (2022a) and use
the real location of channel heads to calculate the migration rate. When the regional
erosion coefficient ($K$) is known and unchanged in the vicinity of the drainage divide,
the drainage-divide migration rate can be estimated by the following equation:
$$D_{mr} = \frac{K\left[k_{sn(\alpha)}{}^n - k_{sn(\beta)}{}^n\right] - \Delta U_{ch}}{\tan\alpha + \tan\beta} = \frac{K\left\{\left[\frac{(z_{ch}-z_b)_\alpha}{\chi_\alpha}\right]^n - \left[\frac{(z_{ch}-z_b)_\beta}{\chi_\beta}\right]^n\right\} - \Delta U_{ch}}{\tan\alpha + \tan\beta} \tag{8}$$

where $z_{ch}$ is the elevation of the channel head, $z_b$ is the elevation of catchment outlet
(at the top part of the channel to make the elevation-$\chi$ profiles quasi-linear between
the channel head and the outlet), and subscripts $\alpha$ and $\beta$ denote the two rivers across a
divide. The detailed derivation of Eq. 8 is in Supplementary Materials. The drainage
divide's unilateral erosion rate ($E_\alpha$ or $E_\beta$) can also be used as a substitution for the $K$
value:
$$D_{mr} = \frac{E_\alpha\left\{1 - \left(\frac{\chi_\alpha}{\chi_\beta}\right)^n \left[\frac{(z_{ch}-z_b)_\alpha}{(z_{ch}-z_b)_\beta}\right]^{-n}\right\} - \Delta U_{ch}}{\tan\alpha + \tan\beta} \tag{9}$$

or:
$$D_{mr} = \frac{E_\beta\left\{\left(\frac{\chi_\alpha}{\chi_\beta}\right)^{-n} \left[\frac{(z_{ch}-z_b)_\alpha}{(z_{ch}-z_b)_\beta}\right]^n - 1\right\} - \Delta U_{ch}}{\tan\alpha + \tan\beta} \tag{10}$$

Alternatively, one can use the regional average erosion rate ($\bar{E}$) to calculate the
migration rate:
$$D_{mr} = \frac{2\bar{E}\left\{\frac{\left[\frac{(z_{ch}-z_b)_\alpha}{(z_{ch}-z_b)_\beta}\right]^n - \left(\frac{\chi_\alpha}{\chi_\beta}\right)^n}{\left[\frac{(z_{ch}-z_b)_\alpha}{(z_{ch}-z_b)_\beta}\right]^n + \left(\frac{\chi_\alpha}{\chi_\beta}\right)^n}\right\} - \Delta U_{ch}}{\tan\alpha + \tan\beta} \tag{11}$$

Based on Eqs. 8-11, the drainage-divide migration rate can be estimated using the $\chi$
values of high-base-level channel segments combined with one of the erosion-related
parameters, erosion coefficient ($K$), erosion rate at one side of a drainage divide ($E_\alpha$ or
$E_\beta$), or regional average erosion rate ($\bar{\bar{E}}$).

## 2.3 Parameter extraction

In this study, we apply the erosion coefficient ($K$) related equations (Eqs. 4 & 8)

to two natural examples in North China, the Wutai Shan in the Shanxi Rift and the
Yingwang Shan in the Loess Plateau, to demonstrate how to calculate the drainage-
divide migration rates (Fig. 1). We calculated the $K$, according to the equation, $K =$
$\frac{E}{k_{sn}{}^n}$, the erosion rate obtained by chronological methods, the $k_{sn}$, and the assumed
slope exponent ($n = 1$). The $k_{sn}$ is calculated based on $S$ and $A$ ($k_{sn} = SA^{\frac{m}{n}}$) extracted
from ALOS DEM (downloaded from https://search.asf.alaska.edu/) using
TopoToolbox (Schwanghart and Scherler, 2014), and the interpolation uses the
Kriging method on ArcGIS (Fig. 2). We use a small four-rotor Unmanned Aerial
Vehicle (UAV), the DJI Phantom 4, to acquire stereo images of the areas. Based on
the Structure-from-Motion (SfM) method and PhotoScan software, we obtained the
DEMs with a spatial resolution of 0.67 m in the Wutai Shan and 0.84 m in the
Yingwang Shan (can be download from https://doi.org/10.5069/G98C9TGT). Both
regions are semi-arid, and the vegetation is dominated by shrubs. We did not compare
the elevations to the standard GPS points, which may bring errors on the elevations.

Based on the high-resolution topography data, we first extract river channels and

drainage divide, using a single-flow-direction algorithm (D8). Then we extract the
relevant parameters, and calculate the drainage-divide migration rate. Data analysis
including slope-area plots, $\chi$-plots, river's long profiles and topographic swath
profiles, are based on the Matlab toolbox TAK (Forte and Whipple, 2019) and
TopoToolbox (Schwanghart and Scherler, 2014). According to the breaking point of
the slope-area regression line, we obtain the value of the critical upstream drainage
area ($A_{cr}$) of each river channel (Duvall et al., 2004). According to these values, we
mark the position (and its elevation, $z_{ch}$) of the channel heads on the $\chi$-plots and the
topography map. An elevation of the catchment outlet ($z_b$) can be assigned at the top
part of the channel to make the elevation-$\chi$ profiles quasi-linear between the channel
head and the outlet. The slope of the channel head ($S_{ch}$) is calculated, according to the
100 m long channel on the river's long profiles around the channel head (50 m
upstream and downstream). Topographic gradient ($\tan\alpha$ or $\tan\beta$) is calculated through
the average slope (in the normal-divide direction) of the hillslope segment (not
including the hilltop part, because of its lower gradient). The cross-divide uplift
difference in the channel-head points ($\Delta U_{ch}$) is estimated according to the location of
the each channel head and the tectonic uplift trend.

# 3. Applications to natural cases

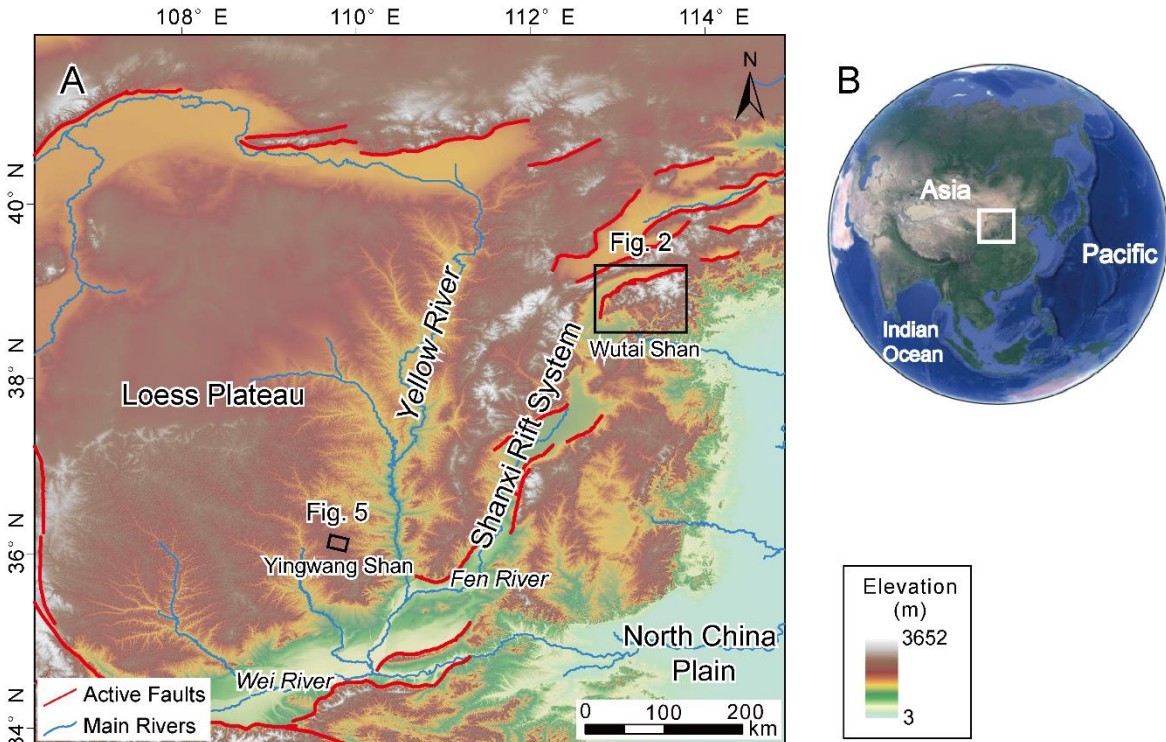

**Figure 1**. Locations and tectonic background of the two nature cases in North China. The figure is modified from Fig. 7 in Shi et al. (2021). (**A**) Red lines represent the main active faults. Black rectangles show the locations of the two nature cases. Red curve denotes active fault, sourced from https://www.activefault-datacenter.cn/. The topography data (ALOS DEM) is downloaded from the Alaska Satellite Facility (ASF) Data Search (https://search.asf.alaska.edu/). (**B**) The satellite image downloaded from Google Earth. White rectangles show the location of Panel A.

## 3.1 Wutai Shan

The Wutai Shan is a tilted fault block on the shoulder of the Shanxi Rift System located in the central North China craton (Fig. 1) (Xu et al., 1993; Su et al., 2021).

The tilting uplift of the Wutai Shan is controlled by the Northern Wutai Shan fault,
and there is no active fault along the south edge of the Wutai Shan horst (Fig. 2). The
bedrock of the Wutai Shan area consists mainly of metamorphic and igneous
basement rocks (Clinkscales et al., 2020) and there is no obvious variation in rock
erodibility and precipitation in this area (Fig. S2 & S3). Zhou et al. (2022b) reveal that
the Wutai Shan drainage divide is migrating northwestward due to the tilting uplift
and predicts the drainage divide will move ~10 km to the northwest to achieve a
steady state if all geological conditions remain. Geomorphic evidence also exhibits a
northwestward migration of the drainage divide (Fig. 3). The plan and satellite views
show several abnormally high junction angles around the Wutai Shan drainage divide,
which indicate that the tributaries formerly part of the northern drainage have become
part of the southern drainage (Fig. 3A&B). The χ-plots analysis shows the southern
side of the drainage divide has steeper channels, higher $k_{sn}$, and lower χ. The χ-plots
of paired rivers illustrate obvious characteristics of shrinking-expanding and captured-
beheaded rivers (Fig. 3C).

To derive the erosion coefficient of the Wutai Shan area, we calculate the

channel steepness ($k_{sn}$) of this region, assuming $n = 1$ and $m = 0.45$ (Wobus et al.,
2006; DiBiase et al., 2010; Perron and Royden, 2012; Wang et al., 2021). We then use
the Kriging interpolation method to generate the $k_{sn}$ distribution map (Fig. 2B). In
addition, results under the assumptions of $m = 0.35$ and 0.55, respectively, are shown
in Supplementary Materials (Fig. S4). The average $k_{sn}$ value of the upthrown side near
the Northern Wutai Shan fault is ~80 $m^{0.9}$ (Fig. 2D). Middleton et al. (2017) showed

that the Quaternary throw rates of the Northern Wutai Shan fault are 0.8-1.6 mm/yr.

Clinkscales et al. (2020) showed, using low-temperature thermochronology, that the

time-averaged long-term throw rates in the late Cenozoic is about 0.25 mm/yr, and

there is an accelerated activity in the Wutai Shan area. According to these studies, we

assume a $0.50 \pm 0.25$ mm/yr uplift/erosion rate in the northern margin of the Wutai

Shan (in the footwall of the Northern Wutai Shan fault). Combining with the equation,

$K = \frac{E}{k_{sn}{}^n}$, and following the approach of previous studies (Kirby and Whipple, 2001;

Kirkpatrick et al., 2020; Ma et al., 2020), the erosion coefficient ($K$) is calculated to

be $(6.25 \pm 3.13) \times 10^{-6}$ m$^{0.1}$yr$^{-1}$ in this area. Because there is no obvious variation in

rock erodibility and precipitation in this area (Figs. S2 & S3), we use this value as the

erosion coefficient ($K$) of the Wutai Shan area.

We then apply the two new methods (Eqs. 4 & 8) to calculate the migration rate

of the drainage divide in the Wutai Shan. We first choose three pairs of rivers (Fig.

4A) and acquire their slope-area plots (Figs. 4B, E, H) and the χ-plots (Figs. 4C, F, I).

According to the breaking point of the slope-area regression line (Duvall et al., 2004)

(Figs. 4B, E, H), we obtain the values of the critical upstream drainage area ($A_{cr}$).

According to these values, we separate hillslope and channel areas and mark the

position of the channel heads on the χ-plots and the topography map (Fig. 4A). For

the χ-plots (Figs. 4C, F, I), we obtain the elevations of channel heads ($z_{ch}$) and χ values

based on the coordinate of the channel-head points. According to the location of the

channel heads on the river's long profiles, we calculate the channel-head gradient

($S_{ch}$). Topographic gradient ($\tan\alpha$ or $\tan\beta$) is calculated through the average slope (in

the normal-divide direction) of the hillslope segment (not including the hilltop part,
Figs. 4D, G, J).

According to the previous studies (Middleton et al., 2017; Clinkscales et al.,

2020) and the $k_{sn}$ distribution (Fig. 2D), we assume the rock uplift rate decreases
linearly from 0.5 to 0 mm/yr from northwest to southeast of the Wutai Shan horst
(~40 km wide). Then we can obtain that the cross-divide uplift difference in the
channel-head points ($\Delta U_{ch}$) (the distance perpendicular to the direction of the
boundary fault is ~600 m) is ~0.008 mm/yr. After determining these parameters, we
adopt the channel-head-point (Eq. 4) and channel-head-segment (Eq. 8) methods,
respectively, to calculate the migration rates. The required data for calculation and the
migration rates are shown in Table 1. The calculated results for $m/n$ = 0.35 and 0.55,
respectively, are shown in Supplementary Materials (Table S1). The migration rates
are higher when $m/n$ = 0.35 and lower when $m/n$ = 0.55, which indicates the $m/n$ value
is sensitive to the result.

The rivers have different characteristics on both sides of the drainage divide, as

illustrated on their slope-area plots (Figs. 4B, E, H) and the χ-plots (Figs. 4C, F, I).
For the first site (Fig. 4D), the migration rates calculated by the channel-head-point
and channel-head-segment methods are 0.21 mm/yr and 0.26 mm/yr, respectively. For
the second site (Fig. 4G), the migration rates are 0.23 mm/yr and 0.27 mm/yr,
respectively. For the third site (Fig. 4J), 0.21 mm/yr and 0.22 mm/yr, respectively. The
drainage divides of all three points are migrating northwestward, which is consistent
with the previous result inferred by the cross-divide contrast of slopes in this area
(Zhou et al., 2022b). Furthermore, the migration rates calculated by the two methods
are comparable in all three sites.

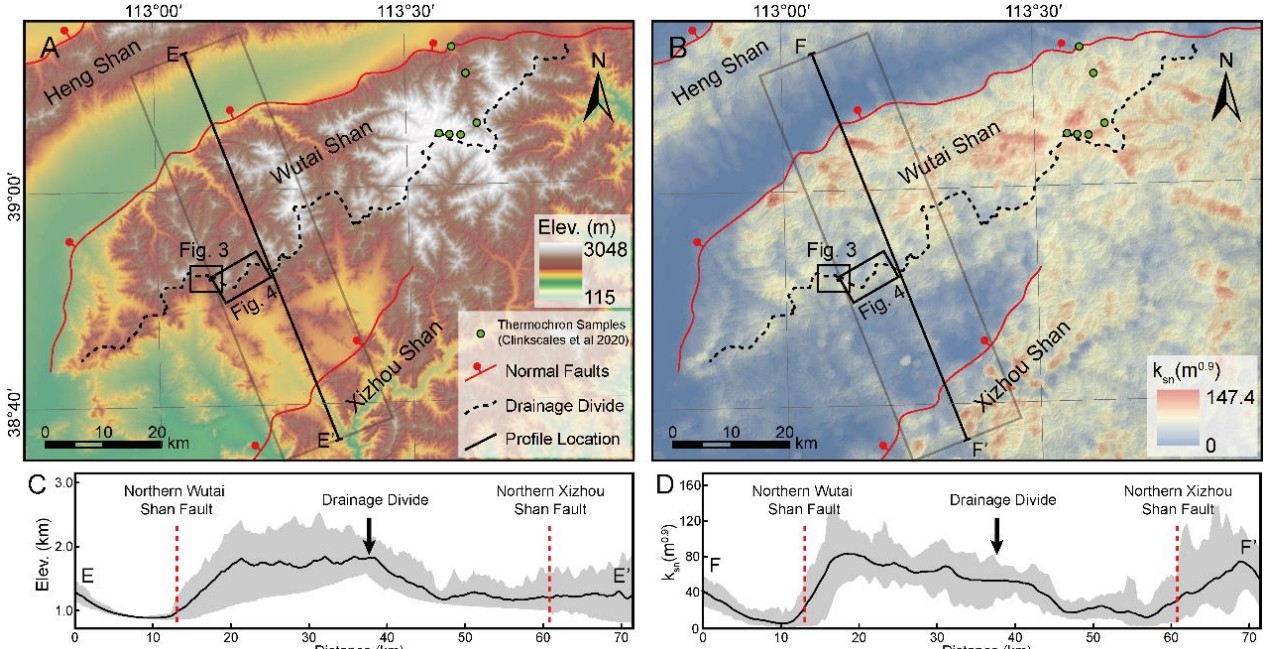

**Figure 2**. Topography (**A**) and normalized channel steepness ($k_{sn}$) (**B**) distribution of
the Wutai Shan horst and surrounding area in the Shanxi Rift System. The black
dashed line shows the location of the main drainage divide. Red lines show the main
active faults. The black lines show the location of profiles E-E' and F-F'. Black
rectangles show the area of Fig. 3B & 4A. Gray boxes show the area of the swath
profiles in Panels C and D. Green dots denote the locations of the low-temperature
thermochronology samples in Clinkscales et al. (2020). The $k_{sn}$ is calculated based on
*S* and *A* extracted from ALOS DEM ($k_{sn} = SA^{\frac{m}{n}}$) and a uniform *m/n* (0.45) using
TopoToolbox (Schwanghart and Scherler, 2014), and the interpolation uses the
Kriging method on ArcGIS. (**C**) Topography swath profile along E-E'. See location in
Panel A. (**D**) $k_{sn}$ swath profile along F-F'. See location in Panel B. The swath profiles

are extracted using TopoToolbox (Schwanghart and Scherler, 2014). The red dashed

lines show the location of the main active normal faults, and the black arrow shows

the location of the main drainage divide. Both swath profiles are 20 km wide (10 km

on each side).

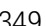

**Figure 3**. Perspective views and χ map of the drainage divide in the Wutai Shan (see

Fig. 2 for location). (A) Perspective views of a captured area and the channels mapped

with $k_{sn}$. The south side of the drainage divide has steeper channels and higher $k_{sn}$ than

the north side. Magenta arrows show drainage divide migration directions. The

satellite image is from Google Earth. (B) χ map of this area with the outlet elevation

of 1300 m. The south side of the drainage divide has lower χ values than the north

side. It should be noted that the catchment outlet at the north side of the drainage

basins (the 1300 m contour) is out of the map. The χ-plots of the rivers in bold lines

are shown in Panel C. (C) χ-plots of the three paired rivers in Panel B. The blue and

red curves correspond to the rivers on the south and north sides, respectively. The χ-

plot of River 1 is steeper on the south side, indicating that the river on the south side

is expanding and the river on the north side is shrinking. The χ-plots of Rivers 2 and 3

in the captured area show obvious characteristics of the captured and beheaded rivers.

The χ-plot is extracted using TAK (Forte and Whipple, 2019) and TopoToolbox

(Schwanghart and Scherler, 2014).

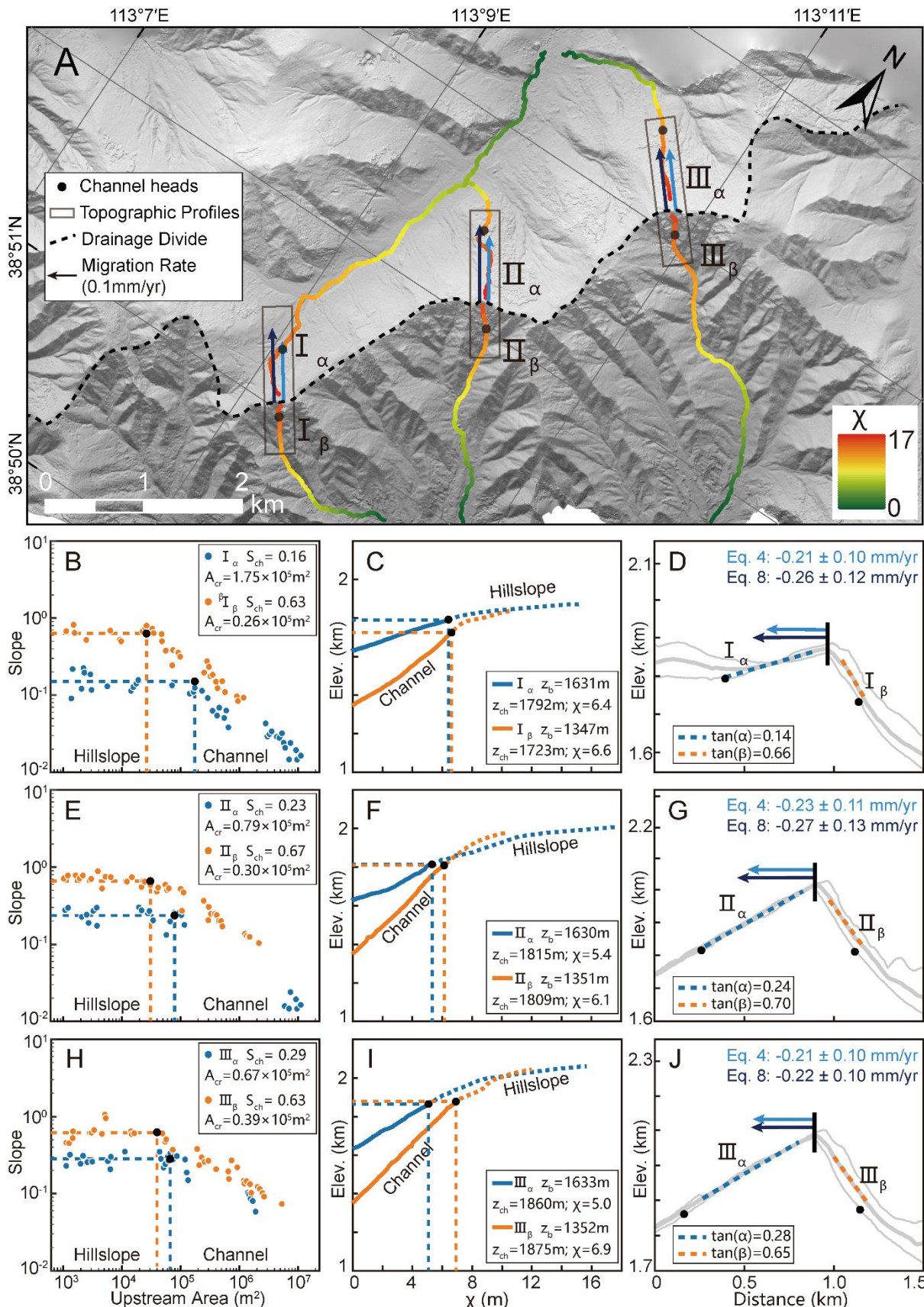

**Figure 4.** Analytical results of the Wutai Shan drainage divide. (**A**) High-resolution

hill-shade map (0.67 m spatial resolution) of the Wutai Shan. The black dashed line

shows the location of the main drainage divide. Colored lines show the three pairs of selected channels used for analysis. The black dots are the channel heads. Black rectangles show the location of the cross-divide topography swath profiles. The black arrows show the direction of drainage-divide migration (**B, E, H**) Slope-area plots of the three pairs of selected channels. The blue and orange dots are the slope-area plots of the north (α) and south (β) sides of the drainage divide respectively. The black dots represent the channel heads. (**C, F, I**) χ-plots of the selected channels. The blue and orange lines are the χ-plots of the north (α) and south (β) sides of the drainage divide respectively. The black dots represent the channel heads. (**D, G, J**) Cross-divide topography swath profiles with the drainage-divide migration rates. The locations of the profiles are in Panel A. The light and dark blue arrows are the drainage-divide migration rates calculated by the channel-head-point (Eq. 4) and channel-head-segment (Eq. 8) methods respectively.

## 3.2 Yingwang Shan

The Loess Plateau is hosted by the tectonically stable Ordos Block of the North China craton (Yin, 2010; Su et al., 2021). Over the past 2.6 million years, it has accumulated tens to hundreds of meters of eolian sediments (Yan et al., 2014), draping preexisting topography (Xiong et al., 2014). There is no active fault and little to no variation in rock erodibility and precipitation within the area (Shi et al., 2020; Zhou et al., 2022b).

We apply the two methods to Yingwang Shan of Loess Plateau to calculate the
drainage-divide migration rate. Similar to the Wutai Shan site, we obtain the slope-
area plots (Figs. 5 B, E, H), the $\chi$-plots (Figs. 5 C, F, I), and extract the values of $A_{cr}$,
$S_{ch}$, $z_b$, $z_{ch}$, $\chi$, tan$\alpha$ and tan$\beta$ of the rivers. The rate of soil erosion in the study area is
about 500 t·km$^{-2}$yr$^{-1}$ according to the distribution of silt discharge (Fu, 1989).
Combining with the assumption of the density of loess, 1.65 t·m$^{-3}$, the present-day
erosion rate in the study area is calculated to be 0.3 mm·yr$^{-1}$. Because there is no
obvious unequal uplift in this region, we assign that $\Delta U_{ch}$ is zero. We also assume $n =$
1 and $m = 0.45$ in the calculation (Wobus et al., 2006; DiBiase et al., 2010; Perron and
Royden, 2012; Wang et al., 2021). Then, we use the methods of channel-head
parameters (Eq. 7) and channel segments (Eq. 11) to calculate the drainage-divide
migration rates. The required data for calculation and the migration rates are shown in
Table 1.
All results of the three points show that the drainage-divide migration rate here is
close to zero, no matter which method is used in the calculation. The results show that
the drainage divide of the study site is in topographical equilibrium, which is
consistent with the inference in previous studies (Willett et al., 2014, Zhou et al.,
2022b).

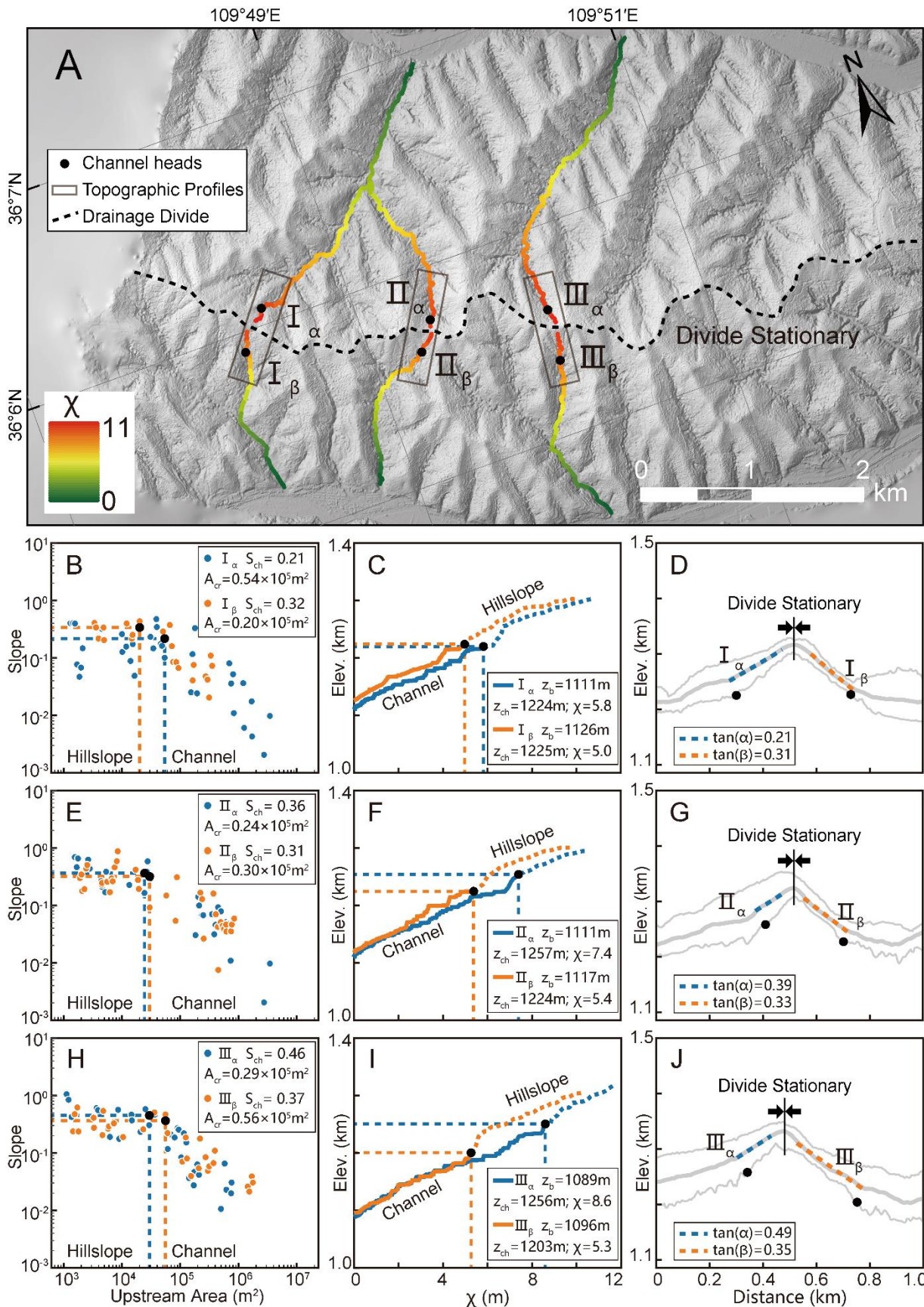


**Figure 5**. Analytical results of the Yingwang Shan in the Loess Plateau. (**A**) High-

resolution hill-shade map (0.84 m spatial resolution). The black dotted line shows the
location of the main drainage divide. Colored lines show the three pairs of selected
channels used for analysis. The black dots represent the channel heads. Black
rectangles show the location of the cross-divide topography swath profiles. (**B, E, H**)
Slope-area plots of the three pairs of selected channels. The blue and orange dots are
the data of the north ($\alpha$) and south ($\beta$) sides of the drainage divide respectively. The
black dots represent the channel heads. (**C, F, I**) $\chi$-plots of the selected channels. The
blue and orange lines are the $\chi$-plots of the north ($\alpha$) and south ($\beta$) sides of the
drainage divide respectively. The black dots represent the channel heads. (**D, G, J**)
The cross-divide topography swath profiles. The locations of the swath profiles are in
Panel A.

**Table 1**. Channel parameters and migration rates of drainage divides in two field cases.

| Natural Cases | No. | $A_{cr}$ ($\times10^5\text{m}^2$) | $S_{ch}$ | $z_b$ (m) | $z_{ch}$ (m) | $\chi$ | $\tan\alpha$ | $\tan\beta$ | $\Delta U_{ch}$ (mm/yr) | $D_{mr}$ (mm/yr) (Channel-head-point method) | $D_{mr}$ (mm/yr) (Channel-head-segment method) |
|---|---|---|---|---|---|---|---|---|---|---|---|
| Wutai Shan | Fig. 4 I$\alpha$ | 1.75 | 0.16 | 1631 | 1792 | 6.4 | 0.14 | 0.66 | ~ 0.008 | -0.21±0.10 | -0.26±0.12 |
| | Fig. 4 I$\beta$ | 0.26 | 0.63 | 1347 | 1723 | 6.6 | | | | | |
| | Fig. 4 II$\alpha$ | 0.79 | 0.23 | 1630 | 1815 | 5.4 | 0.24 | 0.70 | ~ 0.008 | -0.23±0.11 | -0.27±0.13 |
| | Fig. 4 II$\beta$ | 0.30 | 0.67 | 1351 | 1809 | 6.1 | | | | | |
| | Fig. 4 III$\alpha$ | 0.67 | 0.29 | 1633 | 1860 | 5.0 | 0.28 | 0.65 | ~ 0.008 | -0.21±0.10 | -0.22±0.10 |
| | Fig. 4 III$\beta$ | 0.39 | 0.63 | 1352 | 1875 | 6.9 | | | | | |
| Yingwang Shan | Fig. 5 I$\alpha$ | 0.54 | 0.21 | 1111 | 1224 | 5.8 | 0.21 | 0.31 | 0 | ~ 0.03 | ~ -0.01 |
| | Fig. 5 I$\beta$ | 0.20 | 0.32 | 1126 | 1225 | 5.0 | | | | | |
| | Fig. 5 II$\alpha$ | 0.24 | 0.36 | 1111 | 1257 | 7.4 | 0.39 | 0.33 | 0 | ~ 0.02 | ~ -0.01 |
| | Fig. 5 II$\beta$ | 0.30 | 0.31 | 1117 | 1224 | 5.4 | | | | | |
| | Fig. 5 III$\alpha$ | 0.29 | 0.46 | 1089 | 1256 | 8.6 | 0.49 | 0.35 | 0 | ~ 0.02 | ~ -0.01 |
| | Fig. 5 III$\beta$ | 0.56 | 0.37 | 1096 | 1203 | 5.3 | | | | | |


## 4. Discussion

### 4.1 Location of channel heads

Willett et al. (2014) pioneered the use of cross-divide χ contrast to gauge the horizontal motion of drainage divides. According to their method, drainage divides are predicted to move toward the side with a higher χ value to achieve geomorphic equilibrium. However, in a region with spatially variable uplift rates, lithology, or precipitation, χ contrast may fail to reflect the drainage-divide migration (Willett et al., 2014; Whipple et al., 2017; Forte and Whipple, 2018; Wu et al., 2022; Zhou and Tan, 2023). In a tectonically active area, the cross-divide χ contrast can only be used in a small area where rock type, precipitation, and uplift rate are nearly uniform (Willett et al., 2014). Combining the advantages of the χ and Gilbert metrics methods, Zhou et al. (2022a) proposed to use the χ contrast with a high base level to calculate the $k_{sn}$ values at the channel heads on both sides of a drainage divide, and quantified the migration rate of drainage divides at the eastern margin of Tibet.

To reduce the cross-divide difference in uplift rate, precipitation, and rock strength, the Gilbert metrics or χ-comparison method in Zhou et al. (2022a) should compare the parameters of points (slope, relief, elevation, and $k_{sn}$) on both sides of the divide as closely as possible. As the hillslope area (above the channel head) does not follow Eq. 1 (Stock and Dietrich, 2006; Stark, 2010; Braun et al., 2018; Dahlquist et al., 2018), the channel heads are the closest point to the divide, following Eq. 1. Channel heads, therefore, are suitable for measuring the drainage-divide stability with

parameters of the upstream drainage area and channel gradient (Forte and Whipple,
2018; Zhou et al., 2022a). However, limited by the resolution of DEM, the location of
the channel heads cannot always be accurately identified. The channel head
parameters for calculating the migration rates are usually based on empirical values
(both sides are the same value) in previous studies (e.g., $A_{cr} = 10^5$ m$^2$ in Zhou et al.
(2022a)), which may induce uncertainties.

In this study, we advocate the use of high-resolution DEM to determine a more

accurate position and related parameters of the channel head. The use of UAVs to
obtain the local DEM has become highly efficient. We advance the theory to calculate
the drainage-divide migration rate based on the measured channel-head parameters.
With the help of the aerial photography of UAVs and the SfM techniques, it is
possible to obtain the high-resolution topography data of drainage divides (Figs. 4A &
5A) and get the required parameters through topography analysis. The key parameters
includes the exact locations (usually have different $A_{cr}$ across the divides) and the
gradients of the channel heads ($S_{cr}$), which could improve the quantitative research on
the drainage-divide migration. Furthermore, the method provides a new avenue to
combine with catchment-wide [10]Be erosion rate or low-temperature
thermochronology data to calculate the migration rate, which has great potential for
application in places where some variables are hard to be constrained.

## 4.2 Cross-divide difference in the uplift rate of the channel heads

Although the channel heads across the divide are very close on the spatial scale of an orogenic belt, differential uplift between the channel heads ($\Delta U_{ch}$) could still exist, especially in a tilting horst, such as the Wutai Shan. The cross-divide difference in uplift rate could impact the calculation of the migration rate of drainage divides (Zhou et al., 2022a).

In this study, we quantify the influence of the cross-divide difference in rock uplift rate ($\Delta U_{ch}$) on the calculation of the migration rate of drainage divides at the Wutai Shan, benefiting from the available tectonic and chronological research (Clinkscales et al., 2020) and the newly obtained high-resolution topographic data. In the Wutai Shan horst, $\Delta U_{ch}$ across the drainage divide is ~0.008 mm/yr. We estimate the influence of $\Delta U_{ch}$ on the drainage-divide migration rate in this case study, which can reduce the error theoretically. If $\Delta U_{ch}$ is ignored, the drainage-divide migration rate would decrease by ~4% in the Wutai Shan case. Although ~4% seems to be negligible, such a ratio will increase if the mountain belt is narrower, the tilting uplift is stronger, or the divide is closer to the steady state (i.e., the migration rate is lower) (Whipple et al., 2017; Ye et al., 2022). In other words, the differential uplift may play a significant influence on the measurement of drainage-divide stability in some situations. If we consider an extreme example where the main drainage divide of a tilting mountain range (relatively narrow in width) is at a steady state, the gradient, relief, and elevation of the channel heads (collectively called "Gilbert metrics") (Forte and Whipple, 2018) will show a systematic cross-divide difference in theory. In this

case, the drainage divide would be considered unstable if $\Delta U_{ch}$ were neglected.
Therefore, this study highlights that $\Delta U_{ch}$ should be taken into account, either in a
qualitative or a quantitative evaluation of the stability of drainage divides using the
parameters on the channel heads.

**4.3 Limitations and uncertainties**
This study develops the method to calculate the drainage-divide migration rate
based on the measured channel-head parameters. However, uncertainties still exist
because of the limitations of this technique. First, we assume the erosion coefficient
($K$) is the same on both sides of a drainage divide in the derivation of the equations. If
there are differences in rock erodibility or precipitation across the divide, uncertainties
should exist in the results. Second, the calculation of migration rate is based on the
erosion rates at the channel area in this study. However, the occurrence of drainage-
divide migration is directly driven by the differential erosion of the hillslope area
across the divide, mainly via the processes including landslide, collapse, and diffusion
(Stock and Dietrich, 2006; Stark, 2010; Braun et al., 2018; Dahlquist et al., 2018).
Such discontinuous processes in the hillslope area make it challenging to constrain
erosion rates over such short timescales. Over a relatively longer period (i.e., spanning
multiple seismic and climatic cycles), the erosion rate at the channel head area in this
study can be comparable with that at the hillslope area (Hurst et al., 2012; Godard et
al., 2020).
The accuracy of the data and parameters can also impact the reliability of the
results. First, we use the uniform values of $n = 1$ and $m/n = 0.45$ in the two natural
cases to calculate the migration rate, because it is the best choice to align tributaries
with the main stem on the χ-plots in a drainage basin at the northern Wutai Shan (Fig.
6) (Perron and Royden, 2012). If the actual values deviate from the assumption, errors
would be introduced into the results. For this reason, we have added the cases of $m/n$
= 0.35 and 0.55 in Supplementary Materials. Further estimation of these values
(Mudd et al., 2018) could improve the accuracy of the results. Second, in the case of
the Wutai Shan, we refer to the geological and low-temperature thermochronology
studies and assume a 0.50±0.25 mm/yr erosion rate at the northern margin of the
Wutai Shan (i.e., the footwall of the North Wutai Shan fault). Combining with the
present-day $k_{sn}$, we calculate the erosion coefficient ($K$) and derive the migration rates
of the drainage divide. If the present-day erosion rate deviates from the assumption,
errors would be inevitable in the results. Moreover, the horizontal and vertical errors
of the DEM data, as well as the calculation errors in slope, upstream area and channel
steepness can also affect the reliability of the results. In the case study of the
Yingwang Shan, the lush vegetation may bring errors to the DEM data based on the
SfM technology. The application of airborne light detection and ranging (LiDAR)
technology may help reduce this error. Future studies should take these challenges
into account and overcome them.

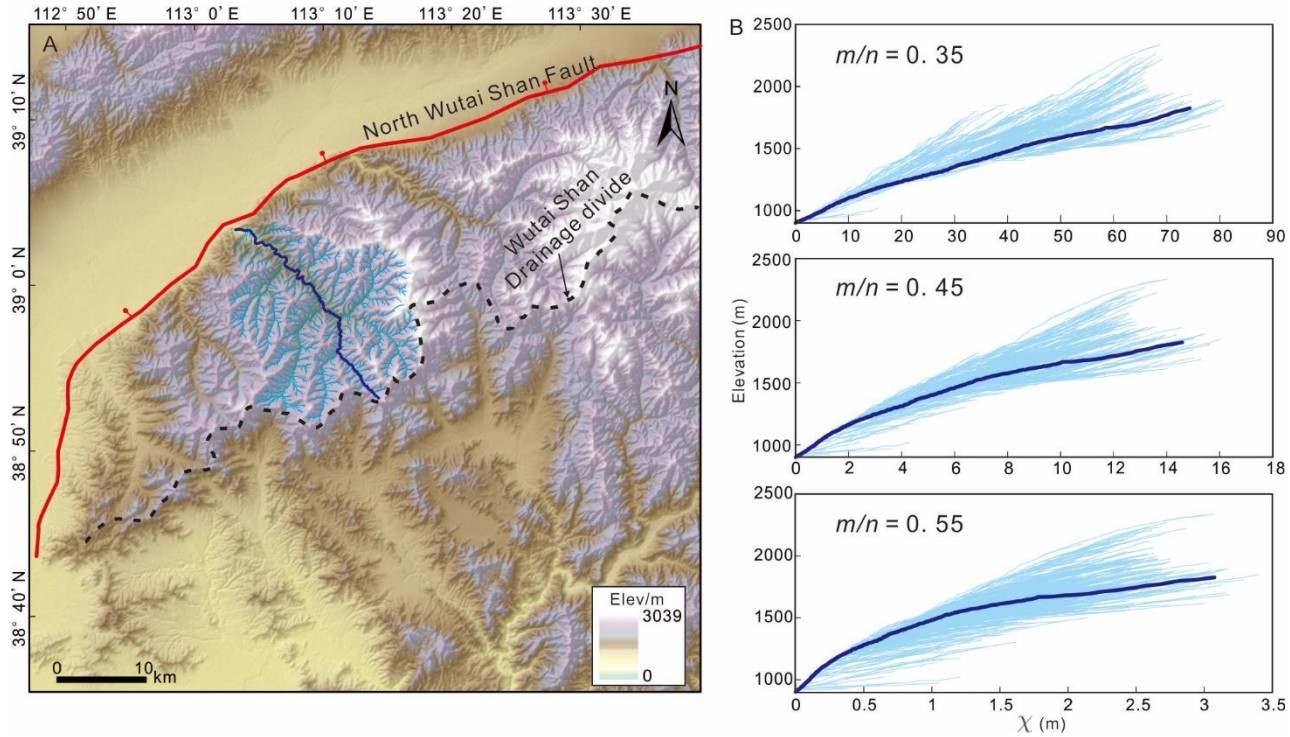

**Figure 6**. (A) Drainage basin in the northern Wutai Shan. (B) $\chi$-plots of channel profiles in the drainage basin, using $A_0 = 1$ m$^2$ and $m/n = 0.35$, $0.45$, and $0.55$. The $\chi$-plots show the best choice of $m/n$ is 0.45, because the tributaries have systematically higher ($m/n = 0.35$) or lower ($m/n = 0.55$) elevations than the main stem for other values of $m/n$ (excluding the channels in the headwaters).

## 5. Conclusions

We have developed a new method (called the "channel-head-point method") to calculate the migration rate of drainage divides based on channel-head parameters. We have also improved the previously proposed "channel-head-segment method" (Zhou et al., 2022a) to adapt the theory to areas where the parameters of channel-heads can be accurately determined.

Using the new methods and high-resolution topographic data, we determined the
exact locations of the channel heads on both sides of the drainage divide and
quantified the drainage-divide migration rates in two natural cases in North China:
Wutai Shan in the Shanxi Rift, and Yingwang Shan in the Loess Plateau. The
migration rates of the study sites in the Wutai Shan are 0.21-0.27 mm/yr
(northwestward). The rates are close to zero in the Yingwang Shan.

Based on the locations of the channel heads and the uplift gradient of the Wutai

Shan, we calculated the cross-divide difference in the uplift rate at the channel heads
($\Delta U_{ch}$), which is taken into account in the calculation of the drainage-divide migration
rate for the first time. If $\Delta U_{ch}$ is overlooked, the drainage-divide migration rate of the
study sites in the Wutai Shan will be underestimated by ~4%. Our study highlights
that $\Delta U_{ch}$ should be considered in the assessment of drainage divide stability based on
the cross-divide difference in channel-head parameters.

**Data availability.** The analysis of data is based on the Matlab toolbox TAK (Forte
and Whipple, 2019) and TopoToolbox (Schwanghart and Scherler, 2014). The
topography data (ALOS DEM) is downloaded from the Alaska Satellite Facility
(ASF) Data Search (https://search.asf.alaska.edu/). The high-resolution DEM of the
two study areas, the Wutai Shan and the Yingwang Shan, can be downloaded from
OpenTopography (https://doi.org/10.5069/G98C9TGT).
**Acknowledgements.** We would like to thank the Editor Simon Mudd, the
Reviewer Thomas Bernard, and an anonymous reviewer whose suggestions have

greatly improved the paper.

**Financial support.** This study is supported by the CAS Pioneer Hundred Talents Program (E2K2010010) and the Fundamental Research Funds for the State Key Laboratory of Earthquake Dynamics (LED2021A02).

**Competing interests.** The authors declare that they have no conflict of interest.

**Author contributions.** XT and CZ contributed to the design of the research scheme. CZ performed the geomorphic analyses. CZ, XT, and FS carried out field data collection. CZ, XT, YL, and FS contributed to the text and reviewed the paper.

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
