# Peer review of "Quantifying the migration rate of drainage divides from"

_EGUsphere, 2023_

## Author Comment (AC1)

**Response to Reviewers**

**1. Response to Thomas Bernard:**

**Thomas Bernard:**

Zhou et al. present in their manuscript a framework to determine the migration rate of drainage divide and apply the methods to two study cases in the Wutai Shan and Loess Plateau. The authors developed new methods of drainage divide migration rate estimation using high-topographic data based on channel-head parameters, values of channel segments, and erosion rate parameters. They argue that by determining the exact location of the channel heads, the migration rate of drainage divide can be accurately calculated. They determined drainage divide migration rate of about 0.10-0.13 mm.yr-1 and closed to zero for the Wutai Shan and Loess Plateau study cases. Finally, they suggest that the difference in uplift on both sides of the drainage divide have to be considered in order to calculate drainage divide migration rate with this method.

The topic of drainage reorganisation by drainage divide migration or river capture is receiving increasing attention these years. Therefore, this contribution, which determined the rate at which drainage divide migrate, is timely and should be of interest to the EGUsphere journal. This study also presents a nice follow-up of drainage divide migration estimation to the study by Zhou et al., 2022 (although I find the two manuscripts really close). The manuscript is interesting and overall well-written. The provided model, methods and equations in the manuscript are sound and well-used in the study area cases. There are a few awkward sentences that I have indicated below. Finally, I found the figures well realized and easy to understand.

My main comments/concerns regarding this manuscript are the use of strong assumptions made in the calculation of the drainage divide rate migration which have not been mentioned in the main text. The first one is the use of erosion rate estimate

from low-temperature thermochronology data, which cannot correspond to a modern rate for the Wutai Shan study case. The second is the systematic use of a "standard" concavity index for both study cases. I feel like these assumptions have to be addressed. Since the main message of the study is to quantify migration rate from high-resolution topographic data, estimation of the equation parameters needs to follow the same logic. More specific comments tied to line number:

**RESPONSE to the general comments:**

1. We agree that the erosion rate estimated from low-temperature thermochronology data may not represent the present-day situation. Therefore, we also refer to the results of geological studies and added the error range to cover this uncertainty: "The geological study shows that the Quaternary throw rates of the Northern Wutai Shan fault are 0.8-1.6 mm/a (Middleton et al., 2017). The low-temperature thermochronology study shows that the time-averaged long-term throw rates in the late Cenozoic is about 0.25 mm/yr, and there is an accelerated activity in the Wutai Shan area (Clinkscales et al., 2020). According to the above geological and low-temperature thermochronology studies, we assume a 0.5±0.25 mm/yr uplift/erosion rate in the northern margin of the Wutai Shan (at the footwall of the Northern Wutai Shan fault).". The error of the erosion rate has also been calculated into the migration rates (Table 1). We have also added a "Limitations and uncertainties" section in the manuscript to discuss the limitations of the methods.

2. We also agree that using the "standard" concavity index for the study cases may bring errors in the results. For this reason, we have added the results of m/n = 0.35 and 0.55 as a reference for different situations: "The calculation of the erosion coefficient ($K$) is using the erosion rate in the north Wutai Shan (0.5±0.25 mm/yr), combing with its $k_{sn}$ values of 20 $m^{0.7}$, 80 $m^{0.9}$, and 350 $m^{1.1}$ at $m = 0.35$, 0.45 and 0.55, respectively ($k_{sn}$ results are in Fig. 2 and Fig. S4). The calculation of the drainage divide's migration rates is based on the channel-head-point method (Eq. 4)". We have also discussed it in the "Limitations and uncertainties" section that further

estimation of the m/n value (Mudd et al., 2018) in the study cases could improve the accuracy of the results.

**Table 1.** Channel parameters and the migration rates of the drainage divides in the two field cases.

| Natural Cases | No. | $A_{cr}$ (×10$^5$m$^2$) | $S_{ch}$ | $z_b$ (m) | $z_{ch}$ (m) | $\chi$ | tanα | tanβ | $\Delta U_{ch}$ (mm/yr) | $D_{mr}$ (mm/yr) (Channel-head-point method) | $D_{mr}$ (mm/yr) (Channel-head-segment method) |
|---|---|---|---|---|---|---|---|---|---|---|---|
| Wutai Shan | Fig. 4 I$_\alpha$ | 1.75 | 0.16 | 1631 | 1792 | 6.4 | 0.14 | 0.66 | ~ 0.008 | -0.21±0.10 | -0.26±0.12 |
| | Fig. 4 I$_\beta$ | 0.26 | 0.63 | 1347 | 1723 | 6.6 | | | | | |
| | Fig. 4 II$_\alpha$ | 0.79 | 0.23 | 1630 | 1815 | 5.4 | 0.24 | 0.70 | ~ 0.008 | -0.23±0.11 | -0.27±0.13 |
| | Fig. 4 II$_\beta$ | 0.30 | 0.67 | 1351 | 1809 | 6.1 | | | | | |
| | Fig. 4 III$_\alpha$ | 0.67 | 0.29 | 1633 | 1860 | 5.0 | 0.28 | 0.65 | ~ 0.008 | -0.21±0.10 | -0.22±0.10 |
| | Fig. 4 III$_\beta$ | 0.39 | 0.63 | 1352 | 1875 | 6.9 | | | | | |
| Yingwang Shan | Fig. 5 I$_\alpha$ | 0.54 | 0.21 | 1111 | 1224 | 5.8 | 0.21 | 0.31 | 0 | ~ 0.03 | ~ -0.01 |
| | Fig. 5 I$_\beta$ | 0.20 | 0.32 | 1126 | 1225 | 5.0 | | | | | |
| | Fig. 5 II$_\alpha$ | 0.24 | 0.36 | 1111 | 1257 | 7.4 | 0.39 | 0.33 | 0 | ~ 0.02 | ~ -0.01 |
| | Fig. 5 II$_\beta$ | 0.30 | 0.31 | 1117 | 1224 | 5.4 | | | | | |
| | Fig. 5 III$_\alpha$ | 0.29 | 0.46 | 1089 | 1256 | 8.6 | 0.49 | 0.35 | 0 | ~ 0.02 | ~ -0.01 |
| | Fig. 5 III$_\beta$ | 0.56 | 0.37 | 1096 | 1203 | 5.3 | | | | | |

[Figure]

**Figure S4**. Normalized channel steepness ($k_{sn}$) distribution in the Wutai Shan. (**A**) The $k_{sn}$ distribution at $m = 0.35$ and $n = 1$. (**B**) The $k_{sn}$ distribution at $m = 0.55$ and $n = 1$. The black dashed line shows the location of the main drainage divide. Red lines show the main active faults. The black straight lines show the location of the profiles G-G' and H-H' and the gray rectangles show the area of the swath profiles in Panel C&D. The topography data (ALOS DEM, 12.5 m resolution) is downloaded from the Alaska Satellite Facility (ASF) Data Search (https://search.asf.alaska.edu/). The $k_{sn}$ is calculated using TopoToolbox (Schwanghart and Scherler, 2014), and the interpolation uses the Kriging method on ArcMap. (**C**) The $k_{sn}$ swath profile along G-G' in Panel A. (**D**) The $k_{sn}$ swath profile along H-H' in panel B. The swath profiles are extracted using TopoToolbox (Schwanghart and Scherler, 2014). The red dashed lines show the location of the main active faults, and the black arrow shows the location of the main drainage divide. Both swath profiles are 20 km wide (10 km on each side). The extent of the swath profiles is represented by the grey boxes in Panel A&B.

**Table S1**. The comparison of drainage divide's migration rates at $m = 0.35$, $0.45$ and $0.55$ in the Wutai Shan area.

| No. | $A_{cr}$ ($\times 10^5 m^2$) | $S_{CH}$ | K ($\times 10^{-6} m^{0.3} yr^{-1}$) (n=1; m=0.35) | K ($\times 10^{-6} m^{0.1} yr^{-1}$) (n=1; m=0.45) | K ($\times 10^{-6} m^{-0.1} yr^{-1}$) (n=1; m=0.55) | $\tan\alpha$ | $\tan\beta$ | Rate (mm/yr) (n=1; m=0.35) | Rate (mm/yr) (n=1; m=0.45) | Rate (mm/yr) (n=1; m=0.55) |
|---|---|---|---|---|---|---|---|---|---|---|
| Fig. 4 $I_\alpha$ | 1.75 | 0.16 | 25.00±12.50 | 6.25±3.13 | 1.43±0.72 | 0.14 | 0.66 | -0.37±0.18 | -0.21±0.10 | -0.10±0.05 |
| Fig. 4 $I_\beta$ | 0.26 | 0.63 | 25.00±12.50 | 6.25±3.13 | 1.43±0.72 | | | | | |
| Fig. 4 $II_\alpha$ | 0.79 | 0.23 | 25.00±12.50 | 6.25±3.13 | 1.43±0.72 | 0.24 | 0.70 | -0.36±0.17 | -0.23±0.11 | -0.14±0.06 |
| Fig. 4 $II_\beta$ | 0.30 | 0.67 | 25.00±12.50 | 6.25±3.13 | 1.43±0.72 | | | | | |
| Fig. 4 $III_\alpha$ | 0.67 | 0.29 | 25.00±12.50 | 6.25±3.13 | 1.43±0.72 | 0.28 | 0.65 | -0.31±0.15 | -0.21±0.10 | -0.13±0.06 |
| Fig. 4 $III_\beta$ | 0.39 | 0.63 | 25.00±12.50 | 6.25±3.13 | 1.43±0.72 | | | | | |

**Thomas Bernard:**

Line 37-39: "The evolution of topography is fundamentally coupled with changes in drainage systems, including river's vertical and lateral movements". Can you be more

precise about vertical and lateral movements? Maybe add river capture as another important process for drainage changes.

**RESPONSE:**

We have changed the sentence: "The evolution of unglaciated terrestrial terrains is fundamentally coupled with changes in drainage systems, which are river's vertical (changes in river long profile) and lateral movements (drainage divide migration and river captures)".

**Thomas Bernard:**

Line 70-71: "For example, Willett et al. (2014) developed the χ method to map the dynamic state of river basins". I don't think Willet et al (2014) developed the χ method. Please change the term "developed" by "applied" or cite "Royden et al., 2000; Perron and Royden, 2012" instead.

**RESPONSE:**

We have changed the term "developed" by "applied".

**Thomas Bernard:**

Line 79-81: "Zhou et al. (2022a) developed a technique to calculate the migration rate through the cross-divide χ ratio of high base-level channel segments". This statement is in contradiction with line 78 "No rates have been obtained".

**RESPONSE:**

We have deleted the sentence "No rates have been obtained." to avoid misunderstanding.

**Thomas Bernard:**

Line 89-90: "to obtain the high-resolution topography data of these two areas". Can you precise the resolution of your topography data here or somewhere in the manuscript.

**RESPONSE:**

We have added the resolution information in the manuscript: "to obtain the high-resolution DEM data of these two areas (0.67 m and 0.84 m spatial resolution in the Wutai Shan and the Yingwang Shan respectively)".

**Thomas Bernard:**

Line 98: "Moreover, benefiting from the detailed tectonic research". Please reformulate or precise what is this "tectonic research".

**RESPONSE:**

We have changed the sentence: "Combining with the geological and low-temperature thermochronology studies (Middleton et al., 2017; Clinkscales et al., 2020)".

**Thomas Bernard:**

Line 127-128: "A large erosion coefficient also creates a high channel-head erosion rate". This sentence is unclear. What does the term "erosion coefficient" refer to?

**RESPONSE:**

We mean the parameter "$K$" in Eq. 1. We have added the symbol, "($K$)", in the manuscript.

**Thomas Bernard:**

Equation 4: This equation is correct only if the erodibility is the same on both side of the drainage divide. Correct the equation or precise this assumption in the text. This is

important, especially, since you demonstrate in your Figure 1 that the erodibility affect the channel-head erosion rate.

**RESPONSE:**

We have clarified this assumption in the text: "Assuming the erosion coefficient (K) is the same on both sides of a drainage divide, Eqs. 2 and 3 allow to derive the equation of the drainage divide's migration rate according to the parameters at the channel-head points".

**Thomas Bernard:**

Line 184-185: integral function of channels' upstream area (A) to horizontal distance (x) (Perron and Royden, 2012; Willet et al., 2014)". Replace the reference "Willet et al., 2014" by "Royden et al., 2000".

**RESPONSE:**

We have replaced the reference "Willet et al., 2014" by "Royden et al., 2000".

**Thomas Bernard:**

Line 241: "If we assume the rock uplift rate decreases linearly from 0.25 to 0 mm/yr from northwest to southeast of the Wutai Shan horst". If I understood correctly the 0.25 mm/yr rate come from the low-temperature thermochronology study. Even if this rate is predicted for the late Cenozoic, it cannot reflect the modern erosion rate (the method is not sensible) needed to accurately calculate the modern drainage divide. It assumes that the erosion rate stayed constant. This assumption is not reported in this paragraph or the discussion.

**RESPONSE:**

We have added the discussion in the "Limitations and uncertainties" section to report this error: "Secondly, in the case study of the Wutai Shan, we refer to the geological

and low-temperature thermochronology studies, assuming a $0.5 \pm 0.25$ mm/yr erosion rate at the northern margin of the Wutai Shan (i.e., the footwall of the North Wutai Shan fault). Combining with the present-day $k_{sn}$, we calculate the erosion coefficient ($K$) and derive the migration rates of the drainage divide. If the present-day erosion rate deviates from the assumption, the error would exist in the results". The calculation and the results (Table 1) are in RESPONSE to the general comments.

**Thomas Bernard:**

Line 246: "We assume n = 1 and m = 0.45 in the calculation following previous studies (Wobus et al., 2006; DiBiase et al., 2010; Perron and Royden, 2012; Wang et al., 2021)". This is a reference concavity of 0.45 that you are using for your calculation. How does this concavity really reflect the geomorphology of your study area? There is available framework to calculate the channel concavity of your catchment, check Mudd et al., 2018. This is even more important since you can have different concavities on both side of your drainage divide and strongly affect your migration rate calculations based on your equations. Line 313: "We also assume n = 1 and m = 0.45 in the calculation". Same as previous comment.

**RESPONSE:**

We have added the discussion in the "Errors and uncertainties" section: "we use the typical values of n = 1 and m/n = 0.45 in the two natural cases to calculate the migration rate. If the actual values largely deviate from the assumption, the error would exist in the results. For this reason, we have added the cases of m/n = 0.35 and 0.55 in the in the supplementary materials. Further estimation of these values (Mudd et al., 2018) could improve the accuracy of the results". The calculation and the results (Figs. S4 and Table S1) are in the RESPONSE to the general comments.

**Thomas Bernard:**

Line 294: "An unnamed mountain range in the Loess Plateau". The term "unnamed" sounds strange. I suggest to just remove it or find another solution.

**RESPONSE:**

We have named the mountain as "Yingwang Shan".

**Thomas Bernard:**

Figure 1: It might be better to directly indicate the value of in the different panel instead of the ratio. You can also precise the erodibility for each panel in the figure description since you did it for the slope and area coefficients.

**RESPONSE:**

We have changed the Fig. 1 (Fig. S1 in the new version) accordingly and clarified the erodibility in the figure description:

**Thomas Bernard:**

Figure 2: Precise the meaning of the white rectangle in the description like for the back rectangles.

**RESPONSE:**

We have clarified it in the figure description: "White rectangles show the location of panel A".

**Thomas Bernard:**

Figure 4: The text of Eq. 4 and 8 could be coloured in black and blue respectively in order to directly identified which arrows correspond to which equations and rates. Or you could add small arrow legends on the left of the text.

**RESPONSE:**

We have changed the colour of the text to black and blue respectively:

**Thomas Bernard:**

Technical corrections:

Line 60-62: "However, these techniques are usually based on samples collected from an outlet that is several kilometers away from the drainage divide and thus may not represent the erosion rates close to the drainage divide". Change "an outlet" to "a catchment outlet".

**RESPONSE:**

We have changed "an outlet" to "a catchment outlet" in the manuscript.

**Thomas Bernard:**

Line 65-67: "Hence, it would be ideal to find an accessible and efficient method that can be applied to the entire landscape and cross-checked to make full use of the 10Be-derived erosion rates". The term "cross-checked" is unclear in this sentence.

**RESPONSE:**

We have deleted the term "cross-checked" to avoid misunderstanding: "Hence, it would be ideal to find an accessible and efficient method that can be applied to the entire landscape and make full use of the 10Be-derived erosion rates".

**Thomas Bernard:**

Line 68-69: "The advancement of remote sensing technology has promoted the development of geomorphic analysis theory". Remove the term "theory".

**RESPONSE:**

We have removed the term "theory" and changed this sentence: "The advancement of the digital elevation model (DEM) has promoted the development of geomorphic analysis,…".

**Thomas Bernard:**

Line 77: "No rates have been obtained". This short sentence might be combined with another one.

**RESPONSE:**

We have deleted this short sentence.

**Thomas Bernard:**

Line 86: "an unnamed mountain range in the Loess plateau". Remove the term "unnamed".

**RESPONSE:**

We have removed it and changed the sentence to "the Yingwang Shan in the Loess Plateau".

**Thomas Bernard:**

Line 128-129: "The results indicate that the side with a higher Acr or Sch can have a higher erosion rate than the other side of the drainage divide". Move the terms "of the drainage divide" to the first occurrence of the term "side".

**RESPONSE:**

We have moved it: "Eq. 2 indicates that the side of a drainage divide with a higher $A_{cr}$ or $S_{ch}$ can have a higher erosion rate than the other side".

**Thomas Bernard:**

Line 134-136: "when one uses the cross-divide erosion rates … one should also consider the influence of differential uplift rates". I suggest to remove the terms "when one" and "one should" and reformulate the sentence.

**RESPONSE:**

We have changed the sentence to "Furthermore, the differential uplift should also be considered when using the cross-divide erosion rates …".

**Thomas Bernard:**

Line 144: "Combining Eqs. 2 and 3, one can derive the equation". Same as previous comment, replace "one can" by "allow to" for example.

**RESPONSE:**

We have replaced "one can" by "allow to": "Eqs. 2 and 3 allow to derive the equation of the drainage divide's migration rate …".

**Thomas Bernard:**

Line 216-217: Correct to "The bedrock of the Wutai Shan area consists mainly of metamorphic and igneous basement rocks".

**RESPONSE:**

We have corrected this sentence accordingly.

**Thomas Bernard:**

Line 263: Change to "Figure 3. Topography (A) and normalized channel steepness (ksn) (B) distribution".

**RESPONSE:**

We have changed it accordingly.

**Thomas Bernard:**

Line 264-265: The black dashed curve shows the location of the main drainage divide". Change the term "curve" to "line".

**RESPONSE:**

We have changed it: "The black dashed line shows the location of the main drainage divide".

**Thomas Bernard:**

Line 272-273: "The topography swath profile along E-E' in Fig. 3A. (D) The ksn swath profile along F-F' in Fig. 3B". Change the terms "Fig. 3" to "panel".

**RESPONSE:**

We have changed the term "Fig. 3" to "panel".

**Thomas Bernard:**

Line 273-274: "The red dotted line shows". Correct to "The red dotted lines show".

**RESPONSE:**

We have changed it accordingly.

**Thomas Bernard:**

Line 279-292: Replace the terms "curve" by "line".

**RESPONSE:**

We have replaced it.

**Thomas Bernard:**

Line 292: Correct to "the channel-head (Eq. 4) and channel-head-segment (Eq. 8) methods respectively".

**RESPONSE:**

We have corrected it accordingly.

**Thomas Bernard:**

Line 296: "(Yin, 2010 Su et al., 2021)". There is a missing coma between the two references.

**RESPONSE:**

We have added the coma here.

**Thomas Bernard:**

Line 306: "the slope-area plots (Figs. 5 B, E, H) and the $\chi$ values". Change "and" by a coma.

**RESPONSE:**

We have changed it.

**Thomas Bernard:**

Line 324-334: Replace the terms "curve" by "line".

**RESPONSE:**

We have replaced it.

**Thomas Bernard:**

Line 346-347: "In the tectonically active area". Change "the" by "a".

**RESPONSE:**

We have changed it.

**Thomas Bernard:**

Line 350-351: Zhou et al. (2022a) combined the advantages of the $\chi$ and Gilbert metrics methods, proposed to use the $\chi$ contrast with a high base level". This sentence does not sound right. Maybe change "combined" by "by combining".

**RESPONSE:**

We have changed it: "Combining the advantages of the $\chi$ and Gilbert metrics methods, Zhou et al. (2022a) proposed to use the $\chi$ contrast with a high base level…".

**Thomas Bernard:**

Line 361: "(Forte and Whipple, 2018; Zhou et al., 2022a; this study)". Remove "this study" to the references.

**RESPONSE:**

We have removed it.

**Thomas Bernard:**

Line 363-366: "In this study, we advocate the use of high-resolution DEM to determine a more accurate position and related parameters of the channel head, given that the use of UAVs to obtain the local DEM has become highly efficient". I suggest to break this sentence in two (maybe around the coma).

**RESPONSE:**

We have broken the sentence in two: "In this study, we advocate the use of high-resolution DEM to determine a more accurate position and related parameters of the channel head. The use of UAVs to obtain the local DEM has become highly efficient".

**Thomas Bernard:**

Line 368-369: "one can obtain the sub-meter resolution topography data of drainage divides". Replace "one can" by "it is possible to".

**RESPONSE:**

We have replaced it: "it is possible to obtain the high-resolution topography data of drainage divides …".

**Thomas Bernard:**

Line 396-397: Change "Consider an extreme example: when the main drainage divide" by "If we consider an extreme example where the main drainage divide".

**RESPONSE:**

We have changed it accordingly.

**2. Response to Anonymous Referee #2:**

**Reviewer:**

This study addresses a highly pertinent question that lately has garnered significant attention in the scientific community focused on landscape evolution: what are the rates at which drainage divides migrate? The study presents two quantitative methods for measuring migration rates, both of which necessitate precisely identifying channel heads through high-resolution DEMs. The first method leverages the channel head's area and slope, while the second method builds upon an enhanced version of Zhou et al.'s (2022) technique, incorporating χ values, elevation at the channel head, and outlet elevation. The study applies these two methods in two distinct field settings—one impacted by tectonic tilting and the other situated in a tectonically quiescent area. The results reveal a similarity between the outcomes of both methods, suggesting that in the tectonically affected area, the divide is migrating in rates of ~0.1 mm/yr, while in the second setting, the divide is stable.

The foundational concept of the study seems promising, and the paper's structure effectively guides the reader through the core idea. Nevertheless, there are several crucial issues that demand the authors' attention, particularly regarding the methods presented in this manuscript. Hereafter I highlight specific points that warrant further clarification and elaboration.

**Overall comments:**

1. In the introduction, I recommend the following improvements: A) Provide a more in-depth motivation regarding the significance of divide migration, emphasizing its relevance to different studies. B) Distinguish between field-based investigations involving natural cases, such as those using cosmogenic nuclides and thermochronology, and modeling-based approaches, which use χ or Gilbert methods. C) Expand upon the method introduced by Zhou et al. (2022), elucidating how it forms the foundation of your research.

**RESPONSE:**

We have revised the introduction accordingly:

A) We have added these sentences in the introduction to demonstrate the significance of divide migration and its relevance to different studies: "Drainage-divide migration, one form of river lateral movement, may not only carry information on geological and/or climatic disturbance (Su et al., 2020; Zondervan et al., 2020; He et al., 2021; Shi et al., 2021; Zhou et al., 2022a; Zeng and Tan, 2023) but also influence the extraction of tectonic information from channel profiles (Goren et al., 2014; Ma et al., 2020; Jiao et al., 2021). Besides, it also has wide-ranging consequences for topography (Scheingross et al., 2020; Stokes et al., 2022), the sedimentary record (Clift & Blusztajn, 2005; Willett et al., 2018; Deng et al., 2020; Zhao et al., 2021), and biological evolution (Waters et al., 2001; Zemlak et al., 2008; Hoorn et al., 2010; Musher et al., 2022). For this reason, the stability of drainage divides has drawn more and more attention in recent years (e.g., Authemayou et al., 2018; Vacherat et al., 2018; Chen et al., 2021; Shelef and Goren, 2021; Sakashita and Endo, 2023; Bian et al., 2024)."

B) We have distinguished the field-based investigations involving natural cases (e.g., cosmogenic nuclides sampling) and modeling-based approaches (e.g., $\chi$ or Gilbert metrics), and describe them in two separate paragraphs in the introduction.

C) We have added these sentences in the introduction to demonstrate the relationship between this study and Zhou et al. (2022): "Zhou et al. (2022a) developed a technique to calculate the migration rate through the high base-level $\chi$ values on both sides of a drainage divide. The above approaches require channel-head parameters to calculate the migration rate. However, the location of the channel heads sometimes cannot be accurately identified, limited by the resolution of DEM in natural cases. For this reason, empirical values of channel-head parameters are used in the above study, which may induce uncertainties".

**Reviewer:**

2. The terminology "channel head method" and "channel head segment method" might confuse readers. I recommend considering alternative names for one of these methods.

**RESPONSE:**

We have changed the terminology "channel-head method" to "channel-head-point method" to avoid confusion.

**Reviewer:**

3. The channel head used in equations 5-7 defines a point that separates the hillslope from the channel. However, the calculation of slope inherently involves two points. Therefore, it is essential to elucidate how you measured the slope in this specific context, particularly within the channel head, where it distinguishes between areas with differing slopes.

**RESPONSE:**

The $S_{ch}$ in equations 5-7 are measured along the tangent lines of the channel-head points on the channels. In the manuscript, we first plot the river long profiles and locate the channel-head points on the profiles. Then, we calculate the average slopes of the channel segments near the channel-head points (100m-long segments are enough), and use them as the $S_{ch}$ values. We have clarified it in the manuscript.

**Reviewer:**

4. I think it is crucial to provide a clear mathematical definition of parameters χ and ksn which are employed consistently throughout the study. Additionally, an explicit explanation of how Equation 8 was derived would be valuable. I am asking because

ksn pertains to a specific point in the channel while $\chi$ represents an integral over a channel segment.

**RESPONSE:**

A) We have provided a clear definition of $\chi$ and $k_{sn}$ in the manuscript: "$k_{sn}$ is a widely used index (Whipple et al., 1999; Wobus et al., 2006; Hilley and Arrowsmith, 2008; Kirby and Whipple, 2012) that is quantitatively related to $E$ and $K$ ($k_{sn} = \left(\frac{E}{K}\right)^{\frac{1}{n}}$). $\chi$ is an integral function ($\chi = \int_{x_b}^{x} \left(\frac{A_0}{A(x)}\right)^{\frac{m}{n}} dx$) of channels' upstream area ($A$) to horizontal distance ($x$) (Royden et al., 2000; Perron and Royden, 2012), and $A_0$ is an arbitrary scaling area to make the integrand dimensionless."

B) We have also provided the derivation of Equation 8 in the supplementary materials: "According to the detachment-limited stream power model (Howard and Kerby, 1983; Howard, 1994):

$$E = KA^m S^n \tag{1}$$

The channel gradient ($S$) can be written as:

$$S = \left(\frac{E}{K}\right)^{\frac{1}{n}} A^{-\left(\frac{m}{n}\right)} \tag{2}$$

A river's longitudinal elevation $z(x)$ can be expressed by the integration of channel gradient ($S$) in the upstream direction from a base level $x_b$ to an observation point $x$:

$$z(x) = z_b + \int_{x_b}^{x} S(x)\, dx = z_b + \int_{x_b}^{x} \left(\frac{E(x)}{K(x)}\right)^{\frac{1}{n}} A(x)^{-\left(\frac{m}{n}\right)} dx \tag{3}$$

In the case of spatially invariant erosion rate ($E$) and erosion coefficient ($K$), Eq. (3) can be reduced to a simpler form:

$$z(x) = z_b + k_{sn}(A_0)^{-\left(\frac{m}{n}\right)}\chi \tag{4}$$

with

$$k_{sn} = \left(\frac{E}{K}\right)^{\frac{1}{n}} \tag{5}$$

and

$$\chi = \int_{x_b}^{x} \left(\frac{A_0}{A(x)}\right)^{\frac{m}{n}} dx \tag{6}$$

$A_0$ is an arbitrary scaling area, to make the integrand dimensionless. Assuming $A_0 =$ 1m$^2$, the steepness ($k_{sn}$) of channel-head-segment can be written as:

$$k_{sn} = \frac{z_{ch} - z_b}{\chi} \tag{7}$$

According to the Eq. (5), the erosion rate ($E$) can be written as:

$$E = Kk_{sn}{}^n = K\left(\frac{z - z_b}{(A_0)^{-\left(\frac{m}{n}\right)}\chi}\right)^n. \tag{8}$$

According to the equation of drainage-divide migration rate ($D_{mr}$):

$$D_{mr} = \frac{\Delta E_{ch} - \Delta U_{ch}}{tan\alpha + tan\beta} = \frac{E_{ch(\alpha)} - E_{ch(\beta)} - \Delta U_{ch}}{tan\alpha + tan\beta} \tag{9}$$

Combing Eq. (7, 8 & 9), the drainage-divide migration rate ($D_{mr}$) can be written as:

$$D_{mr} = \frac{E_{ch(\alpha)} - E_{ch(\beta)} - \Delta U_{ch}}{tan\alpha + tan\beta} = \frac{Kk_{sn(\alpha)}{}^n - Kk_{sn(\beta)}{}^n - \Delta U_{ch}}{tan\alpha + tan\beta} = \frac{K\left\{\left[\frac{(z_{ch} - z_b)_\alpha}{\chi_\alpha}\right]^n - \left[\frac{(z_{ch} - z_b)_\beta}{\chi_\beta}\right]^n\right\} - \Delta U_{ch}}{tan\alpha + tan\beta}$$

(10)".

**Reviewer:**

5. The calculation of the regional erosion coefficient K in this study requires some further clarification and consideration of several points: A) in the case of the Wutai Shan area, it is assumed that rock erodibility and precipitation are uniform. How is this assumption valid in a landscape with a range of elevation ranging between ~100-3000 m? Consider including isohyet maps and geological maps to strengthen your argument.  B) You mention that the erosion rate you used in the study averages a long time frame (since the late Cenozoic). However, the ksn values, that are also used to calculate the erosion coefficient K, are based on the present conditions of lithology and precipitation. Do you think the ksn values also remained uniform since the late

Cenozoic? Please explain the feasibility of these assumptions in the text. C) It is unclear why kriging interpolation is required over all of the landscape, as Ksn values are relevant only across channels. D) One value of erosion rate was considered for calculating the K of all the Wutai Shan area. However, the erosion rates along a tilting block might vary and probably decrease with distance from the northern Wutai Shan fault.

**RESPONSE:**

A) We have included the precipitation and geological maps in the supplementary materials (See the figures below) to support this assumption.

B) We agree that the erosion rate estimated from low-temperature thermochronology data may not represent the present-day situation. Therefore, we also refer to the results of geological studies and added the error range to cover this uncertainty: "The geological study shows that the Quaternary throw rates of the Northern Wutai Shan fault are 0.8-1.6 mm/a (Middleton et al., 2017). The low-temperature thermochronology study shows that the time-averaged long-term throw rates in the late Cenozoic is about 0.25 mm/yr, and there is an accelerated activity in the Wutai Shan area (Clinkscales et al., 2020). According to the above geological and low-temperature thermochronology studies, we assume a 0.5±0.25 mm/yr uplift/erosion rate in the northern margin of the Wutai Shan (at the footwall of the Northern Wutai Shan fault)". The error of the erosion rate also be calculated into the migration rates (see the revised Table 1 below). We have also added a "Limitations and uncertainties" section in the manuscript to report this error: "Secondly, in the case study of the Wutai Shan, we refer to the geological and low-temperature thermochronology studies, assuming a 0.5±0.25 mm/yr erosion rate at the northern margin of the Wutai Shan (i.e., the footwall of the North Wutai Shan fault). Combining with the present-day ksn, we calculate the erosion coefficient (K) and derive the migration rates of the drainage divide. If the present-day erosion rate deviates from the assumption, the error would exist in the results".

C) Indeed, the ksn values are only for channels, but the interpolation result can show the overall spatial variation trend.

D) Yes, the erosion rates should decrease with distance from the northern Wutai Shan fault, as well as the ksn values. But the K should be constant within the Wutai Shan area due to the similar precipitation and erodibility. Theoretically, the K value could be derived using the known erosion rate and its corresponding ksn of any place within the Wutai Shan area, and the calculated results should be the same.

[Figure]

Figure S2. Geological map of the natural example, Wutai Shan. The map is revised according to Clinkscales et al. (2020). The northern catchment evolves mainly through the undifferentiated Neoarchean-Paleoproterozoic metamorphic and igneous basement rocks (pЄu) and Neoarchean granitoids (Xg), whereas the southern catchment evolves mainly through the Paleoproterozoic Hutuo Group (Xh). There is no significant difference in rock erodibility on both sides of the drainage divide.

[Figure]

Figure S3. Precipitation distribution of the Wutai Shan. There is no significant difference in precipitation on both sides of the drainage divide, based on the precipitation data (1970-2000) from http://worldclim.org.

**Table 1.** Channel parameters and the migration rates of the drainage divides in the two field cases.

| Natural Cases | No. | $A_{cr}$ ($\times 10^5$ m$^2$) | $S_{ch}$ | $z_b$ (m) | $z_{ch}$ (m) | $\chi$ | tanα | tanβ | $\Delta U_{ch}$ (mm/yr) | $D_{mr}$ (mm/yr) (Channel-head-point method) | $D_{mr}$ (mm/yr) (Channel-head-segment method) |
|---|---|---|---|---|---|---|---|---|---|---|---|
| | Fig. 4 I$_\alpha$ | 1.75 | 0.16 | 1631 | 1792 | 6.4 | 0.14 | 0.66 | ~ 0.008 | -0.21 ±0.10 | -0.26 ±0.12 |
| | Fig. 4 I$_\beta$ | 0.26 | 0.63 | 1347 | 1723 | 6.6 | | | | | |
| Wutai Shan | Fig. 4 IIα | 0.79 | 0.23 | 1630 | 1815 | 5.4 | 0.24 | 0.70 | ~ 0.008 | -0.23 ±0.11 | -0.27 ±0.13 |
| | Fig. 4 IIβ | 0.30 | 0.67 | 1351 | 1809 | 6.1 | | | | | |
| | Fig. 4 III$_\alpha$ | 0.67 | 0.29 | 1633 | 1860 | 5.0 | 0.28 | 0.65 | ~ 0.008 | -0.21 ±0.10 | -0.22 ±0.10 |
| | Fig. 4 III$_\beta$ | 0.39 | 0.63 | 1352 | 1875 | 6.9 | | | | | |
| Yingwang Shan | Fig. 5 I$_\alpha$ | 0.54 | 0.21 | 1111 | 1224 | 5.8 | 0.21 | 0.31 | 0 | ~ 0.03 | ~ -0.01 |
| | Fig. 5 I$_\beta$ | 0.20 | 0.32 | 1126 | 1225 | 5.0 | | | | | |
| | Fig. 5 II$_\alpha$ | 0.24 | 0.36 | 1111 | 1257 | 7.4 | 0.39 | 0.33 | 0 | ~ 0.02 | ~ -0.01 |
| | Fig. 5 II$_\beta$ | 0.30 | 0.31 | 1117 | 1224 | 5.4 | | | | | |

| | | | | | | | | | | |
|---|---|---|---|---|---|---|---|---|---|---|
| Fig. 5 III$_\alpha$ | 0.29 | 0.46 | 1089 | 1256 | 8.6 | 0.49 | 0.35 | 0 | ~ 0.02 | ~ -0.01 |
| Fig. 5 III$_\beta$ | 0.56 | 0.37 | 1096 | 1203 | 5.3 | | | | | |

**Reviewer:**

6. Errors and Uncertainties: Including error estimates and uncertainties in the study is crucial. These include horizontal and vertical errors of the DEM data and uncertainties related to slope measurements, erosion rates, and ksn Please explain how these errors propagate into the migration rate calculations and discuss their implications for the study's findings.

**RESPONSE:**

We have added the "Limitations and uncertainties" section in the manuscript and explained the above errors: "Moreover, the horizontal and vertical errors of the DEM data as well as the calculation errors in slope, upstream area and channel steepness can also affect the reliability of the results. In the case study of the Yingwang Shan, the lush vegetation may bring errors to the DEM data based on the SfM technology. The application of airborne light detection and ranging (LiDAR) technology may help reduce this error." The details of the erosion rates' error are in the RESPONSE of Overall comments 5.

**Reviewer:**

7. Accessibility and resolution of the data: Please provide the precise resolution of the DEM and not only a general "sub-meter" description. If possible, include these DEM datasets as supplementary materials or explain any limitations or reasons why they cannot be.

**RESPONSE:**

We have provided the resolution of the DEM in the manuscript (0.67 m and 0.84 m spatial resolution in the Wutai Shan and the Yingwang Shan respectively), and include the DEM datasets as supplementary materials.

**Reviewer:**

8. Maybe evaluate the relevance of Section 4.1 and consider incorporating it into the introduction. Currently, in my opinion, it fits to the introduction, as it provides motivation for the study rather than introducing new information.

**RESPONSE:**

We have revised the introduction according to the overall comments 1 (see the response there) and clarified the motivation of the study. Therefore, we retained Section 4.1 and revised it.

**Reviewer:**

9. Can you identify morphological evidence from the field that supports the inference of drainage divide migration? For instance, wind gaps along the divide or evidence for stream capture? Such evidence can meaningfully support your results.

**RESPONSE:**

Thank you for your suggestion. We have added the evidence for stream capture in the Wutai Shan area (Fig. 3 in the new manuscript): "Morphological evidence also shows a clear northwestward migration of the drainage divide (Fig. 3). The plan and satellite views show several barbed tributaries and a captured area around the Wutai Shan drainage divide, which indicate the tributaries formerly part of the north drainage became part of the south drainage (Fig. 3A&B). The $\chi$-plots analysis shows the south side of the drainage divide has steeper channels, higher $k_{sn}$, and lower $\chi$. The $\chi$-plots

of paired rivers show obvious characteristics of shrinking-expanding and captured-
beheaded rivers (Fig. 3C)."

[Figure]

**Figure 3**. Perspective views and χ map of the drainage divide in the Wutai Shan (The
location is shown in Fig. 2). (A) Perspective views of the capture area and the
channels mapped with ksn. The south side of the drainage divide has steeper channels
and higher ksn than the north side. Red arrows show drainage divide migration
directions. The satellite image is downloaded from Google Earth. (B) The χ Map in
this area with the outlet elevation of 1300 m. The south side of the drainage divide has
lower χ values than the north side. The χ-plots of the rivers in bold lines are shown in
panel C. The topography data (ALOS DEM, 12.5 m resolution) is downloaded from
the Alaska Satellite Facility (ASF) Data Search (https://search.asf.alaska.edu/). (C) χ-
plots of the three paired rivers in panel B. The blue and red curves are the rivers on
the south and north sides respectively. The χ-plot of River 1 is steeper on the south
side, indicating that the river on the south side is the aggressor and is the victim on the

north side. The χ-plots of Rivers 2&3 in the captured area show obvious characteristics of the captured and beheaded rivers. The χ-plot is extracted using TAK (Forte and Whipple, 2019) and TopoToolbox (Schwanghart and Scherler, 2014).

**Reviewer:**

10. I provided some suggestions for edits that can assist in the flow and structure of the manuscript, such as connecting too short sentences e.g. lines 293, repetitive sentences, e.g., lines 54 and 132, or unclear sentences such as 251-252. The sentences have been revised accordingly. See the responses below.

**Line-by-line comments:**

20: "normalized channel steepness"- I don't think this term can be used in the abstract without explanation.

**RESPONSE:**

We have removed this term in the abstract and provided a detailed explanation in the methods section.

**Reviewer:**

22: Maybe give these mountains a name of your own, as you refer to them several times in the manuscript. Same for line 86 and further in the manuscript.

**RESPONSE:**

We have named the mountain "Yingwang Shan" and changed the manuscript accordingly.

**Reviewer:**

23-25: I suggest connecting these two short sentences to one sentence, e.g., "Our results find that the divide in Wutai Shan range is migrating north in a rate of between

0.10 to 0.13 mm/yr, whereas the migration rates at the mountain range in the Loess Plateau are approximately zero."

**RESPONSE:**

We have connected the two sentences to one accordingly.

**Reviewer:**

25-27: How is this demonstrated?

**RESPONSE:**

We have changed this sentence to "This study indicates that the drainage-divide migration state can be determined more accurately using high-resolution topographic data. Furthermore, this study takes the cross-divide differences in uplift rate of channel heads into account in the measurement of drainage-divide migration rate for the first time".

**Reviewer:**

34: "…providing a basis.."

**RESPONSE:**

We have changed it: "The evolution of the Earth's surface is jointly controlled by tectonics, lithology, and climatic conditions (e.g., Molnar and England, 1990; Whipple, 2009; Gallen, 2018; Bernard et al., 2021; Hoskins et al., 2023), providing a basis for reconstructing …".

**Reviewer:**

37: What is the difference between the evolution of earth's surface and the evolution of topography?

**RESPONSE:**

We have changed "the evolution of topography" to "the evolution of unglaciated terrestrial terrains" to make the sentence clearer.

**Reviewer:**

38-39: The sentences imply that there are other changes in drainage systems that are not included in the river's vertical or lateral movements. If yes, what are they?

**RESPONSE:**

We have changed "including" to "which are" to be more precise: "The evolution of unglaciated terrestrial terrains is fundamentally coupled with changes in drainage systems, which are river's vertical (changes in river long profile) and lateral movements (drainage divide migration and river captures) (Whipple, 2001; Clark et al., 2004; Bonnet, 2009; Willett et al., 2014)".

**Reviewer:**

40-43: Maybe write shortly about the findings of all these investigations.

**RESPONSE:**

We have added the findings of all these investigations in the manuscript: "River long profiles have been used to study the earthquake events (e.g., Burbank and Anderson, 2001; Wei et al., 2015) and the spatio-temporal differences of uplift (e.g., Whipple et al., 1999; Kirby et al., 2003; Pritchard et al., 2009; Goren et al., 2014).".

**Reviewer:**

44: Why however? i.e., how does this sentence contrast with the previous one?

**RESPONSE:**

We have deleted "however".

**Reviewer:**

47: I think you must define here what is drainage divide migration.

**RESPONSE:**

We have added the explanation here: "Drainage-divide migration, one form of lateral movement of rivers, may…".

**Reviewer:**

47-51: It is unclear why and how the divide migration carries information about the disturbances and the tectonic information extraction.

**RESPONSE:**

We added the discription on the relationship between the drainage divide migration and the geological and/or climatic disturbance: "Recent studies show that the widespread lateral movement of river basins driven by geological and/or climatic disturbance (Yang et al., 2019; Zondervan et al., 2020; Zhou et al., 2022a; Bian et al., 2024)."

**Reviewer:**

55: The migration is also controlled by the topographic slope in the citations you provided.

**RESPONSE:**

Yes, we have changed the sentence to "Drainage-divide migration is essentially controlled by the cross-divide difference in erosion and topographic slope".

**Reviewer:**

59: In what cases do you use migration velocity, and when do you use migration rate? Same for line 79, 145 and further in the manuscript.

**RESPONSE:**

The migration velocity including the migration rate and direction. The migration rate is calculated through the formulas in the manuscript, and the migration direction is determined through the strike of the drainage divide. This study focused on how to constrain the migration rate. We have replaced the "migration velocity" by "migration rate".

**Reviewer:**

63: Do you mean "..of sample processing"?

**RESPONSE:**

Yes, we have changed it in the manuscript.

**Reviewer:**

65 and 66: Perhaps replace "entire landscape" with "large landscapes" or alike. It is not clear to which entire landscape you are referring to.

**RESPONSE:**

Yes, we have replaced it in the manuscript: "Besides, the high cost of sample processing makes it very challenging to determine the drainage divide's motion by measuring the erosion rates throughout the large landscapes".

**Reviewer:**

68: The term "remote sensing" includes a very wide range of data types. Maybe be more specific.

**RESPONSE:**

We have changed the sentence to make it more specific: "The advancement of the digital elevation model (DEM) has promoted the development of geomorphic analysis …".

**Reviewer:**

69-70: What do you mean by "determine the drainage divide's motion through topography analysis "? determine the past motion (i.e., infer divide migration that occurred in the past)? Or predict future motion?

**RESPONSE:**

We mean the transient motion reflected by the DEM data. We have noted it in the manuscript: "… making it possible to determine the drainage divide's transient motion through topography analysis".

**Reviewer:**

71: See overall comments- you should explain more about the χ parameter and devote an equation to this, especially if you use it in your equations later.

**RESPONSE:**

Yes, we have explained more about the χ parameter and added the equation (see the response of overall comment 4).

**Reviewer:**

73: What do you mean in determine a drainage divide motion? The direction? Rate?

**RESPONSE:**

We mean the migration direction here. We have clarified it in the manuscript.

**Reviewer:**

74: What is the difference between slope angle and the slope mentioned in the previous sentence?

**RESPONSE:**

The "slope" is the tangent value of "slope angle" here.

**Reviewer:**

75-77: The fact these methods are quantitative does not necessarily mean they can provide rates. No need to add the contrast term "although" in the beginning of the sentence.

**RESPONSE:**

We have removed the word "although" from the sentence.

**Reviewer:**

78: Please explain better about this "method". I assume you mean he used numerical simulations. Also, as I wrote in the overall comments, I recommend separating between modelling-based studies and all of the other studies you cite, that are field based.

**RESPONSE:**

Besides the numerical simulations, the study also provides a formula. We have explained it in the manuscript.

**Reviewer:**

81-80: Explain "cross-divide χ ratio". Ratio between what and what?

**RESPONSE:**

It is the ratio of χ values on both sides of a drainage divide ($\frac{\chi_\alpha}{\chi_\beta}$). We have revised this sentence.

**Reviewer:**

81-84: This sentence doesn't make sense to me. First, something is missing here- you need to mention that the technique of Zhou et al 2022 requires channel head parameters before saying that these parameters are problematic. Second, what are the previous studies you are talking about? Are you referring to a case-specific study? and why this parameter is not applicable in specific natural areas?

**RESPONSE:**

We have revised these sentences: "Braun (2018) provided an equation that considers both alluvial and fluvial areas to calculate the migration velocity of an escarpment (also a drainage divide). Zhou et al. (2022a) developed a technique to calculate the migration rate through the high base-level χ values on both sides of a drainage divide. The above approaches require channel-head parameters to calculate the migration rate. However, the location of the channel heads sometimes cannot be accurately identified, limited by the resolution of DEM in natural cases. For this reason, empirical values of channel-head parameters are used in the above study, which may induce uncertainties".

**Reviewer:**

85: Before this sentence, I suggest providing the knowledge gap and saying what problem you are addressing (e.g., "Currently, there is no method for quantifying divide migration rates across large landscapes.. ").

**RESPONSE:**

We have added a sentence here and provided the knowledge gap: "… empirical values of channel-head parameters are used in the above study, which may induce uncertainties. This study aims to establish an approach to derive the migration rate of drainage divides, at a high precision and low cost, based on topographic analysis".

**Reviewer:**

91-93: When first reading this, it is unclear how you calculated the migration rates from the parameters. Perhaps write that you developed two methods for this and that one is based on the method of Zhou 2022.

**RESPONSE:**

We have revised this sentence: "We then developed two methods to calculate the drainage-divide migration rates. One is based on the measured channel-head parameters, and the other is based on the improved method of Zhou et al (2022a)".

**Reviewer:**

93: When you mentioned the method of Zhou et al. 2022, you didn't say that this method doesn't apply to cases where the elevations of outlets and channel heads are different across the divide (isn't this almost always the case?).

**RESPONSE:**

In natural cases, the elevations of channel heads are usually different. However, limited by the resolution of DEM, the location of the channel heads sometimes cannot be accurately identified. Therefore, the channel head parameters across the divide for calculating the migration rates are based on the same empirical values in Zhou et al. (2022a), which may induce uncertainties. We have clarified it in the manuscript.

**Reviewer:**

96: This sounds like a good sentence to open the paragraph.

**RESPONSE:**

We have moved this sentence to the beginning of this paragraph.

**Reviewer:**

98: Maybe delete "moreover."

**RESPONSE:**

We have deleted it.

**Reviewer:**

99: "Quantify the influence" is not clear to me. Do you mean Quantify the cross-divide difference in uplift rates?

**RESPONSE:**

We have changed the phrase to "quantify the cross-divide difference in uplift rates to improve the precision of drainage-divide migration rate".

**Reviewer:**

109-111: Landslide threshold and erosion threshold are not mechanisms- they are thresholds. If you mention them, you should explain what these thresholds are, and why do they cause the channels to emerge at a certain distance from the divide.

**RESPONSE:**

We have changed the word "mechanisms" to "thresholds" and added explanations of the thresholds: "Because of thresholds such as erosion threshold (the shear stress of overland flow must exceed the threshold of the cohesion of bed material to generate river incision) (Howard and Kerby, 1983; Perron et al., 2008) or landslide threshold (landslides occur when the threshold of soil or rock strength is exceeded in high relief region) (Burbank et al., 1996; Tucker et al., 1998), river channels (following Eq. 1) emerge at a certain distance from the drainage divide".

**Reviewer:**

121: Slope is measured between two points. What are these two points in this case? Do you measure only from the channel head and downstream (because the channel head is the highest point of the fluvial segment)

**RESPONSE:**

We first plot the river long profiles and locate the channel-head points on the profiles. Then, we calculate the average slopes of the channel segments near the channel-head points (100m-long segments are enough) and use them as the tangent slopes of the channel-head points. We have revised this sentence to "$S_{ch}$ is the channel-head gradient measured along the channel near the channel-head point."

**Reviewer:**

122-131, including figure 1: I think it is obvious that increasing S, A, and K, or decreasing the m/n ratio, will result in higher erosion rates. Do I miss something here? Please clarify or eliminate this part+ this figure.

**RESPONSE:**

We have removed this part and added the sentence: "Eq. 2 indicates that the side of the drainage divide with a higher $A_{cr}$ or $S_{ch}$ can have a higher erosion rate than the other side, and is more likely to pirate the opposite drainage basin. Besides, a high erosion coefficient can amplify the drainage basin's erosion rate". We also moved Figure 1 to the supplementary materials.

132: Repetitive with the sentence in line 54, see comment there.

**RESPONSE:**

We have also changed this sentence: "Drainage-divide migration is essentially controlled by the cross-divide difference in erosion rates and topographic slope (Beeson et al., 2017; Dahlquist et al., 2018; Chen et al., 2021; Zhou et al., 2022a; Stokes et al., 2022)".

**Reviewer:**

139: Eq 3 suggests that migration is also a function of the gradients of the slopes across the divide.

**RESPONSE:**

Yes, we have added this information in the sentences (see the response of Line 54 & 132).

**Reviewer:**

142: I think you can delete "rock."

**RESPONSE:**

We have deleted it.

**Reviewer:**

147: I suggest moving this sentence to line 141 where α and β are first presented.

**RESPONSE:**

We have moved this sentence there.

**Reviewer:**

157: Add citation for measuring erosion rates from cosmogenic nuclides

**RESPONSE:**

We have added the citation here: "Eα and Eβ are the erosion rates of the α to β side of drainage divide, respectively, which can be derived through the cosmogenic nuclides (10Be) concentration measurements (Beeson et al., 2017; Godard et al., 2019; Hu et al., 2021)".

**Reviewer:**

157: Please clarify what is the regional average erosion rate. Do you mean (Eα + Eβ)/2 ?

**RESPONSE:**

Yes, it is. We have clarified it in the manuscript: "The regional average erosion rate $(\bar{E} = \frac{E_\alpha + E_\beta}{2})$ can also be used to calculate the migration rate".

**Reviewer:**

162: It can help to add the symbol, i.e., "erosion coefficient (K).." etc.

**RESPONSE:**

We have added the symbols here: "… combined with one of the erosion-related parameters, erosion coefficient ($K$), erosion rate at one side of a drainage divide ($E_\alpha$ or $E_\beta$), or regional average erosion rate ($\bar{E}$)".

**Reviewer:**

173: What do mean by channel head segments? The segment between the divide and the channel head? If yes, better define in the text.

**RESPONSE:**

We have added the definition in the manuscript: "Channel-head-segment is the channel segment just below the channel-head".

**Reviewer:**

174-176. About Ksn -See comment to line 71 and general comments.

**RESPONSE:**

We have provided a clear definition of ksn in the manuscript (see the RESPONSE of general comments 4).

**Reviewer:**

This is the missing information for the comment in line 84. Nonetheless, I don't understand how the channel head and the outlet of two channels across the divide can be the same elevation in natural settings.

**RESPONSE:**

The elevation of the two channel heads across a divide is usually different in natural settings. But limited by the resolution of DEM, we sometimes cannot accurately identify the location of the channel heads. Therefore, we simplified the calculation according to an empiric value ($A_{cr} = 10^5$ m$^2$) in Zhou et al., 2022a. In this study, benefiting from the high-resolution DEM, we can accurately identify the location of channel heads and do not need the empiric value.

**Reviewer:**

183: need to explain about $\chi$ in the place it first appears (see comment on line 74)

**RESPONSE:**

Yes, we have explained more about the $\chi$ parameter and added the equation (see the RESPONSE of overall comment 4).

**Reviewer:**

199-206: This reads more as part of the methods, in particular lines 203-206.

**RESPONSE:**

We have moved the sentences to the Methods section.

**Reviewer:**

208, Fig 2: Beautiful map! Some notes:

-Please provide a source of information for the maps.

**RESPONSE:**

We have provided the source of information in the figure caption: "The topography data (ALOS DEM, 12.5 m resolution) is downloaded from the Alaska Satellite Facility (ASF) Data Search (https://search.asf.alaska.edu/). The fault data is downloaded from the site (https://www.activefault-datacenter.cn/). The satellite image is downloaded from Google Earth".

**Reviewer:**

-I suggest annotating each map differently (A and B), and refer to them in the captions accordingly.

**RESPONSE:**

We have revised it accordingly.

**Reviewer:**

-Maybe add a legend for the faults. Are they all normal faults? If not, mark them with simple curves without the small lines that stick out.

**RESPONSE:**

We have added the legend for the faults and marked them with simple curves. Most of them are normal faults, with several thrust fault and strike-slip faults in the southwest corner. We have revised accordingly. See the revised version of the figure below.

[Figure]

**Reviewer:**

-How come the highest elevation in the map is exactly 5000 m (should be m asl-above sea level)? If this is not exactly the value, please fix to the correct max elevation in the map.

**RESPONSE:**

The highest elevation on the map is 3652m. We have fixed the range of the colour bar to the min-max elevation (3-3652m) and redrawn the map (See the revised figure in the RESPONSE above).

**Reviewer:**

214: Refer to fig 2.

**RESPONSE:**

We have referred to it here.

**Reviewer:**

216: But figure 3A does show a fault in the south (Xizaou chen)? Do you mean that the tilting of the Wutai Shan is dominated mostly by the activity of the northern fault?

**RESPONSE:**

Yes, the tilting of the Wutai Shan is dominated by the activity of the Northern Wutai Shan fault. The Xizhou Shan fault in the south controls the tilting of the Xizhou Shan.

**Reviewer:**

218-219: As I wrote in the overall comment 5- how can you explain no variation in precipitation in an area with an elevation difference of ~2500 m?

**RESPONSE:**

We have provided the precipitation and geological maps in the supplementary materials (See the response of the overall comment 5).

**Reviewer:**

220-228: Maybe this part can fit better in the methods section. Aside from the comments in overall comment 5, the calculation of erosion coefficient here is not clear unless you provide the equation that relates between kns, k , and erosion rates.

**RESPONSE:**

We have provided the equation for calculating K in the manuscript: "Combing with the equation, $K = \frac{E}{k_{sn}{}^n}$, and following the approach of previous studies (Kirby and Whipple, 2001; Kirkpatrick et al., 2020; Ma et al., 2020), the erosion coefficient (K) is calculated to $(6.25 \pm 3.13) \times 10^{-6}$ $m^{0.1}yr^{-1}$ in this area". Because this calculation is only for the Wutai Shan area, we prefer to leave it in the case study section.

**Reviewer:**

224-225: Add a reference to fig 3. Also- is the erosion rate based on all five samples illustrated in figure 3?

**RESPONSE:**

We have added a reference to Fig. 3 here. The low-temperature thermochronology result is based on all five samples. Besides, we have changed the estimation of erosion rate, considering both the geological and low-temperature thermochronology studies: "The geological study shows that the Quaternary throw rates of the Northern Wutai Shan fault are 0.8-1.6 mm/a (Middleton et al., 2017). The low-temperature thermochronology study shows that the time-averaged long-term throw rates in the late Cenozoic is about 0.25 mm/yr, and there is an accelerated activity in the Wutai Shan area (Clinkscales et al., 2020). According to the above geological and low-temperature thermochronology studies, we assume a 0.5±0.25 mm/yr uplift/erosion rate in the northern margin of the Wutai Shan (at the footwall of the Northern Wutai Shan fault)".

**Reviewer:**

230: What do you mean by "randomly"? maybe delete. Same for line 302

**RESPONSE:**

We have deleted the word "randomly".

**Reviewer:**

231: How did you measure the slope (see overall comment 3)? Was it extracted from the high-resolution DEM?

**RESPONSE:**

Yes, the slope is extracted from the high-resolution DEM using the TAK (Forte and Whipple, 2019) and TopoToolbox (Schwanghart and Scherler, 2014).

**Reviewer:**

232: You probably mean "the breaking point of the slope-area regression line? Was this point determined manually or by an automatic algorithm?

**RESPONSE:**

Yes, we have changed the sentence to "the breaking point of the slope-area regression line". This point is determined manually.

**Reviewer:**

237: What average- the average slope of all points measured between the channel head and the divide? Also- why do use only the points that are upstream from the channel head? They represent the hillslope section and not the fluvial segment of the channel.

**RESPONSE:**

We agree that the points that are upstream from the channel head represent the hillslope section. We have changed the way to calculate $S_{ch}$, which is calculated along the tangent lines of the channel-head points on the channels. The calculation process is in the RESPONSE of overall comment 3. Besides, the tanα and tanβ are calculated through the average slope of the hillslope segments near the channel head (not including the hilltop part).

**Reviewer:**

238-240: I'm not sure I understood from the text: zb and zch and the χ values were extracted from the χ plot? If yes, please rephrase for making the text more clear.

**RESPONSE:**

We have clarified it in the manuscript: "According to the location of the river's outlets (Fig. 4A), we obtain the river's outlet elevations (zb) from the river's long profiles", "For the χ-plots (Figs. 4C, F, I), we obtain the elevations of channel heads (zch) and χ values according to the coordinate of the channel-head points".

**Reviewer:**

243: "The normal direction of the boundary fault"? what does this mean? Parralal or perpendicular to the direction of the fault?

**RESPONSE:**

We mean the distance perpendicular to the direction of the boundary fault. We have changed the sentence in the manuscript.

**Reviewer:**

246: Are you following previous studies that generally used these values or specific studies that used similar values in your study area?

**RESPONSE:**

We followed previous studies that generally used these values. We have also added the results of m = 0.35 and 0.55 in the supplementary materials as references for other situations.

**Reviewer:**

251-252: The sentence "…show that distinct character of the rivers across the drainage divide" doesn't make sense to me. Please try a different phrasing.

**RESPONSE:**

We have changed this sentence to "The rivers have different characteristics on both sides of the drainage divide according to their slope-area plots (Figs. 4B, E, H) and the χ-plots (Figs. 4C, F, I)".

**Reviewer:**

259: Suggest adding "in this area" at the sentence's end.

**RESPONSE:**

We have added the phrase "in this area" at the sentence's end.

**Reviewer:**

Fig 3: A. cite Clinkscales et al 2020 also in the legend. B. See comment on elevation scale in fig 2- same for the ksn scale here.

**RESPONSE:**

We have changed the figure accordingly:

[Figure]

**Reviewer:**

265: Same comment about the faults in Fig 2.

**RESPONSE:**

They are all normal faults. We have added the dots on the fault symbols (see the revised figure above).

**Reviewer:**

267: Maybe write "Black rectangles show the area of Fig. 4A".

**RESPONSE:**

We have changed the sentence here accordingly.

**Reviewer:**

271: I think that "based on matlab" can be deleted.

**RESPONSE:**

We have deleted the phrase.

**Reviewer:**

272: What software did you apply for the kringing method?

**RESPONSE:**

It is on ArcGIS. We have noted it in the manuscript.

**Reviewer:**

273: Did you use topotoolbox also for the swath profile? if yes please mention it.

**RESPONSE:**

Yes, we use topotoolbox also for the swath profile. We have mentioned it in the manuscript.

**Reviewer:**

273: Not dotted, dashed.

**RESPONSE:**

Yes, we have changed it.

**Reviewer:**

275: I recommend showing the extent of the swath profile in figs A&B.

**RESPONSE:**

We have added the extent of the swath profile in the figure (see the RESPONSE of Fig. 3 for the revised version).

**Reviewer:**

Suggestions to figs. 4 and 5. fig A: i) Specify "profiles" in the legend – I think you mean topographic profiles. ii) Add the arrows and the divide to the legend - maybe make the arrow length proportional to the migration rate. iii) Complete the divide line. iv) Maybe change the scale in the bottom left corner to white. v) I think that you can eliminate m=0.45 and n=1 in the middle columns. V) I am not sure it is right to continue the $\chi$ profile up to the divide (ie, upstream from the channel head), as this segment is not fluvial and is not applicable for $\chi$ analysis.

**RESPONSE:**

We have changed the figs. 4 & 5 accordingly (see the revised fig. 4 below):

i) We have changed "profiles" to "topographic profiles" in the figure.

ii) We have added the arrows and the divide to the legend and made the arrow length proportional to the migration rate.

iii) We have completed the divide line.

iv) We have changed the scale in the bottom left corner to white.

v) We have eliminated the m=0.45 and n=1 in the middle columns.

vi) We do not use the $\chi$ profile in the hillslope area (shown in dashed lines) for calculation. We plot them only to show the location of channel heads.

[Figure]

**Reviewer:**

280: Please explicitly specify the resolution. Also, this is not a topographic map but a hillshade map.

**RESPONSE:**

We have added the resolution information in the manuscript (0.67 m and 0.84 m spatial resolution in the Wutai Shan and the Yingwang Shan respectively). We also have changed the word "topographic" to "hillshade".

**Reviewer:**

292: Should be "methods."

**RESPONSE:**

We have changed it to "methods".

**Reviewer:**

294: Maybe give it a name of your own?

**RESPONSE:**

We have named it "Yingwang Shan".

**Reviewer:**

296-7: Please check the grammar and correct accordingly.

**RESPONSE:**

We have corrected it: "Over the past 2.6 million years, it has accumulated tens to hundreds of meters of eolian sediments (Yan et al., 2014), draping preexisting topography (Xiong et al., 2014)".

**Reviewer:**

298: Maybe connect this short sentence to one of the sentences before or after it.

**RESPONSE:**

We have connected this sentence with the following sentence: "There is no active fault and is little to no variation in rock erodibility and precipitation within the area (Shi et al., 2020; Zhou et al., 2022b)".

**Reviewer:**

301-309: This subsection is repetitive with the previous one. Perhaps just write that you extracted the migration rates similar to Wutai Shan site.

**RESPONSE:**

We have simplified the sentences: "Similar to the Wutai Shan site, we make the slope-area plots (Figs. 5 B, E, H) and the $\chi$-plots (Figs. 5 C, F, I), and extract the values of Acr, Sch, zb, zch, $\chi$, tan$\alpha$ and tan$\beta$ of the rivers".

**Reviewer:**

311: Is this an average erosion rate? If yes, what timescale does it average? How was this rate extracted? Please elaborate on this.

**RESPONSE:**

The data can represent the present-day erosion rate because it is calculated through the distribution of the current river's silt discharge. We have clarified it in the manuscript: "The rate of soil erosion in the study area is about 500 t·km-2yr-1 according to the distribution of silt discharge (Fu, 1989). Combining with the assumption of the density of loess, 1.65 t m-3, the present-day erosion rate in the study area is calculated to be 0.3 mm yr$^{-1}$".

**Reviewer:**

337: Should be "in the two field cases."

**RESPONSE:**

We have changed it to "in the two field cases.".

**Reviewer:**

Table 1: The lack of uncertainties here is substantial.

**RESPONSE:**

We have added the uncertainties in Table 1. See the RESPONSE to the overall comment 5 for details.

**Reviewer:**

349-350: These sentences are almost similar to those in line 72. As I wrote in overall comment 8, I think that all of this subsection should be merged with the introduction.

**RESPONSE:**

We have deleted this sentence to avoid repetition. We also revised the introduction section.

**Reviewer:**

355: "These methods"- do you refer to $\chi$, Gilbert methods, or Zhou's? Or all?

**RESPONSE:**

We have changed "these methods" to "the Gilbert metrics or χ-comparison method in Zhou et al. (2022a)" to avoid ambiguity.

**Reviewer:**

362: What are the empirical parameters on which the channel head location relies on? And what is the problem with them? Do you mean that the location depends on the break of the slope, which cannot be determined appropriately in DEMs with a coarse resolution?

**RESPONSE:**

Yes, the location of a channel head is hard to accurately identified with a coarse resolution of DEM. Therefore, the calculation in Zhou et al. (2022a) uses the empiric value ($A_{cr} = 10^5$ m$^2$), which may induce uncertainties. We have added more explanation in the manuscript.

**Reviewer:**

363: Perhaps: "which can induce uncertainties in determining the stability of drainage divides".

**RESPONSE:**

We have changed the sentence accordingly.

**Reviewer:**

382: I am confused- you are saying that the differential uplift rate should be ignored or taken into account?

**RESPONSE:**

It should be taken into account. We have deleted the sentence "Although the differential uplift rate is …" to avoid misunderstanding.

**Reviewer:**

394: You assume that divide migration rate decreases as the divide becomes closer to steady state. Please provide a reference to this argument. However, this also implies that the divide migration rate may change over long timescales. This might not be consistent with the parameters that were used for calculating k (Wutai Shan) or E (Loess plateau), which represent an average over long timescales.

**RESPONSE:**

The evidence is mainly from numerical simulation (Whipple et al., 2017; Ye et al., 2022), and we have added the reference here. We have also added the error range of the erosion rate in the Wutai Shan (see the RESPONSE to the overall comment 5). In the Loess Plateau, the drainage divide is stable and the data can represent the present-day erosion rate because it is calculated through the distribution of the current river's silt discharge.

[Figure]

Fig. 10 in Whipple et al. (2017)

[Figure]

Fig. 1 in Ye et al. (2022)

---

## Author Response (AR1)

**Response to Reviewers**

**1. Response to Thomas Bernard:**

**Thomas Bernard:**

Zhou et al. present in their manuscript a framework to determine the migration rate of drainage divide and apply the methods to two study cases in the Wutai Shan and Loess Plateau. The authors developed new methods of drainage divide migration rate estimation using high-topographic data based on channel-head parameters, values of channel segments, and erosion rate parameters. They argue that by determining the exact location of the channel heads, the migration rate of drainage divide can be accurately calculated. They determined drainage divide migration rate of about 0.10-0.13 mm.yr-1 and closed to zero for the Wutai Shan and Loess Plateau study cases. Finally, they suggest that the difference in uplift on both sides of the drainage divide have to be considered in order to calculate drainage divide migration rate with this method.

The topic of drainage reorganisation by drainage divide migration or river capture is receiving increasing attention these years. Therefore, this contribution, which determined the rate at which drainage divide migrate, is timely and should be of interest to the EGUsphere journal. This study also presents a nice follow-up of drainage divide migration estimation to the study by Zhou et al., 2022 (although I find the two manuscripts really close). The manuscript is interesting and overall well-written. The provided model, methods and equations in the manuscript are sound and well-used in the study area cases. There are a few awkward sentences that I have indicated below. Finally, I found the figures well realized and easy to understand.

My main comments/concerns regarding this manuscript are the use of strong assumptions made in the calculation of the drainage divide rate migration which have not been mentioned in the main text. The first one is the use of erosion rate estimate

from low-temperature thermochronology data, which cannot correspond to a modern rate for the Wutai Shan study case. The second is the systematic use of a "standard" concavity index for both study cases. I feel like these assumptions have to be addressed. Since the main message of the study is to quantify migration rate from high-resolution topographic data, estimation of the equation parameters needs to follow the same logic. More specific comments tied to line number:

**RESPONSE to the general comments and CHANGES:**

1. We agree that the erosion rate estimated from low-temperature thermochronology data may not represent the present-day situation. Therefore, we also refer to the results of geological studies and added the error range to cover this uncertainty: "Previous geological study shows that the Quaternary throw rates of the Northern Wutai Shan fault are 0.8-1.6 mm/a (Middleton et al., 2017). The low-temperature thermochronology study shows that the time-averaged long-term throw rates in the late Cenozoic is about 0.25 mm/yr, and there is an accelerated activity in the Wutai Shan area (Clinkscales et al., 2020). According to these studies, we assume a 0.50± 0.25 mm/yr uplift/erosion rate in the northern margin of the Wutai Shan (at the footwall of the Northern Wutai Shan fault)." Lines 336-346 (**in the manuscript with changes marked, same hereinafter**). The error of the erosion rate has also been calculated into the migration rates (Table 1). We have also added a "Limitations and uncertainties" section in the manuscript to discuss the limitations of the methods. Lines 595-632.

2. We also agree that using the "standard" concavity index for the study cases may bring errors in the results. For this reason, we have added the results of m/n = 0.35 and 0.55 as a reference for different situations. We have also discussed it in the "Limitations and uncertainties" section that further estimation of the m/n value (Mudd et al., 2018) in the study cases could improve the accuracy of the results. Lines 595-632.

**Table 1. Channel parameters and the migration rates of the drainage divides in the two field cases.**

| Natural Cases | No. | $A_{cr}$ (×10⁵m²) | $S_{ch}$ | $z_b$ (m) | $z_{ch}$ (m) | $\chi$ | $\tan\alpha$ | $\tan\beta$ | $\Delta U_{ch}$ (mm/yr) | $D_{mr}$ (mm/yr) (Channel-head-point method) | $D_{mr}$ (mm/yr) (Channel-head-segment method) |
|---|---|---|---|---|---|---|---|---|---|---|---|
| Wutai Shan | Fig. 4 Iα | 1.75 | 0.16 | 1631 | 1792 | 6.4 | 0.14 | 0.66 | ~ 0.008 | -0.21 ±0.10 | -0.26 ±0.12 |
| | Fig. 4 Iβ | 0.26 | 0.63 | 1347 | 1723 | 6.6 | | | | | |
| | Fig. 4 IIα | 0.79 | 0.23 | 1630 | 1815 | 5.4 | 0.24 | 0.70 | ~ 0.008 | -0.23 ±0.11 | -0.27 ±0.13 |
| | Fig. 4 IIβ | 0.30 | 0.67 | 1351 | 1809 | 6.1 | | | | | |
| | Fig. 4 IIIα | 0.67 | 0.29 | 1633 | 1860 | 5.0 | 0.28 | 0.65 | ~ 0.008 | -0.21 ±0.10 | -0.22 ±0.10 |
| | Fig. 4 IIIβ | 0.39 | 0.63 | 1352 | 1875 | 6.9 | | | | | |
| Yingwang Shan | Fig. 5 Iα | 0.54 | 0.21 | 1111 | 1224 | 5.8 | 0.21 | 0.31 | 0 | ~ 0.03 | ~ -0.01 |
| | Fig. 5 Iβ | 0.20 | 0.32 | 1126 | 1225 | 5.0 | | | | | |
| | Fig. 5 IIα | 0.24 | 0.36 | 1111 | 1257 | 7.4 | 0.39 | 0.33 | 0 | ~ 0.02 | ~ -0.01 |
| | Fig. 5 IIβ | 0.30 | 0.31 | 1117 | 1224 | 5.4 | | | | | |
| | Fig. 5 IIIα | 0.29 | 0.46 | 1089 | 1256 | 8.6 | 0.49 | 0.35 | 0 | ~ 0.02 | ~ -0.01 |
| | Fig. 5 IIIβ | 0.56 | 0.37 | 1096 | 1203 | 5.3 | | | | | |

[Figure]

**Figure S4**. Normalized channel steepness ($k_{sn}$) distribution in the Wutai Shan. (**A**) The $k_{sn}$ distribution at $m = 0.35$ and $n = 1$. (**B**) The $k_{sn}$ distribution at $m = 0.55$ and $n = 1$. The black dashed line shows the location of the main drainage divide. Red lines show

the main active faults. The black straight lines show the location of the profiles G-G'
and H-H' and the gray rectangles show the area of the swath profiles in Panel C&D.
The topography data (ALOS DEM, 12.5 m resolution) is downloaded from the Alaska
Satellite Facility (ASF) Data Search (https://search.asf.alaska.edu/). The $k_{sn}$ is
calculated using TopoToolbox (Schwanghart and Scherler, 2014), and the
interpolation uses the Kriging method on ArcMap. (**C**) The $k_{sn}$ swath profile along G-
G' in Panel A. (**D**) The $k_{sn}$ swath profile along H-H' in panel B. The swath profiles are
extracted using TopoToolbox (Schwanghart and Scherler, 2014). The red dashed lines
show the location of the main active faults, and the black arrow shows the location of
the main drainage divide. Both swath profiles are 20 km wide (10 km on each side).
The extent of the swath profiles is represented by the grey boxes in Panel A&B.

**Table S1**. The comparison of drainage divide's migration rates at $m$ = 0.35, 0.45 and
0.55 in the Wutai Shan area.

| No. | $A_{cr}$ ($\times10^5 m^2$) | $S_{CH}$ | K ($\times10^{-6}m^{0.3}yr^{-1}$) (n=1; m=0.35) | K ($\times10^{-6}m^{0.1}yr^{-1}$) (n=1; m=0.45) | K ($\times10^{-6}m^{-0.1}yr^{-1}$) (n=1; m=0.55) | tanα | tanβ | Rate (mm/yr) (n=1; m=0.35) | Rate (mm/yr) (n=1; m=0.45) | Rate (mm/yr) (n=1; m=0.55) |
|---|---|---|---|---|---|---|---|---|---|---|
| Fig. 4 $I_\alpha$ | 1.75 | 0.16 | 25.00±12.50 | 6.25±3.13 | 1.43±0.72 | 0.14 | 0.66 | -0.37±0.18 | -0.21±0.10 | -0.10±0.05 |
| Fig. 4 $I_\beta$ | 0.26 | 0.63 | 25.00±12.50 | 6.25±3.13 | 1.43±0.72 | | | | | |
| Fig. 4 $II_\alpha$ | 0.79 | 0.23 | 25.00±12.50 | 6.25±3.13 | 1.43±0.72 | 0.24 | 0.70 | -0.36±0.17 | -0.23±0.11 | -0.14±0.06 |
| Fig. 4 $II_\beta$ | 0.30 | 0.67 | 25.00±12.50 | 6.25±3.13 | 1.43±0.72 | | | | | |
| Fig. 4 $III_\alpha$ | 0.67 | 0.29 | 25.00±12.50 | 6.25±3.13 | 1.43±0.72 | 0.28 | 0.65 | -0.31±0.15 | -0.21±0.10 | -0.13±0.06 |
| Fig. 4 $III_\beta$ | 0.39 | 0.63 | 25.00±12.50 | 6.25±3.13 | 1.43±0.72 | | | | | |

**Thomas Bernard:**

Line 37-39: "The evolution of topography is fundamentally coupled with changes in
drainage systems, including river's vertical and lateral movements". Can you be more
precise about vertical and lateral movements? Maybe add river capture as another
important process for drainage changes.

**RESPONSE and CHANGES:**

We have changed the sentence: "The evolution of unglaciated terrestrial terrains is fundamentally coupled with changes in drainage systems, which are river's vertical (changes in river long profile) and lateral movements (drainage divide migration and river captures)". Lines 59-62.

**Thomas Bernard:**

Line 70-71: "For example, Willett et al. (2014) developed the χ method to map the dynamic state of river basins". I don't think Willet et al (2014) developed the χ method. Please change the term "developed" by "applied" or cite "Royden et al., 2000; Perron and Royden, 2012" instead.

**RESPONSE and CHANGES:**

We have changed the term "developed" by "applied". Line 109.

**Thomas Bernard:**

Line 79-81: "Zhou et al. (2022a) developed a technique to calculate the migration rate through the cross-divide χ ratio of high base-level channel segments". This statement is in contradiction with line 78 "No rates have been obtained".

**RESPONSE and CHANGES:**

We have deleted the sentence "No rates have been obtained." to avoid misunderstanding. Line 116.

**Thomas Bernard:**

Line 89-90: "to obtain the high-resolution topography data of these two areas". Can you precise the resolution of your topography data here or somewhere in the manuscript.

**RESPONSE and CHANGES:**

We have added the resolution information in the manuscript: "to obtain the high-resolution DEM data of these two areas (0.67 m and 0.84 m spatial resolution in the Wutai Shan and the Yingwang Shan respectively)". Lines 136-138.

**Thomas Bernard:**

Line 98: "Moreover, benefiting from the detailed tectonic research". Please reformulate or precise what is this "tectonic research".

**RESPONSE and CHANGES:**

We have changed the sentence: "Combining with the geological and low-temperature thermochronology studies (Middleton et al., 2017; Clinkscales et al., 2020)". Lines 150-152.

**Thomas Bernard:**

Line 127-128: "A large erosion coefficient also creates a high channel-head erosion rate". This sentence is unclear. What does the term "erosion coefficient" refer to?

**RESPONSE and CHANGES:**

We mean the parameter "$K$" in Eq. 1. We have added the symbol, "($K$)", in the manuscript. Line 207.

**Thomas Bernard:**

Equation 4: This equation is correct only if the erodibility is the same on both side of the drainage divide. Correct the equation or precise this assumption in the text. This is important, especially, since you demonstrate in your Figure 1 that the erodibility affect the channel-head erosion rate.

**RESPONSE and CHANGES:**

We have clarified this assumption in the text: "Assuming erosion coefficient (K) is the same on both sides of a drainage divide, Eqs. 2 and 3 allow us to derive the equation of drainage divide's migration rate according to the parameters at the channel-head points". Lines 207-210.

**Thomas Bernard:**

Line 184-185: integral function of channels' upstream area (A) to horizontal distance (x) (Perron and Royden, 2012; Willet et al., 2014)". Replace the reference "Willet et al., 2014" by "Royden et al., 2000".

**RESPONSE and CHANGES:**

We have replaced the reference "Willet et al., 2014" by "Royden et al., 2000".  Lines 245-246.

**Thomas Bernard:**

Line 241: "If we assume the rock uplift rate decreases linearly from 0.25 to 0 mm/yr from northwest to southeast of the Wutai Shan horst". If I understood correctly the 0.25 mm/yr rate come from the low-temperature thermochronology study. Even if this rate is predicted for the late Cenozoic, it cannot reflect the modern erosion rate (the method is not sensible) needed to accurately calculate the modern drainage divide. It assumes that the erosion rate stayed constant. This assumption is not reported in this paragraph or the discussion.

**RESPONSE and CHANGES:**

We have added the discussion in the "Limitations and uncertainties" section to report this error: "Second, in the case study of the Wutai Shan, we refer to the geological and low-temperature thermochronology studies, and assume a 0.50±0.25 mm/yr erosion

rate at the northern margin of the Wutai Shan (i.e., the footwall of the North Wutai Shan fault). Combining with the present-day $k_{sn}$, we calculate the erosion coefficient ($K$) and derive the migration rates of the drainage divide. If the present-day erosion rate deviates from the assumption, the error would exist in the results". Lines 617-623. The calculation and the results (Table 1) are in RESPONSE to the general comments.

**Thomas Bernard:**

Line 246: "We assume n = 1 and m = 0.45 in the calculation following previous studies (Wobus et al., 2006; DiBiase et al., 2010; Perron and Royden, 2012; Wang et al., 2021)". This is a reference concavity of 0.45 that you are using for your calculation. How does this concavity really reflect the geomorphology of your study area? There is available framework to calculate the channel concavity of your catchment, check Mudd et al., 2018. This is even more important since you can have different concavities on both side of your drainage divide and strongly affect your migration rate calculations based on your equations. Line 313: "We also assume n = 1 and m = 0.45 in the calculation". Same as previous comment.

**RESPONSE and CHANGES:**

We have added the discussion in the "Limitations and uncertainties" section: "we use the typical values of n = 1 and m/n = 0.45 in the two natural cases to calculate the migration rate. If the actual values largely deviate from the assumption, errors would be introduced into the results. For this reason, we have added the cases of m/n = 0.35 and 0.55 in the in the Supplementary Materials. Further estimation of these values (Mudd et al., 2018) could improve the accuracy of the results". Line 612-617. The calculation and the results (Figs. S4 and Table S1) are in the RESPONSE to the general comments.

**Thomas Bernard:**

Line 294: "An unnamed mountain range in the Loess Plateau". The term "unnamed" sounds strange. I suggest to just remove it or find another solution.

**RESPONSE and CHANGES:**

We have named the mountain as "Yingwang Shan". Line 470.

**Thomas Bernard:**

Figure 1: It might be better to directly indicate the value of in the different panel instead of the ratio. You can also precise the erodibility for each panel in the figure description since you did it for the slope and area coefficients.

**RESPONSE and CHANGES:**

We have changed the Fig. 1 (Fig. S1 in the new version) accordingly and clarified the erodibility in the figure description. Line 231.

**Thomas Bernard:**

Figure 2: Precise the meaning of the white rectangle in the description like for the back rectangles.

**RESPONSE and CHANGES:**

We have clarified it in the figure description: "White rectangles show the location of Panel A". Line 305.

**Thomas Bernard:**

Figure 4: The text of Eq. 4 and 8 could be coloured in black and blue respectively in order to directly identified which arrows correspond to which equations and rates. Or you could add small arrow legends on the left of the text.

**RESPONSE and CHANGES:**

We have changed the colour of the text to black and blue respectively. Line 451.

**Thomas Bernard:**

Technical corrections:

Line 60-62: "However, these techniques are usually based on samples collected from an outlet that is several kilometers away from the drainage divide and thus may not represent the erosion rates close to the drainage divide". Change "an outlet" to "a catchment outlet".

**RESPONSE and CHANGES:**

We have changed "an outlet" to "a catchment outlet" in the manuscript. Lines 98-99.

**Thomas Bernard:**

Line 65-67: "Hence, it would be ideal to find an accessible and efficient method that can be applied to the entire landscape and cross-checked to make full use of the 10Be-derived erosion rates". The term "cross-checked" is unclear in this sentence.

**RESPONSE and CHANGES:**

We have deleted the term "cross-checked" to avoid misunderstanding: "Hence, it would be ideal to find an accessible and efficient method that can be applied to the entire landscape and make full use of the 10Be-derived erosion rates". Lines 103-105.

**Thomas Bernard:**

Line 68-69: "The advancement of remote sensing technology has promoted the development of geomorphic analysis theory". Remove the term "theory".

**RESPONSE and CHANGES:**

We have removed the term "theory" and changed this sentence: "The advancement of the digital elevation model (DEM) has promoted the development of geomorphic analysis,…". Lines 106-107.

**Thomas Bernard:**

Line 77: "No rates have been obtained". This short sentence might be combined with another one.

**RESPONSE and CHANGES:**

We have deleted this short sentence. Line 116.

**Thomas Bernard:**

Line 86: "an unnamed mountain range in the Loess plateau". Remove the term "unnamed".

**RESPONSE and CHANGES:**

We have removed it and changed the sentence to "the Yingwang Shan in the Loess Plateau". Line 134.

**Thomas Bernard:**

Line 128-129: "The results indicate that the side with a higher Acr or Sch can have a higher erosion rate than the other side of the drainage divide". Move the terms "of the drainage divide" to the first occurrence of the term "side".

**RESPONSE and CHANGES:**

We have moved it: "Eq. 2 indicates that the side of a drainage divide with a higher $A_{cr}$ or $S_{ch}$ can have a higher erosion rate than the other side". Lines 182-187.

**Thomas Bernard:**

Line 134-136: "when one uses the cross-divide erosion rates … one should also consider the influence of differential uplift rates". I suggest to remove the terms "when one" and "one should" and reformulate the sentence.

**RESPONSE and CHANGES:**

We have changed the sentence to "Furthermore, the differential uplift should also be considered when using the cross-divide erosion rates …". Lines 192-194.

**Thomas Bernard:**

Line 144: "Combining Eqs. 2 and 3, one can derive the equation". Same as previous comment, replace "one can" by "allow to" for example.

**RESPONSE and CHANGES:**

We have replaced "one can" by "allow to": "Eqs. 2 and 3 allow us to derive the equation of drainage divide's migration rate …". Lines 208-209.

**Thomas Bernard:**

Line 216-217: Correct to "The bedrock of the Wutai Shan area consists mainly of metamorphic and igneous basement rocks".

**RESPONSE and CHANGES:**

We have corrected this sentence accordingly. Lines 313-314.

**Thomas Bernard:**

Line 263: Change to "Figure 3. Topography (A) and normalized channel steepness (ksn) (B) distribution".

**RESPONSE and CHANGES:**

We have changed it accordingly. Line 410.

**Thomas Bernard:**

Line 264-265: The black dashed curve shows the location of the main drainage divide". Change the term "curve" to "line".

**RESPONSE and CHANGES:**

We have changed it: "The black dashed line shows the location of the main drainage divide". Lines 412-413.

**Thomas Bernard:**

Line 272-273: "The topography swath profile along E-E' in Fig. 3A. (D) The ksn swath profile along F-F' in Fig. 3B". Change the terms "Fig. 3" to "panel".

**RESPONSE and CHANGES:**

We have changed the term "Fig. 3" to "panel". Line 422.

**Thomas Bernard:**

Line 273-274: "The red dotted line shows". Correct to "The red dotted lines show".

**RESPONSE and CHANGES:**

We have changed it accordingly. Line 424.

**Thomas Bernard:**

Line 279-292: Replace the terms "curve" by "line".

**RESPONSE and CHANGES:**

We have replaced it. Lines 453-468.

**Thomas Bernard:**

Line 292: Correct to "the channel-head (Eq. 4) and channel-head-segment (Eq. 8) methods respectively".

**RESPONSE and CHANGES:**

We have corrected it accordingly. Line 467.

**Thomas Bernard:**

Line 296: "(Yin, 2010 Su et al., 2021)". There is a missing coma between the two references.

**RESPONSE and CHANGES:**

We have added the coma here. Line 472.

**Thomas Bernard:**

Line 306: "the slope-area plots (Figs. 5 B, E, H) and the $\chi$ values". Change "and" by a coma.

**RESPONSE and CHANGES:**

We have changed it. Line 481.

**Thomas Bernard:**

Line 324-334: Replace the terms "curve" by "line".

**RESPONSE and CHANGES:**

We have replaced it. Lines 507-513.

**Thomas Bernard:**

Line 346-347: "In the tectonically active area". Change "the" by "a".

**RESPONSE and CHANGES:**

We have changed it. Line 528.

**Thomas Bernard:**

Line 350-351: Zhou et al. (2022a) combined the advantages of the $\chi$ and Gilbert metrics methods, proposed to use the $\chi$ contrast with a high base level". This sentence does not sound right. Maybe change "combined" by "by combining".

**RESPONSE and CHANGES:**

We have changed it: "Combining the advantages of the $\chi$ and Gilbert metrics methods, Zhou et al. (2022a) proposed to use the $\chi$ contrast with a high base level…". Lines 531-533.

**Thomas Bernard:**

Line 361: "(Forte and Whipple, 2018; Zhou et al., 2022a; this study)". Remove "this study" to the references.

**RESPONSE and CHANGES:**

We have removed it. Line 544.

**Thomas Bernard:**

Line 363-366: "In this study, we advocate the use of high-resolution DEM to determine a more accurate position and related parameters of the channel head, given that the use of UAVs to obtain the local DEM has become highly efficient". I suggest to break this sentence in two (maybe around the coma).

**RESPONSE and CHANGES:**

We have broken the sentence in two: "In this study, we advocate the use of high-resolution DEM to determine a more accurate position and related parameters of the channel head. The use of UAVs to obtain the local DEM has become highly efficient". Lines 550-552.

**Thomas Bernard:**

Line 368-369: "one can obtain the sub-meter resolution topography data of drainage divides". Replace "one can" by "it is possible to".

**RESPONSE and CHANGES:**

We have replaced it: "it is possible to obtain the high-resolution topography data of drainage divides …". Lines 556-557.

**Thomas Bernard:**

Line 396-397: Change "Consider an extreme example: when the main drainage divide" by "If we consider an extreme example where the main drainage divide".

**RESPONSE and CHANGES:**

We have changed it accordingly. Line 586.

**2. Response to Anonymous Referee #2:**

**Reviewer:**

This study addresses a highly pertinent question that lately has garnered significant attention in the scientific community focused on landscape evolution: what are the rates at which drainage divides migrate? The study presents two quantitative methods for measuring migration rates, both of which necessitate precisely identifying channel heads through high-resolution DEMs. The first method leverages the channel head's area and slope, while the second method builds upon an enhanced version of Zhou et al.'s (2022) technique, incorporating χ values, elevation at the channel head, and outlet elevation. The study applies these two methods in two distinct field settings—one impacted by tectonic tilting and the other situated in a tectonically quiescent area. The results reveal a similarity between the outcomes of both methods, suggesting that in the tectonically affected area, the divide is migrating in rates of ~0.1 mm/yr, while in the second setting, the divide is stable.

The foundational concept of the study seems promising, and the paper's structure effectively guides the reader through the core idea. Nevertheless, there are several crucial issues that demand the authors' attention, particularly regarding the methods presented in this manuscript. Hereafter I highlight specific points that warrant further clarification and elaboration.

**Overall comments:**

1. In the introduction, I recommend the following improvements: A) Provide a more in-depth motivation regarding the significance of divide migration, emphasizing its relevance to different studies. B) Distinguish between field-based investigations involving natural cases, such as those using cosmogenic nuclides and thermochronology, and modeling-based approaches, which use χ or Gilbert methods.

C) Expand upon the method introduced by Zhou et al. (2022), elucidating how it forms the foundation of your research.

**RESPONSE and CHANGES:**

We have revised the introduction accordingly:

A) We have added these sentences in the introduction to demonstrate the significance of divide migration and its relevance to different studies: "Drainage-divide migration, one form of river lateral movement, may not only carry information on geological and/or climatic disturbance (Su et al., 2020; Zondervan et al., 2020; He et al., 2021; Shi et al., 2021; Zhou et al., 2022a; Zeng and Tan, 2023) but also influence the extraction of tectonic information from channel profiles (Goren et al., 2014; Ma et al., 2020; Jiao et al., 2021). Moreover, it has multi-facet consequences for landscape evolution (Scheingross et al., 2020; Stokes et al., 2022), sedimentary process (Clift & Blusztajn, 2005; Willett et al., 2018; Deng et al., 2020; Zhao et al., 2021), and biological evolution (Waters et al., 2001; Zemlak et al., 2008; Hoorn et al., 2010; Musher et al., 2022). For this reason, the stability of drainage divides has drawn more and more attention in recent years (e.g., Authemayou et al., 2018; Vacherat et al., 2018; Chen et al., 2021; Shelef and Goren, 2021; Sakashita and Endo, 2023; Bian et al., 2024)." Lines 77-90.

B) We have distinguished the field-based investigations involving natural cases (e.g., cosmogenic nuclides sampling) and modeling-based approaches (e.g., χ or Gilbert metrics), and describe them in two separate paragraphs in the introduction.

C) We have added these sentences in the introduction to demonstrate the relationship between this study and Zhou et al. (2022): "Zhou et al. (2022a) developed a technique to calculate the migration rate through the high base-level χ values on both sides of a drainage divide. These new approaches require channel-head parameters to calculate the migration rate. However, the location of the channel heads sometimes cannot be accurately identified, because of the limitation in the resolution of DEMs in natural

cases. For this reason, empirical values of channel-head parameters are used in these study, which may induce uncertainties". Lines 119-126.

**Reviewer:**

2. The terminology "channel head method" and "channel head segment method" might confuse readers. I recommend considering alternative names for one of these methods.

**RESPONSE and CHANGES:**

We have changed the terminology "channel-head method" to "channel-head-point method" to avoid confusion. Line 158.

**Reviewer:**

3. The channel head used in equations 5-7 defines a point that separates the hillslope from the channel. However, the calculation of slope inherently involves two points. Therefore, it is essential to elucidate how you measured the slope in this specific context, particularly within the channel head, where it distinguishes between areas with differing slopes.

**RESPONSE and CHANGES:**

The $S_{ch}$ in equations 5-7 are measured along the tangent lines of the channel-head points on the channels. In the manuscript, we first plot the river long profiles and locate the channel-head points on the profiles. Then, we calculate the average slopes of the channel segments near the channel-head points (100m-long segments are enough), and use them as the $S_{ch}$ values. We have clarified it in the manuscript. Lines 371-373.

**Reviewer:**

4. I think it is crucial to provide a clear mathematical definition of parameters $\chi$ and ksn which are employed consistently throughout the study. Additionally, an explicit explanation of how Equation 8 was derived would be valuable. I am asking because ksn pertains to a specific point in the channel while $\chi$ represents an integral over a channel segment.

**RESPONSE and CHANGES:**

A) We have provided a clear definition of $\chi$ and $k_{sn}$ in the manuscript: "$k_{\text{sn}}$ is a widely used index (Whipple et al., 1999; Wobus et al., 2006; Hilley and Arrowsmith, 2008; Kirby and Whipple, 2012) that is quantitatively related to $E$ and $K$ ($k_{sn} = \left(\frac{E}{K}\right)^{\frac{1}{n}}$). $\chi$ is an integral function ($\chi = \int_{x_b}^{x} \left(\frac{A_0}{A(x)}\right)^{\frac{m}{n}} dx$) of a channel's upstream area ($A$) to horizontal distance ($x$) (Royden et al., 2000; Perron and Royden, 2012), and $A_0$ is an arbitrary scaling area to make the integrand dimensionless." Lines 241-247.

B) We have also provided the derivation of Equation 8 in the supplementary materials: "According to the detachment-limited stream power model (Howard and Kerby, 1983; Howard, 1994):

$$E = KA^m S^n \tag{1}$$

The channel gradient ($S$) can be written as:

$$S = \left(\frac{E}{K}\right)^{\frac{1}{n}} A^{-\left(\frac{m}{n}\right)} \tag{2}$$

A river's longitudinal elevation $z(x)$ can be expressed by the integration of channel gradient ($S$) in the upstream direction from a base level $x_b$ to an observation point $x$:

$$z(x) = z_b + \int_{x_b}^{x} S(x)\, dx = z_b + \int_{x_b}^{x} \left(\frac{E(x)}{K(x)}\right)^{\frac{1}{n}} A(x)^{-\left(\frac{m}{n}\right)} dx \tag{3}$$

In the case of spatially invariant erosion rate ($E$) and erosion coefficient ($K$), Eq. (3) can be reduced to a simpler form:

$$z(x) = z_b + k_{sn}(A_0)^{-\left(\frac{m}{n}\right)} \chi \tag{4}$$

with

$$k_{sn} = \left(\frac{E}{K}\right)^{\frac{1}{n}} \tag{5}$$

and

$$\chi = \int_{x_b}^{x} \left(\frac{A_0}{A(x)}\right)^{\frac{m}{n}} dx \tag{6}$$

$A_0$ is an arbitrary scaling area, to make the integrand dimensionless. Assuming $A_0 =$ 1m$^2$, the steepness ($k_{sn}$) of channel-head-segment can be written as:

$$k_{sn} = \frac{z_{ch} - z_b}{\chi} \tag{7}$$

According to the Eq. (5), the erosion rate ($E$) can be written as:

$$E = K k_{sn}{}^n = K\left(\frac{z - z_b}{(A_0)^{-\left(\frac{m}{n}\right)}\chi}\right)^n. \tag{8}$$

According to the equation of drainage-divide migration rate ($D_{mr}$):

$$D_{mr} = \frac{\Delta E_{ch} - \Delta U_{ch}}{tan\alpha + tan\beta} = \frac{E_{ch(\alpha)} - E_{ch(\beta)} - \Delta U_{ch}}{tan\alpha + tan\beta} \tag{9}$$

Combing Eq. (7, 8 & 9), the drainage-divide migration rate ($D_{mr}$) can be written as:

$$D_{mr} = \frac{E_{ch(\alpha)} - E_{ch(\beta)} - \Delta U_{ch}}{tan\alpha + tan\beta} = \frac{K k_{sn(\alpha)}{}^n - K k_{sn(\beta)}{}^n - \Delta U_{ch}}{tan\alpha + tan\beta} = \frac{K\left\{\left[\frac{(z_{ch} - z_b)_\alpha}{\chi_\alpha}\right]^n - \left[\frac{(z_{ch} - z_b)_\beta}{\chi_\beta}\right]^n\right\} - \Delta U_{ch}}{tan\alpha + tan\beta}$$

(10)".

**Reviewer:**

5. The calculation of the regional erosion coefficient K in this study requires some further clarification and consideration of several points: A) in the case of the Wutai Shan area, it is assumed that rock erodibility and precipitation are uniform. How is this assumption valid in a landscape with a range of elevation ranging between ~100-3000 m? Consider including isohyet maps and geological maps to strengthen your argument.  B) You mention that the erosion rate you used in the study averages a long

time frame (since the late Cenozoic). However, the ksn values, that are also used to calculate the erosion coefficient K, are based on the present conditions of lithology and precipitation. Do you think the ksn values also remained uniform since the late Cenozoic? Please explain the feasibility of these assumptions in the text. C) It is unclear why kriging interpolation is required over all of the landscape, as Ksn values are relevant only across channels. D) One value of erosion rate was considered for calculating the K of all the Wutai Shan area. However, the erosion rates along a tilting block might vary and probably decrease with distance from the northern Wutai Shan fault.

**RESPONSE and CHANGES:**

A) We have included the precipitation and geological maps in the supplementary materials (See the figures below) to support this assumption.

B) We agree that the erosion rate estimated from low-temperature thermochronology data may not represent the present-day situation. Therefore, we also refer to the results of geological studies and added the error range to cover this uncertainty: "Previous geological study shows that the Quaternary throw rates of the Northern Wutai Shan fault are 0.8-1.6 mm/a (Middleton et al., 2017). The low-temperature thermochronology study shows that the time-averaged long-term throw rates in the late Cenozoic is about 0.25 mm/yr, and there is an accelerated activity in the Wutai Shan area (Clinkscales et al., 2020). According to these studies, we assume a 0.50±0.25 mm/yr uplift/erosion rate in the northern margin of the Wutai Shan (in the footwall of the Northern Wutai Shan fault)". Lines 336-346. The error of the erosion rate also be calculated into the migration rates (see the revised Table 1 below). We have also added a "Limitations and uncertainties" section in the manuscript to report this error: "Second, in the case of the Wutai Shan, we refer to the geological and low-temperature thermochronology studies, and assume a 0.50±0.25 mm/yr erosion rate at the northern margin of the Wutai Shan (i.e., the footwall of the North Wutai Shan fault). Combining with the present-day ksn, we calculate the erosion coefficient (K) and derive the migration rates of the drainage divide. If the present-day erosion rate

deviates from the assumption, errors would be inevitable in the results". Line 617-623.

C) Indeed, the ksn values are only for channels, but the interpolation result can show the overall spatial variation trend.

D) Yes, the erosion rates should decrease with distance from the northern Wutai Shan fault, as well as the ksn values. But the K should be constant within the Wutai Shan area due to the similar precipitation and erodibility. Theoretically, the K value could be derived using the known erosion rate and its corresponding ksn of any place within the Wutai Shan area, and the calculated results should be the same.

[Figure]

Figure S2. Geological map of the natural example, Wutai Shan. The map is revised according to Clinkscales et al. (2020). The northern catchment evolves mainly through the undifferentiated Neoarchean-Paleoproterozoic metamorphic and igneous basement rocks (pЄu) and Neoarchean granitoids (Xg), whereas the southern catchment evolves mainly through the Paleoproterozoic Hutuo Group (Xh). There is no significant difference in rock erodibility on both sides of the drainage divide.

[Figure]

Figure S3. Precipitation distribution of the Wutai Shan. There is no significant difference in precipitation on both sides of the drainage divide, based on the precipitation data (1970-2000) from http://worldclim.org.

**Table 1.** Channel parameters and the migration rates of the drainage divides in the two field cases.

| Natural Cases | No. | $A_{cr}$ ($\times 10^5 m^2$) | $S_{ch}$ | $z_b$ (m) | $z_{ch}$ (m) | $\chi$ | $\tan\alpha$ | $\tan\beta$ | $\Delta U_{ch}$ (mm/yr) | $D_{mr}$ (mm/yr) (Channel-head-point method) | $D_{mr}$ (mm/yr) (Channel-head-segment method) |
|---|---|---|---|---|---|---|---|---|---|---|---|
| | Fig. 4 I$_\alpha$ | 1.75 | 0.16 | 1631 | 1792 | 6.4 | 0.14 | 0.66 | ~ 0.008 | -0.21 ±0.10 | -0.26 ±0.12 |
| | Fig. 4 I$_\beta$ | 0.26 | 0.63 | 1347 | 1723 | 6.6 | | | | | |
| Wutai Shan | Fig. 4 IIα | 0.79 | 0.23 | 1630 | 1815 | 5.4 | 0.24 | 0.70 | ~ 0.008 | -0.23 ±0.11 | -0.27 ±0.13 |
| | Fig. 4 IIβ | 0.30 | 0.67 | 1351 | 1809 | 6.1 | | | | | |
| | Fig. 4 III$_\alpha$ | 0.67 | 0.29 | 1633 | 1860 | 5.0 | 0.28 | 0.65 | ~ 0.008 | -0.21 ±0.10 | -0.22 ±0.10 |
| | Fig. 4 III$_\beta$ | 0.39 | 0.63 | 1352 | 1875 | 6.9 | | | | | |
| Yingwang Shan | Fig. 5 I$_\alpha$ | 0.54 | 0.21 | 1111 | 1224 | 5.8 | 0.21 | 0.31 | 0 | ~ 0.03 | ~ -0.01 |
| | Fig. 5 I$_\beta$ | 0.20 | 0.32 | 1126 | 1225 | 5.0 | | | | | |
| | Fig. 5 II$_\alpha$ | 0.24 | 0.36 | 1111 | 1257 | 7.4 | 0.39 | 0.33 | 0 | ~ 0.02 | ~ -0.01 |
| | Fig. 5 II$_\beta$ | 0.30 | 0.31 | 1117 | 1224 | 5.4 | | | | | |

| | | | | | | | | | | |
|---|---|---|---|---|---|---|---|---|---|---|
| Fig. 5 III$_\alpha$ | 0.29 | 0.46 | 1089 | 1256 | 8.6 | | | | | |
| | | | | | | 0.49 | 0.35 | 0 | ~ 0.02 | ~ -0.01 |
| Fig. 5 III$_\beta$ | 0.56 | 0.37 | 1096 | 1203 | 5.3 | | | | | |

**Reviewer:**

6. Errors and Uncertainties: Including error estimates and uncertainties in the study is crucial. These include horizontal and vertical errors of the DEM data and uncertainties related to slope measurements, erosion rates, and ksn Please explain how these errors propagate into the migration rate calculations and discuss their implications for the study's findings.

**RESPONSE and CHANGES:**

We have added the "Limitations and uncertainties" section in the manuscript and explained the above errors: "Moreover, the horizontal and vertical errors of the DEM data as well as the calculation errors in slope, upstream area and channel steepness can also affect the reliability of the results. In the case study of the Yingwang Shan, the lush vegetation may bring errors to the DEM data based on the SfM technology. The application of airborne light detection and ranging (LiDAR) technology may help reduce this error." Lines 623-628. The details of the erosion rates' error are in the RESPONSE of Overall comments 5.

**Reviewer:**

7. Accessibility and resolution of the data: Please provide the precise resolution of the DEM and not only a general "sub-meter" description. If possible, include these DEM datasets as supplementary materials or explain any limitations or reasons why they cannot be.

**RESPONSE and CHANGES:**

We have provided the resolution of the DEM in the manuscript (0.67 m and 0.84 m spatial resolution in the Wutai Shan and the Yingwang Shan respectively), and include the DEM datasets as supplementary materials. Line 279.

**Reviewer:**

8. Maybe evaluate the relevance of Section 4.1 and consider incorporating it into the introduction. Currently, in my opinion, it fits to the introduction, as it provides motivation for the study rather than introducing new information.

**RESPONSE and CHANGES:**

We have revised the introduction according to the overall comments 1 (see the response there) and clarified the motivation of the study. Therefore, we retained Section 4.1 and revised it.

**Reviewer:**

9. Can you identify morphological evidence from the field that supports the inference of drainage divide migration? For instance, wind gaps along the divide or evidence for stream capture? Such evidence can meaningfully support your results.

**RESPONSE and CHANGES:**

Thank you for your suggestion. We have added the evidence for stream capture in the Wutai Shan area (Fig. 3 in the new manuscript): "Geomorphic evidence also exhibits a northwestward migration of the drainage divide (Fig. 3). The plan and satellite views show several barbed tributaries and a captured area around the Wutai Shan drainage divide, which indicate that the tributaries formerly part of the north drainage became part of the south drainage (Fig. 3A&B). The $\chi$-plots analysis shows the south side of the drainage divide has steeper channels, higher $k_{sn}$, and lower $\chi$. The $\chi$-plots

of paired rivers show obvious characteristics of shrinking-expanding and captured-beheaded rivers (Fig. 3C)." Lines 319-327.

[Figure]

**Figure 3**. Perspective views and χ map of the drainage divide in the Wutai Shan (The location is shown in Fig. 2). (A) Perspective views of the capture area and the channels mapped with ksn. The south side of the drainage divide has steeper channels and higher ksn than the north side. Red arrows show drainage divide migration directions. The satellite image is downloaded from Google Earth. (B) The χ Map in this area with the outlet elevation of 1300 m. The south side of the drainage divide has lower χ values than the north side. The χ-plots of the rivers in bold lines are shown in panel C. The topography data (ALOS DEM, 12.5 m resolution) is downloaded from the Alaska Satellite Facility (ASF) Data Search (https://search.asf.alaska.edu/). (C) χ-plots of the three paired rivers in panel B. The blue and red curves are the rivers on the south and north sides respectively. The χ-plot of River 1 is steeper on the south side, indicating that the river on the south side is the aggressor and is the victim on the

north side. The χ-plots of Rivers 2&3 in the captured area show obvious characteristics of the captured and beheaded rivers. The χ-plot is extracted using TAK (Forte and Whipple, 2019) and TopoToolbox (Schwanghart and Scherler, 2014).

**Reviewer:**

10. I provided some suggestions for edits that can assist in the flow and structure of the manuscript, such as connecting too short sentences e.g. lines 293, repetitive sentences, e.g., lines 54 and 132, or unclear sentences such as 251-252. The sentences have been revised accordingly. See the responses below.

**Line-by-line comments:**

20: "normalized channel steepness"- I don't think this term can be used in the abstract without explanation.

**RESPONSE and CHANGES:**

We have removed this term in the abstract and provided a detailed explanation in the methods section. Line 31.

**Reviewer:**

22: Maybe give these mountains a name of your own, as you refer to them several times in the manuscript. Same for line 86 and further in the manuscript.

**RESPONSE and CHANGES:**

We have named the mountain "Yingwang Shan" and changed the manuscript accordingly. Line 35.

**Reviewer:**

23-25: I suggest connecting these two short sentences to one sentence, e.g., "Our results find that the divide in Wutai Shan range is migrating north in a rate of between 0.10 to 0.13 mm/yr, whereas the migration rates at the mountain range in the Loess Plateau are approximately zero."

**RESPONSE and CHANGES:**

We have connected the two sentences to one accordingly. "Our results show that the Wutai Shan drainage divide is migrating northwestward at a rate between 0.21 to 0.27 mm/yr, whereas the migration rates at the Yingwang Shan are approximately zero." Lines 37-40.

**Reviewer:**

25-27: How is this demonstrated?

**RESPONSE and CHANGES:**

We have changed this sentence to "This study indicates that the drainage-divide migration state can be determined more accurately using high-resolution topographic data. Furthermore, this study takes the cross-divide differences in uplift rate of channel heads into account in the measurement of drainage-divide migration rate for the first time". Lines 42-48.

**Reviewer:**

34: "…providing a basis.."

**RESPONSE and CHANGES:**

We have changed it: "The evolution of the Earth's surface is jointly controlled by tectonics, lithology, and climatic conditions (e.g., Molnar and England, 1990; Whipple, 2009; Gallen, 2018; Bernard et al., 2021; Hoskins et al., 2023), providing a basis for reconstructing …". Lines 53-56.

**Reviewer:**

37: What is the difference between the evolution of earth's surface and the evolution of topography?

**RESPONSE and CHANGES:**

We have changed "the evolution of topography" to "the evolution of unglaciated terrestrial terrains" to make the sentence clearer. Line 59.

**Reviewer:**

38-39: The sentences imply that there are other changes in drainage systems that are not included in the river's vertical or lateral movements. If yes, what are they?

**RESPONSE and CHANGES:**

We have changed "including" to "which are" to be more precise: "The evolution of unglaciated terrestrial terrains is fundamentally coupled with changes in drainage systems, which are river's vertical (changes in river long profile) and lateral movements (drainage divide migration and river captures) (Whipple, 2001; Clark et al., 2004; Bonnet, 2009; Willett et al., 2014)". Lines 59-63.

**Reviewer:**

40-43: Maybe write shortly about the findings of all these investigations.

**RESPONSE and CHANGES:**

We have added the findings of all these investigations in the manuscript: "River's long profiles have been used to study the earthquake events (e.g., Burbank and Anderson, 2001; Wei et al., 2015) and the spatio-temporal variations of uplift (e.g.,

Whipple et al., 1999; Kirby et al., 2003; Pritchard et al., 2009; Goren et al., 2014).".

Lines 69-72.

**Reviewer:**

44: Why however? i.e., how does this sentence contrast with the previous one?

**RESPONSE and CHANGES:**

We have deleted "however". Line 73.

**Reviewer:**

47: I think you must define here what is drainage divide migration.

**RESPONSE and CHANGES:**

We have added the explanation here: "Drainage-divide migration, one form of lateral movement of rivers, may…". Line 77.

**Reviewer:**

47-51: It is unclear why and how the divide migration carries information about the disturbances and the tectonic information extraction.

**RESPONSE and CHANGES:**

We added the discription on the relationship between the drainage divide migration and the geological and/or climatic disturbance: "Recent studies show that the widespread lateral movement of river basins driven by geological and/or climatic disturbance (Yang et al., 2019; Zondervan et al., 2020; Zhou et al., 2022a; Bian et al., 2024)." Lines 73-76.

**Reviewer:**

55: The migration is also controlled by the topographic slope in the citations you provided.

**RESPONSE and CHANGES:**

Yes, we have changed the sentence to "Drainage-divide migration is essentially controlled by the cross-divide difference in erosion and topographic slope". Line 92.

**Reviewer:**

59: In what cases do you use migration velocity, and when do you use migration rate? Same for line 79, 145 and further in the manuscript.

**RESPONSE and CHANGES:**

The migration velocity including the migration rate and direction. The migration rate is calculated through the formulas in the manuscript, and the migration direction is determined through the strike of the drainage divide. This study focused on how to constrain the migration rate. We have replaced the "migration velocity" by "migration rate". Lines 96-97

**Reviewer:**

63: Do you mean "..of sample processing"?

**RESPONSE and CHANGES:**

Yes, we have changed it in the manuscript. Line 102.

**Reviewer:**

65 and 66: Perhaps replace "entire landscape" with "large landscapes" or alike. It is not clear to which entire landscape you are referring to.

**RESPONSE and CHANGES:**

Yes, we have replaced it in the manuscript: "Besides, the high cost of sample processing makes it challenging to determine the drainage divide's motion by measuring the erosion rates throughout the large landscapes". Lines 101-103.

**Reviewer:**

68: The term "remote sensing" includes a very wide range of data types. Maybe be more specific.

**RESPONSE and CHANGES:**

We have changed the sentence to make it more specific: "The advancement of the digital elevation model (DEM) has promoted the development of geomorphic analysis …". Lines 106-107.

**Reviewer:**

69-70: What do you mean by "determine the drainage divide's motion through topography analysis "? determine the past motion (i.e., infer divide migration that occurred in the past)? Or predict future motion?

**RESPONSE and CHANGES:**

We mean the transient motion reflected by the DEM data. We have noted it in the manuscript: "… making it possible to determine the drainage divide's transient motion through topography analysis". Lines 107-108.

**Reviewer:**

71: See overall comments- you should explain more about the χ parameter and devote an equation to this, especially if you use it in your equations later.

**RESPONSE and CHANGES:**

Yes, we have explained more about the χ parameter and added the equation (see the response of overall comment 4).

**Reviewer:**

73: What do you mean in determine a drainage divide motion? The direction? Rate?

**RESPONSE and CHANGES:**

We mean the migration direction here. We have clarified it in the manuscript. Line 112.

**Reviewer:**

74: What is the difference between slope angle and the slope mentioned in the previous sentence?

**RESPONSE and CHANGES:**

The "slope" is the tangent value of "slope angle" here.

**Reviewer:**

75-77: The fact these methods are quantitative does not necessarily mean they can provide rates. No need to add the contrast term "although" in the beginning of the sentence.

**RESPONSE and CHANGES:**

We have removed the word "although" from the sentence. Line 115.

**Reviewer:**

78: Please explain better about this "method". I assume you mean he used numerical simulations. Also, as I wrote in the overall comments, I recommend separating between modelling-based studies and all of the other studies you cite, that are field based.

**RESPONSE and CHANGES:**

Besides the numerical simulations, the study also provides a formula. We have explained it in the manuscript.

**Reviewer:**

81-80: Explain "cross-divide χ ratio". Ratio between what and what?

**RESPONSE and CHANGES:**

It is the ratio of χ values on both sides of a drainage divide $(\frac{\chi_\alpha}{\chi_\beta})$. We have revised this sentence.  Lines 120-121.

**Reviewer:**

81-84: This sentence doesn't make sense to me. First, something is missing here- you need to mention that the technique of Zhou et al 2022 requires channel head parameters before saying that these parameters are problematic. Second, what are the previous studies you are talking about? Are you referring to a case-specific study? and why this parameter is not applicable in specific natural areas?

**RESPONSE and CHANGES:**

We have revised these sentences: "Braun (2018) provided an equation that considers both alluvial and fluvial areas to calculate the migration velocity of an escarpment

(also a drainage divide). Zhou et al. (2022a) developed a technique to calculate the migration rate through the high base-level χ values on both sides of a drainage divide. These new approaches require channel-head parameters to calculate the migration rate. However, the location of the channel heads sometimes cannot be accurately identified, because of the limitation in by the resolution of DEM in natural cases. For this reason, empirical values of channel-head parameters are used in these studies, which may induce uncertainties". Lines 117-126.

**Reviewer:**

85: Before this sentence, I suggest providing the knowledge gap and saying what problem you are addressing (e.g., "Currently, there is no method for quantifying divide migration rates across large landscapes.. ").

**RESPONSE and CHANGES:**

We have added a sentence here and provided the knowledge gap: "… empirical values of channel-head parameters are used in these studies, which may induce uncertainties. This study aims to establish an approach to derive the migration rate of drainage divides, at a high precision and low cost, based on topographic analysis". Lines 125-131.

**Reviewer:**

91-93: When first reading this, it is unclear how you calculated the migration rates from the parameters. Perhaps write that you developed two methods for this and that one is based on the method of Zhou 2022.

**RESPONSE and CHANGES:**

We have revised this sentence: "We then develop two methods to calculate the drainage-divide migration rates. One is based on the measured channel-head

parameters, and the other is based on an improved method of Zhou et al (2022a)".
Lines 140-143.

**Reviewer:**

93: When you mentioned the method of Zhou et al. 2022, you didn't say that this method doesn't apply to cases where the elevations of outlets and channel heads are different across the divide (isn't this almost always the case?).

**RESPONSE and CHANGES:**

In natural cases, the elevations of channel heads are usually different. However, limited by the resolution of DEM, the location of the channel heads sometimes cannot be accurately identified. Therefore, the channel head parameters across the divide for calculating the migration rates are based on the same empirical values in Zhou et al. (2022a), which may induce uncertainties. We have clarified it in Lines 119-126.

**Reviewer:**

96: This sounds like a good sentence to open the paragraph.

**RESPONSE and CHANGES:**

We have moved this sentence to the beginning of this paragraph. Line 130.

**Reviewer:**

98: Maybe delete "moreover."

**RESPONSE and CHANGES:**

We have deleted it. Line 150.

**Reviewer:**

99: "Quantify the influence" is not clear to me. Do you mean Quantify the cross-divide difference in uplift rates?

**RESPONSE and CHANGES:**

We have changed the phrase to "quantify the cross-divide difference in uplift rates to improve the precision of drainage-divide migration rate". Lines 153-155.

**Reviewer:**

109-111: Landslide threshold and erosion threshold are not mechanisms- they are thresholds. If you mention them, you should explain what these thresholds are, and why do they cause the channels to emerge at a certain distance from the divide.

**RESPONSE and CHANGES:**

We have changed the word "mechanisms" to "thresholds" and added explanations of the thresholds: "Because of thresholds such as erosion threshold (the shear stress of overland flow must exceed the threshold of the cohesion of bed material to generate river incision) (Howard and Kerby, 1983; Perron et al., 2008) or landslide threshold (landslides occur when the threshold of soil or rock strength is exceeded in high relief region) (Burbank et al., 1996; Tucker et al., 1998), river channels (following Eq. 1) emerge at a certain distance from the drainage divide". Lines 164-169.

**Reviewer:**

121: Slope is measured between two points. What are these two points in this case? Do you measure only from the channel head and downstream (because the channel head is the highest point of the fluvial segment)

**RESPONSE and CHANGES:**

We first plot the river long profiles and locate the channel-head points on the profiles. Then, we calculate the average slopes of the channel segments near the channel-head points (100m-long segments are enough) and use them as the tangent slopes of the channel-head points. We have revised this sentence to "$S_{ch}$ is the channel-head gradient measured along the channel near the channel-head point." Line 178.

**Reviewer:**

122-131, including figure 1: I think it is obvious that increasing S, A, and K, or decreasing the m/n ratio, will result in higher erosion rates. Do I miss something here? Please clarify or eliminate this part+ this figure.

**RESPONSE and CHANGES:**

We have removed this part and added the sentence: "Eq. 2 indicates that the side of the drainage divide with a higher $A_{cr}$ or $S_{ch}$ can have a higher erosion rate than the other side, and is more likely to pirate the opposite drainage basin. Besides, a high erosion coefficient can amplify the drainage basin's erosion rate". Lines 182-188. We also moved Figure 1 to the supplementary materials.

132: Repetitive with the sentence in line 54, see comment there.

**RESPONSE and CHANGES:**

We have also changed this sentence: "Drainage-divide migration is essentially controlled by the cross-divide difference in erosion rates and topographic slope (Beeson et al., 2017; Dahlquist et al., 2018; Chen et al., 2021; Zhou et al., 2022a; Stokes et al., 2022)". Lines 190-192.

**Reviewer:**

139: Eq 3 suggests that migration is also a function of the gradients of the slopes across the divide.

**RESPONSE and CHANGES:**

We have added this information in the sentence. Line 198.

**Reviewer:**

142: I think you can delete "rock."

**RESPONSE and CHANGES:**

We have deleted it. Line 205.

**Reviewer:**

147: I suggest moving this sentence to line 141 where α and β are first presented.

**RESPONSE and CHANGES:**

We have moved this sentence there. Lines 202-204.

**Reviewer:**

157: Add citation for measuring erosion rates from cosmogenic nuclides

**RESPONSE and CHANGES:**

We have added the citation here: "Eα and Eβ are the erosion rates of the α to β side of drainage divide, respectively, which can be derived through the cosmogenic nuclides (10Be) concentration measurements (Beeson et al., 2017; Godard et al., 2019; Hu et al., 2021)". Lines 220-223.

**Reviewer:**

157: Please clarify what is the regional average erosion rate. Do you mean (Eα + Eβ)/2 ?

**RESPONSE and CHANGES:**

Yes, it is. We have clarified it in the manuscript: "The regional average erosion rate ($\bar{E} = \frac{E_\alpha + E_\beta}{2}$) can also be used to calculate the migration rate". Line 223.

**Reviewer:**

162: It can help to add the symbol, i.e., "erosion coefficient (K).." etc.

**RESPONSE and CHANGES:**

We have added the symbols here: "… combined with one of the erosion-related parameters, erosion coefficient ($K$), erosion rate at one side of a drainage divide ($E_\alpha$ or $E_\beta$), or regional average erosion rate ($\bar{E}$)". Lines 228-229.

**Reviewer:**

173: What do mean by channel head segments? The segment between the divide and the channel head? If yes, better define in the text.

**RESPONSE and CHANGES:**

We have added the definition in the manuscript: "A channel-head segment is the channel segment just below the channel head". Lines 237-238.

**Reviewer:**

174-176. About Ksn -See comment to line 71 and general comments.

**RESPONSE and CHANGES:**

We have provided a clear definition of ksn in the manuscript (see the RESPONSE of general comments 4).

**Reviewer:**

This is the missing information for the comment in line 84. Nonetheless, I don't understand how the channel head and the outlet of two channels across the divide can be the same elevation in natural settings.

**RESPONSE and CHANGES:**

The elevation of the two channel heads across a divide is usually different in natural settings. But limited by the resolution of DEM, we sometimes cannot accurately identify the location of the channel heads. Therefore, we simplified the calculation according to an empiric value ($A_{cr} = 10^5$ m$^2$) in Zhou et al., 2022a. In this study, benefiting from the high-resolution DEM, we can accurately identify the location of channel heads and do not need the empiric value.

**Reviewer:**

183: need to explain about $\chi$ in the place it first appears (see comment on line 74)

**RESPONSE and CHANGES:**

We have explained more about the $\chi$ parameter and added the equation (see the RESPONSE of overall comment 4).

**Reviewer:**

199-206: This reads more as part of the methods, in particular lines 203-206.

**RESPONSE and CHANGES:**

We have moved the sentences to the Methods section. Lines 276-284.

**Reviewer:**

208, Fig 2: Beautiful map! Some notes:

-Please provide a source of information for the maps.

**RESPONSE and CHANGES:**

We have provided the source of information in the figure caption: "The topography data (ALOS DEM, 12.5 m resolution) is downloaded from the Alaska Satellite Facility (ASF) Data Search (https://search.asf.alaska.edu/). The fault data is downloaded from the site (https://www.activefault-datacenter.cn/). The satellite image is downloaded from Google Earth". Lines 302-306.

**Reviewer:**

-I suggest annotating each map differently (A and B), and refer to them in the captions accordingly.

**RESPONSE and CHANGES:**

We have revised it accordingly. Line 296.

**Reviewer:**

-Maybe add a legend for the faults. Are they all normal faults? If not, mark them with simple curves without the small lines that stick out.

**RESPONSE and CHANGES:**

We have added the legend for the faults and marked them with simple curves. Most of them are normal faults, with several thrust fault and strike-slip faults in the southwest corner. We have revised accordingly. See the revised version of the figure below.

[Figure]

**Reviewer:**

-How come the highest elevation in the map is exactly 5000 m (should be m asl-above sea level)? If this is not exactly the value, please fix to the correct max elevation in the map.

**RESPONSE and CHANGES:**

The highest elevation on the map is 3652m. We have fixed the range of the colour bar to the min-max elevation (3-3652m) and redrawn the map (See the revised figure in the RESPONSE above).

**Reviewer:**

214: Refer to fig 2.

**RESPONSE and CHANGES:**

We have referred to it here. Line 289.

**Reviewer:**

216: But figure 3A does show a fault in the south (Xizaou chen)? Do you mean that the tilting of the Wutai Shan is dominated mostly by the activity of the northern fault?

**RESPONSE and CHANGES:**

Yes, the tilting of the Wutai Shan is dominated by the activity of the Northern Wutai Shan fault. The Xizhou Shan fault in the south controls the tilting of the Xizhou Shan.

**Reviewer:**

218-219: As I wrote in the overall comment 5- how can you explain no variation in precipitation in an area with an elevation difference of ~2500 m?

**RESPONSE and CHANGES:**

We have provided the precipitation and geological maps in the supplementary materials (See the response of the overall comment 5).

**Reviewer:**

220-228: Maybe this part can fit better in the methods section. Aside from the comments in overall comment 5, the calculation of erosion coefficient here is not clear unless you provide the equation that relates between kns, k , and erosion rates.

**RESPONSE and CHANGES:**

We have provided the equation for calculating K in the manuscript: "Combing with the equation, $K = \dfrac{E}{k_{sn}{}^n}$, and following the approach of previous studies (Kirby and Whipple, 2001; Kirkpatrick et al., 2020; Ma et al., 2020), the erosion coefficient (K) is calculated to be $(6.25 \pm 3.13) \times 10^{-6}$ m$^{0.1}$yr$^{-1}$ in this area". Because this calculation is

only for the Wutai Shan area, we prefer to leave it in the case study section. Lines 347-350.

**Reviewer:**

224-225: Add a reference to fig 3. Also- is the erosion rate based on all five samples illustrated in figure 3?

**RESPONSE and CHANGES:**

We have added a reference to Fig. 3 here (Line 429). The low-temperature thermochronology result is based on all five samples. Besides, we have changed the estimation of erosion rate, considering both the geological and low-temperature thermochronology studies: "The geological study shows that the Quaternary throw rates of the Northern Wutai Shan fault are 0.8-1.6 mm/a (Middleton et al., 2017). The low-temperature thermochronology study shows that the time-averaged long-term throw rates in the late Cenozoic is about 0.25 mm/yr, and there is an accelerated activity in the Wutai Shan area (Clinkscales et al., 2020). According to these studies, we assume a 0.5±0.25 mm/yr uplift/erosion rate in the northern margin of the Wutai Shan (at the footwall of the Northern Wutai Shan fault)". Lines 338-346.

**Reviewer:**

230: What do you mean by "randomly"? maybe delete. Same for line 302

**RESPONSE and CHANGES:**

We have deleted the word "randomly". Lines 355&480.

**Reviewer:**

231: How did you measure the slope (see overall comment 3)? Was it extracted from the high-resolution DEM?

**RESPONSE and CHANGES:**

Yes, the slope is extracted from the high-resolution DEM using the TAK (Forte and Whipple, 2019) and TopoToolbox (Schwanghart and Scherler, 2014).

**Reviewer:**

232: You probably mean "the breaking point of the slope-area regression line? Was this point determined manually or by an automatic algorithm?

**RESPONSE and CHANGES:**

Yes, we have changed the sentence to "the breaking point of the slope-area regression line" (Line 359). This point is determined manually.

**Reviewer:**

237: What average- the average slope of all points measured between the channel head and the divide? Also- why do use only the points that are upstream from the channel head? They represent the hillslope section and not the fluvial segment of the channel.

**RESPONSE and CHANGES:**

We agree that the points that are upstream from the channel head represent the hillslope section. We have changed the way to calculate $S_{ch}$, which is calculated along the tangent lines of the channel-head points on the channels. The calculation process is in the RESPONSE of overall comment 3. Besides, the tanα and tanβ are calculated through the average slope of the hillslope segments near the channel head (not including the hilltop part).

**Reviewer:**

238-240: I'm not sure I understood from the text: zb and zch and the $\chi$ values were extracted from the $\chi$ plot? If yes, please rephrase for making the text more clear.

**RESPONSE and CHANGES:**

We have clarified it in the manuscript: "According to the location of the catchment outlets (Fig. 4A), we obtain the outlet elevations (zb) from the river's long profiles" (Lines 357-358), "For the $\chi$-plots (Figs. 4C, F, I), we obtain the elevations of channel heads (zch) and $\chi$ values based on the coordinate of the channel-head points" (Lines 368-369).

**Reviewer:**

243: "The normal direction of the boundary fault"? what does this mean? Parralal or perpendicular to the direction of the fault?

**RESPONSE and CHANGES:**

We mean the distance perpendicular to the direction of the boundary fault. We have changed the sentence in the manuscript. Lines 378-379.

**Reviewer:**

246: Are you following previous studies that generally used these values or specific studies that used similar values in your study area?

**RESPONSE and CHANGES:**

We followed previous studies that generally used these values. We have also added the results of m = 0.35 and 0.55 in the supplementary materials as references for other situations.

**Reviewer:**

251-252: The sentence "…show that distinct character of the rivers across the drainage divide" doesn't make sense to me. Please try a different phrasing.

**RESPONSE and CHANGES:**

We have changed this sentence to "The rivers have different characteristics on both sides of the drainage divide as illustrated on their slope-area plots (Figs. 4B, E, H) and the χ-plots (Figs. 4C, F, I)". Lines 388-390.

**Reviewer:**

259: Suggest adding "in this area" at the sentence's end.

**RESPONSE and CHANGES:**

We have added the phrase "in this area" at the sentence's end. Line 399.

**Reviewer:**

Fig 3: A. cite Clinkscales et al 2020 also in the legend. B. See comment on elevation scale in fig 2- same for the ksn scale here.

**RESPONSE and CHANGES:**

We have changed the figure accordingly (Line 408):

[Figure]

**Reviewer:**

265: Same comment about the faults in Fig 2.

**RESPONSE and CHANGES:**

They are all normal faults. We have added the dots on the fault symbols (see the revised figure above).

**Reviewer:**

267: Maybe write "Black rectangles show the area of Fig. 4A".

**RESPONSE and CHANGES:**

We have changed the sentence here accordingly.

**Reviewer:**

271: I think that "based on matlab" can be deleted.

**RESPONSE and CHANGES:**

We have deleted the phrase. Line 420.

**Reviewer:**

272: What software did you apply for the kringing method?

**RESPONSE and CHANGES:**

It is on ArcGIS. We have noted it in the manuscript. Line 421.

**Reviewer:**

273: Did you use topotoolbox also for the swath profile? if yes please mention it.

**RESPONSE and CHANGES:**

Yes, we use topotoolbox also for the swath profile. We have mentioned it in the manuscript. Line 423.

**Reviewer:**

273: Not dotted, dashed.

**RESPONSE and CHANGES:**

Yes, we have changed it. Line 424.

**Reviewer:**

275: I recommend showing the extent of the swath profile in figs A&B.

**RESPONSE and CHANGES:**

We have added the extent of the swath profile in the figure (see the RESPONSE of Fig. 3 for the revised version). Line 408.

**Reviewer:**

Suggestions to figs. 4 and 5. fig A: i) Specify "profiles" in the legend – I think you mean topographic profiles. ii) Add the arrows and the divide to the legend - maybe make the arrow length proportional to the migration rate. iii) Complete the divide line. iv) Maybe change the scale in the bottom left corner to white. v) I think that you can eliminate m=0.45 and n=1 in the middle columns. V) I am not sure it is right to continue the $\chi$ profile up to the divide (ie, upstream from the channel head), as this segment is not fluvial and is not applicable for $\chi$ analysis.

**RESPONSE and CHANGES:**

We have changed the figs. 4 & 5 accordingly (see the revised fig. 4 below):

i) We have changed "profiles" to "topographic profiles" in the figure.

ii) We have added the arrows and the divide to the legend and made the arrow length proportional to the migration rate.

iii) We have completed the divide line.

iv) We have changed the scale in the bottom left corner to white.

v) We have eliminated the m=0.45 and n=1 in the middle columns.

vi) We do not use the $\chi$ profile in the hillslope area (shown in dashed lines) for calculation. We plot them only to show the location of channel heads.

[Figure]

**Reviewer:**

280: Please explicitly specify the resolution. Also, this is not a topographic map but a hillshade map.

**RESPONSE and CHANGES:**

We have added the resolution information in the manuscript (0.67 m and 0.84 m spatial resolution in the Wutai Shan and the Yingwang Shan respectively). We also have changed the word "topographic" to "hillshade". Line 454.

**Reviewer:**

292: Should be "methods."

**RESPONSE and CHANGES:**

We have changed it to "methods". Line 467.

**Reviewer:**

294: Maybe give it a name of your own?

**RESPONSE and CHANGES:**

We have named it "Yingwang Shan". Line 470.

**Reviewer:**

296-7: Please check the grammar and correct accordingly.

**RESPONSE and CHANGES:**

We have corrected it: "Over the past 2.6 million years, it has accumulated tens to hundreds of meters of eolian sediments (Yan et al., 2014), draping preexisting topography (Xiong et al., 2014)". Lines 472-474.

**Reviewer:**

298: Maybe connect this short sentence to one of the sentences before or after it.

**RESPONSE and CHANGES:**

We have connected this sentence with the following sentence: "There is no active fault and is little to no variation in rock erodibility and precipitation within the area (Shi et al., 2020; Zhou et al., 2022b)". Lines 475-477.

**Reviewer:**

301-309: This subsection is repetitive with the previous one. Perhaps just write that you extracted the migration rates similar to Wutai Shan site.

**RESPONSE and CHANGES:**

We have simplified the sentences: "Similar to the Wutai Shan site, we make the slope-area plots (Figs. 5 B, E, H) and the $\chi$-plots (Figs. 5 C, F, I), and extract the values of Acr, Sch, zb, zch, $\chi$, tan$\alpha$ and tan$\beta$ of the rivers". Lines 479-487.

**Reviewer:**

311: Is this an average erosion rate? If yes, what timescale does it average? How was this rate extracted? Please elaborate on this.

**RESPONSE and CHANGES:**

The data can represent the present-day erosion rate because it is calculated through the distribution of the current river's silt discharge. We have clarified it in the manuscript: "The rate of soil erosion in the study area is about 500 t·km-2yr-1 according to the distribution of silt discharge (Fu, 1989). Combining with the assumption of the density of loess, 1.65 t ·m-3, the present-day erosion rate in the study area is calculated to be 0.3 mm yr$^{-1}$". Lines 488-491.

**Reviewer:**

337: Should be "in the two field cases."

**RESPONSE and CHANGES:**

We have changed it to "in the two field cases.". Line 518.

**Reviewer:**

Table 1: The lack of uncertainties here is substantial.

**RESPONSE and CHANGES:**

We have added the uncertainties in Table 1. See the RESPONSE to the overall comment 5 for details.

**Reviewer:**

349-350: These sentences are almost similar to those in line 72. As I wrote in overall comment 8, I think that all of this subsection should be merged with the introduction.

**RESPONSE and CHANGES:**

We have deleted this sentence to avoid repetition. We also revised the introduction section.

**Reviewer:**

355: "These methods"- do you refer to $\chi$, Gilbert methods, or Zhou's? Or all?

**RESPONSE and CHANGES:**

We have changed "these methods" to "the Gilbert metrics or χ-comparison method in Zhou et al. (2022a)" to avoid ambiguity. Lines 537-538.

**Reviewer:**

362: What are the empirical parameters on which the channel head location relies on? And what is the problem with them? Do you mean that the location depends on the break of the slope, which cannot be determined appropriately in DEMs with a coarse resolution?

**RESPONSE and CHANGES:**

Yes, the location of a channel head is hard to accurately identified with a coarse resolution of DEM. Therefore, the calculation in Zhou et al. (2022a) uses the empiric value ($A_{cr} = 10^5$ m$^2$), which may induce uncertainties. We have added more explanation in the manuscript. Lines 544-549.

**Reviewer:**

363: Perhaps: "which can induce uncertainties in determining the stability of drainage divides".

**RESPONSE and CHANGES:**

We have changed the sentence accordingly. Lines 546-549.

**Reviewer:**

382: I am confused- you are saying that the differential uplift rate should be ignored or taken into account?

**RESPONSE and CHANGES:**

It should be taken into account. We have deleted the sentence "Although the differential uplift rate is …" to avoid misunderstanding. Lines 571-573.

**Reviewer:**

394: You assume that divide migration rate decreases as the divide becomes closer to steady state. Please provide a reference to this argument. However, this also implies that the divide migration rate may change over long timescales. This might not be consistent with the parameters that were used for calculating k (Wutai Shan) or E (Loess plateau), which represent an average over long timescales.

**RESPONSE and CHANGES:**

The evidence is mainly from numerical simulation (Whipple et al., 2017; Ye et al., 2022), and we have added the reference here. We have also added the error range of the erosion rate in the Wutai Shan (see the RESPONSE to the overall comment 5). In the Loess Plateau, the drainage divide is stable and the data can represent the present-day erosion rate because it is calculated through the distribution of the current river's silt discharge.

[Figure]

Fig. 10 in Whipple et al. (2017)

[Figure]

Fig. 1 in Ye et al. (2022)

---

## Editor Decision (ED1)

[revised manuscript text omitted]
α | 1.75 | 0.16 | 1631 | 1792 | 6.4 | 0.14 | 0.66 | ~ 0.008 | -0.21±0.10 | -0.26±0.12 |
| | Fig. 4 Iβ | 0.26 | 0.63 | 1347 | 1723 | 6.6 | | | | | |
| | Fig. 4 IIα | 0.79 | 0.23 | 1630 | 1815 | 5.4 | 0.24 | 0.70 | ~ 0.008 | -0.23±0.11 | -0.27±0.13 |
| | Fig. 4 IIβ | 0.30 | 0.67 | 1351 | 1809 | 6.1 | | | | | |
| | Fig. 4 IIIα | 0.67 | 0.29 | 1633 | 1860 | 5.0 | 0.28 | 0.65 | ~ 0.008 | -0.21±0.10 | -0.22±0.10 |
| | Fig. 4 IIIβ | 0.39 | 0.63 | 1352 | 1875 | 6.9 | | | | | |
| Yingwang Shan | Fig. 5 Iα | 0.54 | 0.21 | 1111 | 1224 | 5.8 | 0.21 | 0.31 | 0 | ~ 0.03 | ~ -0.01 |
| | Fig. 5 Iβ | 0.20 | 0.32 | 1126 | 1225 | 5.0 | | | | | |
| | Fig. 5 IIα | 0.24 | 0.36 | 1111 | 1257 | 7.4 | 0.39 | 0.33 | 0 | ~ 0.02 | ~ -0.01 |
| | Fig. 5 IIβ | 0.30 | 0.31 | 1117 | 1224 | 5.4 | | | | | |
| | Fig. 5 IIIα | 0.29 | 0.46 | 1089 | 1256 | 8.6 | 0.49 | 0.35 | 0 | ~ 0.02 | ~ -0.01 |
| | Fig. 5 IIIβ | 0.56 | 0.37 | 1096 | 1203 | 5.3 | | | | | |

[revised manuscript text omitted]

---

## Author Response (AR2)

**Response to Editor**

**Simon Mudd:**

I have read both the responses to reviewers and the revised manuscript. Most of the issues raised by the reviewers have been addressed. The manuscript includes a new way to assess divide migration rates and has two interesting field sites. I think the paper will be well received by the community. However I think the paper could use one more round of revisions before it is accepted in ESURF. You can find my detailed comments in the annotated pdf.

**RESPONSE and CHANGES:**

Dear Editor,

Thank you very much for your handling our manuscript and the comments. In the revision, we have addressed each issue and modified associated texts and figures. We also check carefully the article to avoid the errors in the references. The line numbers in our response are from the manuscript with changes marked.

**Simon Mudd:**

The main issues are:

Some of the information on the methodology is distributed through the manuscript rather than collected in one place. It should be more clear that equations 4 and 8 are those used to calculate the divide migration rates. The methods should have these equations, and then an explanation of how each of the inputs into the equations are measured all located at the same place in the manuscript. At present this information is spread across a few sections so the reader has to do a lot of searching to find out how the numbers are calculated (everything readers need is there, it is just spread out over a number of sections). In the same place in the text, the authors should state the method of extracting the channel head (which is done with S-A plots), and explain how A is calculated (D8? D-infinity?). The gradient at the channel head should be described (in the text later it says the tangent to the profile is used: explain). For the chi-based gradient metric the authors should explain the start and end points of the segment used

to calculated the gradient (the upstream end is the channel head: where is the downstream end?) Finally, a bit more detail needs to be included about the calculation of the hillslope gradient (why is the hilltop ignored, why on the figures does the segment extend below the channel head)? I think clearing up these issues will help readers of the paper better understand the outcomes.

**RESPONSE and CHANGES:**

We have added a section (2.3 Parameter extraction) Lines 218-261.

In this new section, we make it more clear that in this study we use Eqs. 4 & 8 to calculate the divide migration rates, and how each of the inputs into the equations are measured: "Based on the high-resolution topography data, we first extract river channels and drainage divide, using a single-flow-direction algorithm (D8)." "According to the breaking point of the slope-area regression line, we obtain the value of the critical upstream drainage area ($A_{cr}$) of each river channel (Duvall et al., 2004)." "The slope of the channel head ($S_{ch}$) is calculated, according to the 100 m long channel on the river's long profiles around the channel head (50 m upstream and downstream)." "An elevation of the catchment outlet ($z_b$) can be assigned at the top part of the channel to make the elevation-$\chi$ profiles quasi-linear between the channel head and the outlet." "Topographic gradient ($\tan\alpha$ or $\tan\beta$) is calculated through the average slope (in the normal-divide direction) of the hillslope segment (not including the hilltop part, because of its lower gradient)."

We have changed the Figures 4 DGJ & 5 DJG to reveal the topographic gradient ($\tan\alpha$ or $\tan\beta$) is calculated above the channel heads.

**Simon Mudd:**

In addition, the results are quite contingent on the theta (or m/n) values, as highlighted by a reviewer. In the revision two new values are used (bringing those analysed to 0.35, 0.45, 0.55). There should be some text about how sensitive the results are to varying theta values, It would also help to show a chi-elevation plot of one of the larger basins with tributaries included so the reader can see if theta=0.45 is reasonable.

**RESPONSE and CHANGES:**

Thanks very much for this suggestion. According to this suggestion, we have added the following figure in the Discussion, to show why m/n = 0.45 is better than others.

[Figure]

**Simon Mudd:**

Finally one of the conclusions is that the new method does a better job than just assuming a critical drainage area. But we don't actually know if assuming a critical drainage area results in the same answers. There needs to be a discussion of this. It is pertinent because we know that the slope area method of extracting the channel head does not work very well, so this might be important if the method is sensitive to the location of the channel head.

There requests will involve some new figures and calculations, but they mostly clarify the existing analysis so I characterise my recommendations as constituting minor edits.

**RESPONSE and CHANGES:**

The main difference between our new methods and previous methods is that in our methods the critical drainage area ($A_{cr}$) are based on actual DEM data, and the two sides across the drainage divide can have different $A_{cr}$ values. We have added more discussion on the improvement of the critical drainage area in our new methods. (Lines 480, 488-491)

**Simon Mudd:**

Line 14: Change to "the sedimentary".

**RESPONSE and CHANGES:**

As suggested, we have added "the". (Line 14)

**Simon Mudd:**

Line 16: Change to "determine drainage divides' migration".

**RESPONSE and CHANGES:**

Changed as suggested. (Line 16)

**Simon Mudd:**

Line 49: differences.

**RESPONSE and CHANGES:**

Changed as suggested. (Line 49)

**Simon Mudd:**

Line 51: Change to "River long profiles have been used to study earthquake events".

**RESPONSE and CHANGES:**

Changed as suggested. (Line 51)

**Simon Mudd:**

Line 74: Add: Holly H. Young, George E. Hilley; Millennial-scale denudation rates of the Santa Lucia Mountains, California: Implications for landscape evolution in steep, high-relief, coastal mountain ranges. GSA Bulletin 2018;; 130 (11-12): 1809–1824. doi: https://doi.org/10.1130/B31907.1. One of the few papers that uses 10Be to try ans ascertain if divides are migrating. Not many people cite it because the title doesn't tell you what is in the paper.

**RESPONSE and CHANGES:**

Added as suggested. (Lines 74-75)

**Simon Mudd:**

Line 132: I would cite the 1972 Carson and Kirkby "Hillslope Form and Process" book here. typo. Should be Stock

**RESPONSE and CHANGES:**

Changed as suggested. (Line 133)

**Simon Mudd:**

Line 154: α or β: Formatting: these need to be italic throughout, since they are italic in the equation.

**RESPONSE and CHANGES:**

Changed as suggested. (Line 156)

**Simon Mudd:**

Line 157: This makes some assumptions about the geometry of the hillslope. It is basically assuming (I think) that the gradient upslope of the channel head remains constant if the channel head moves. I think this is probably not a bad assumption but it should be stated.

**RESPONSE and CHANGES:**

We have revised it make the representation of tan$\alpha$ and tan$\beta$ more clear. (Lines 158-160). We agree that the gradient remains constant in short time scale (kyr) when the channel head or the drainage divide move. Because the drainage divide migration rates calculated in this study are the instantaneous rate, we only need the current value of the topography gradient (tan$\alpha$ and tan$\beta$) across the drainage divide. Therefore, we are not discussing whether they changed during the divide migration.

**Simon Mudd:**

Line 158: Assuming the erosion coefficient.

**RESPONSE and CHANGES:**

Changed as suggested. (Line 161)

**Simon Mudd:**

Line 196: Explain where the outlet is taken. I think this is important. The inset figures in figure 3c show why it is important: the most distorted part of the channel is near the top (which you have highlighted in the figure). Say that you concentrate on the top part of the channel here.

**RESPONSE and CHANGES:**

Changed as suggested. (Lines 201-204)

"$z_b$ is the elevation of catchment outlet (at the top part of the channel to make the elevation-χ profiles quasi-linear between the channel head and the outlet)."

**Simon Mudd:**

Line 207-209: This is quite a nice method.

**RESPONSE and CHANGES:**

Thank you for your appreciation.

**Simon Mudd:**

Line 214-216: I will repeat a question from one of the reviewers: what is the uncertainty on these elevations. Do you know? The landscape is unvegetated so you should get a good point cloud. Were the elevations compared to GPS points? If this wasn't done it should be stated.

**RESPONSE and CHANGES:**

To be honest, we don't know the uncertainty of the elevations, and we didn't compared the elevations to the GPS points. We have added the statement in Lines 235-237. We believe that the data quality is sufficient for this study.

**Simon Mudd:**

Line 228: Note: If you look at the documentation for this DEM, it appears that it is a DEM that is derived from 30m SRTM data and then post-process to match the pixel spacing of the PALSAR radar data. It is not a "true" 12.5 DEM like TanDEM-X.

**RESPONSE and CHANGES:**

We have deleted "12.5 m resolution" in the full text.

**Simon Mudd:**

Line 245: I agree that these look captured and i am happy with this evidence. However, for me "barbed" means that the tributary joins the main stem at an oblique angle. That is, to be barbed the tributary needs to point upstream. Your examples don't do this. You could say tributaries have "abnormally high junction angles, which can suggest drainage capture".

**RESPONSE and CHANGES:**

We have changed the "barbed tributaries" to "captured channels" in Fig. 3 and the sentence in Line 290.

**Simon Mudd:**

Line 251-252: Can you specifically say if you determined the ksn from the chi profiles or from S-A analysis. I am fairly sure it is the former but it isn't stated.

**RESPONSE and CHANGES:**

In fact, both methods in this paper use ksn to calculate the erosion rate and the drainage-divide migration rate. The Channel-head-point method corresponds using S-A analysis to get ksn, while the Channel-head-segment corresponds using chi profiles to get ksn. When we get the ksn map in the whole Wutai Shan (Fig. 2B), we use the S-A analysis. We have added more description on how $k_{sn}$ is calculated. "The ksn is calculated based on S and A (k_sn=SA^(m/n)) extracted from ALOS DEM (downloaded from https://search.asf.alaska.edu/) using TopoToolbox (Schwanghart and Scherler, 2014)," (Lines 224-227, and Lines 365-367)

**Simon Mudd:**

Line 257-259: This sentence would sound better if it said "Middleton et al (2017) showed that the Quaternary...".

**RESPONSE and CHANGES:**

Changed as suggested. (Lines 302-304)

**Simon Mudd:**

Line 259: If you take my advice on the previous sentence, this becomes: "They showed, using low-temperature thermochronology, that...".

**RESPONSE and CHANGES:**

This sentence is changed to "Clinkscales et al. (2020) showed, using low-temperature thermochronology, that the time-averaged long-term throw rates in the late Cenozoic is about 0.25 mm/yr". (Lines 304-307)

**Simon Mudd:**

Line 270: refer to equations earlier in the paper.

**RESPONSE and CHANGES:**

Added. (Line 316)

**Simon Mudd:**

Line 281: Unclear. What do you mean by this?

**RESPONSE and CHANGES:**

We have changed this sentence. Lines 327-328.

How the $S_{ch}$ is calculated is now described in the Section 2.3. "The slope of the channel head ($S_{ch}$) is calculated, according to the 100 m long channel on the river's long profiles around the channel head (50 m upstream and downstream)." Lines 255-256.

**Simon Mudd:**

Line 281-282: Earlier in the paper you said that you were measuring the gradient in the segment downstream of the channel head. This seems like a different measurement. You need to state clearly what part of the first order channel you are measuring. We find out later that you do two analyses, using equations 4 and 8. Say that here. Basically I think the explanation of which of the equations you sue and then how you calculate either the slope or the k_sn (measured via the chi profile) in one place. And at that place explain why you use the two different methods.

**RESPONSE and CHANGES:**

We have added how the $S_{ch}$ is calculated in the Method (Lines 255-256). "The slope of the channel head ($S_{ch}$) is calculated, according to the 100 m long channel on the river's long profiles around the channel head (50 m upstream and downstream)."

**Simon Mudd:**

Line 284: Refer to figure 4 D,G, J here.

**RESPONSE and CHANGES:**

Added as suggested. (Line 332)

**Simon Mudd:**

Line 285-286: Explain why this is assumed.

**RESPONSE and CHANGES:**

We have added the explanation for this assumption. (Lines 333-334)

**Simon Mudd:**

Line 291-292: 1. It should say here how sensitive the results are to m.

**RESPONSE and CHANGES:**

We have added the how the result (migration rates) changes following to the different *m/n* values. (Lines 342-344)

**Simon Mudd:**

2. There should be a chi plot with tributaries so we can see which of the m values is more consistent with the profiles. So this would be of a larger basin than that shown in figure 4 with several tributaries and the profiles run with m = 0.35, 0.45, 0.55.

**RESPONSE and CHANGES:**

We have added a figure (Figure 6) in the Discussion, to show why m/n = 0.45 is better than others.

**Simon Mudd:**

Line 313: Say with what method.

**RESPONSE and CHANGES:**

Added in Lines 365-366.

**Simon Mudd:**

Fig. 5D: Can you explain somewhere why the channel head point is upstream of the lower point of the dotted line used to calculate alpha and beta?

**RESPONSE and CHANGES:**

We have changed the Figures 4 DGJ & 5 DJG to reveal the topographic gradient ($\tan\alpha$ or $\tan\beta$) is calculated above the channel heads.

**Simon Mudd:**

Line 423-428: 1. Are the answers different if you use A_cr $10^5$ m^2? Do we really need a very high quality DEM to use this method? 2. I ask this because you have used the S-A plot to get the channel head, which we know does not work very well (see the Clubb et al 2014 paper that is already cited in this manuscript). So does the extraction method make a big difference to the result?

**RESPONSE and CHANGES:**

This response is same with a previous response: The main difference between our new methods and previous methods is that in our methods the critical drainage area ($A_{cr}$) are based on actual DEM data, and the two sides across the drainage divide can have different $A_{cr}$ values. We have added more discussion on the improvement of the critical drainage area in our new methods. (Lines 480, 488-491)

**Simon Mudd:**

Line 476: typo. Should be Stock.

**RESPONSE and CHANGES:**

Changed. (Line 534)